# Does service heterogeneity have an impact on acute hospital length of stay in stroke? A UK-based multicentre prospective cohort study

Michelle Tørnes,[1] David McLernon,[2] Max Bachmann,[3] Stanley Musgrave,[3] Elizabeth A Warburton,[4] John F Potter,[3,5] Phyo Kyaw Myint,[1,5] on behalf of the Anglia Stroke Clinical Network Evaluation Study (ASCNES) Group

For numbered affiliations see end of article.

**Correspondence to**
Michelle Tørnes;
michelle.tornes@abdn.ac.uk

## ABSTRACT

**Objectives** To determine whether stroke patients' acute hospital length of stay (AHLOS) varies between hospitals, over and above case mix differences and to investigate the hospital-level explanatory factors.

**Design** A multicentre prospective cohort study.

**Setting** Eight National Health Service acute hospital trusts within the Anglia Stroke & Heart Clinical Network in the East of England, UK.

**Participants** The study sample was systematically selected to include all consecutive patients admitted within a month to any of the eight hospitals, diagnosed with stroke by an accredited stroke physician every third month between October 2009 and September 2011.

**Primary and secondary outcome measures** AHLOS was defined as the number of days between date of hospital admission and discharge or death, whichever came first. We used a multiple linear regression model to investigate the association between hospital (as a fixed-effect) and AHLOS, adjusting for several important patient covariates, such as age, sex, stroke type, modified Rankin Scale score (mRS), comorbidities and inpatient complications. Exploratory data analysis was used to examine the hospital-level characteristics which may contribute to variance between hospitals. These included hospital type, stroke monthly case volume, service provisions (ie, onsite rehabilitation) and staffing levels.

**Results** A total of 2233 stroke admissions (52% female, median age (IQR) 79 (70 to 86) years, 83% ischaemic stroke) were included. The overall median AHLOS (IQR) was 9 (4 to 21) days. After adjusting for patient covariates, AHLOS still differed significantly between hospitals (p<0.001). Furthermore, hospitals with the longest adjusted AHLOS's had predominantly smaller stroke volumes.

**Conclusions** We have clearly demonstrated that AHLOS varies between different hospitals, and that the most important patient-level explanatory variables are discharge mRS, dementia and inpatient complications. We highlight the potential importance of stroke volume in influencing these differences but cannot discount the potential effect of unmeasured confounders.

## INTRODUCTION

Stroke is the second leading cause of mortality and the third leading cause of disability in the

### Strengths and limitations of this study

► This is a comprehensive study that has used multicentre data to determine whether acute hospital length of stay of patients with stroke varies across hospitals in the UK, after adjustment for patient-level covariates, such as age, sex, prestroke and discharge modified Rankin Scale score, stroke type, residence prior to stroke, comorbidities and inpatient complications.

► With a wealth of detailed patient data, we were able to adjust for the important covariates, inpatient complications and discharge modified Rankin Scale score, which previous studies have not addressed when investigating hospital-level factors.

► Although hospital-level effect estimates were not calculated due to the limited hospital sample size of eight, we explored these factors descriptively and adjusted for clustering by including hospital as a fixed-effect.

► Although National Institute for Health Stroke Scale, which is used to measure the severity of stroke, is known to be associated with acute hospital length of stay, we were unable to take this variable into account since it was only calculated on admission for patients who were potentially eligible for thrombolysis, and would have introduced information bias.

world, with a global incidence of 16.9 million in 2010.[1 2] While acute hospitalisation for stroke in the US has been estimated at a cost of $31 667 per patient, total direct stroke-related annual medical costs are expected to triple, from $71.6 billion in 2012 to $184.1 billion by 2030.[3 4]

Considerable differences in stroke-related outcomes exist worldwide, with the highest age-standardised stroke-related mortality and disability adjusted life-years rates observed in Russia and Eastern European countries.[1] Stark regional disparities within countries are also apparent. In the UK, for example, there exists

a clear north–south divide where the lowest stroke-related mortality rates are observed almost exclusively in the South of England.[5] Such differences in outcomes likely reflect underlying stroke incidence rates and variations in exposure to relevant risk factors.[5][6] However, we and others have demonstrated that some of the differences in poststroke survival have also been explained by disparities in available resources and medical care.[7–11] Studies assessing the effect of stroke care heterogeneities have largely focused on mortality as the primary outcome.

However, it is possible that heterogeneities in stroke care also impact other important stroke-related outcomes, such as a patient's acute hospital length of stay (AHLOS). To date, researchers have mainly identified patient-related determinants of AHLOS,[12–15] with little exploration into hospital-level influences. Of the few studies that have investigated hospital-level variance, factors such as hospital type, size, teaching status and location have been implicated in partially explaining differences in AHLOS.[12][16–19] None such studies have been conducted in a UK National Health Service (NHS) setting.

During acute hospitalisation, AHLOS is the main driver of acute care costs.[20] Determining the hospital-level factors influencing AHLOS therefore provides invaluable information to service providers and policymakers who can develop optimal management strategies and enhance patient care by minimising service deficiencies, costs and bed shortages.

The aim of this study was to investigate whether there are variations in stroke patients' AHLOS which can be partly explained by heterogeneities in characteristics of stroke care between hospitals in a UK NHS setting. We also aimed to explore which hospital-level factors explain such hospital variations in AHLOS.

## METHODS
### Study design
A multicentre prospective cohort study was conducted at eight acute NHS Trusts within the Anglia Stroke & Heart Clinical Network (ASHCN) which covers the three counties of Suffolk, Norfolk and Cambridgeshire, in the East of England with a catchment population of approximately 2.5 million. The detailed study protocol has previously been published (see online supplementary file 1).[21]

### Participants
The study population included all patients, aged 18 years or older, admitted to any of the eight hospitals within the ASHCN diagnosed with stroke by an accredited stroke physician between October 2009 and September 2011. Stroke was defined as a focal neurological impairment of sudden onset and lasting more than 24 hours (or leading to death) as a consequence of an intracerebral ischaemic or haemorrhagic event. This definition excludes diagnoses of transient-ischaemic attacks (TIAs), subdural haematomas and subarachnoid haemorrhages. Stroke diagnosis was confirmed in all patients with stroke through

cerebral imaging (either using CT or MRI). Diagnoses by the stroke physician were coded using International Classification of Diseases-10. The study sample was systematically selected to include all consecutive patients with stroke admitted every third month of this 2-year period, resulting in a total of eight study months and a sample size of 2656. The robustness of this sampling technique has been confirmed.[22]

### Participant hospitals
The participating hospitals, although part of the same network, do not coordinate the care of patients or work together to provide regional care. They are independent NHS Trusts that serve their local communities and therefore are individually responsible for managing patients with stroke. Admission, transfer and discharge policies should be similar across these hospitals. There are also no known differences in access to rehabilitation, home care or nursing homes.

Stroke services available at each site should be proportionate to the hospital's catchment population. However, as stroke volumes differ, some hospitals may experience greater pressure on their resources and facilities than others. Access to available resources also varies between the hospitals, with some providing onsite rehabilitation, neurosurgery and vascular surgery. Palliative care management may also differ between the sites.

### Data collection
Clinical teams responsible for the care of patients with stroke in each of the hospitals prospectively recorded individual patient data. Patient data routinely collected by each participating site for the ASHCN surveys was used in this study. Additional baseline patient and outcome data were also retrieved from case records, discharge summaries and patient administrative systems by the clinical teams. Data were anonymised and sent to the ASHCN coordinating centre where it was collated and sent to the research team. Any identifiable patient information was held only at the local NHS Trusts—the network and investigators did not have access to these details.

Data on health service characteristics were collected from clinical leads or service managers at each stroke unit and updated every 6 months over the 2-year study period by research staff.[21] No major changes in health service characteristics occurred during the study data collection period. Some changes that did occur included: minor fluctuations in staffing levels, number of non-stroke patients treated on the stroke unit and number of patients with stroke treated outside the stroke unit. In the final year of study, hospital 5 introduced a further CT scanner, increasing their total to three. Furthermore, for hospitals 5 and 6 some reconfigurations from acute stroke unit beds to hyperacute stroke unit beds were made. Hospital 4 also introduced hyperacute stroke unit beds in the final year of study and increased the number of acute stroke unit beds available. We have accounted for these fluctuations

by calculating and reporting the weighted average across the four study periods for these measures.

## Definition of variables

Our outcome measure, AHLOS, was treated as a continuous variable and defined as the number of days from, and including, the patients' date of hospital admission to their date of discharge or death, whichever came first.

Patient-level covariates adjusted for were: age (treated as a continuous variable), sex, prestroke modified Rankin Scale (mRS) as an indicator of prestroke frailty, prestroke residence status, stroke type, Oxfordshire Community Stroke Project (OCSP) (a stroke classification system), presence or absence of lateralisation signs, acute inpatient complications (such as another stroke, pneumonia, urinary tract infection (UTI), seizures, myocardial infarction, acute coronary syndrome), established comorbidities (including previous stroke/TIA, previous myocardial infarction or ischaemic heart disease, previous cancer), presence of other relevant comorbidities (including diabetes mellitus, dementia, hypercholesterolaemia, hypertension, cancer, depression, rheumatoid arthritis and chronic obstructive pulmonary disease), day and season of admission and discharge mRS (including in-hospital death). An inpatient complication was defined as any disease, disorder or condition that developed after the index stroke that is, during the acute admission, whereas comorbidities were defined as those that were known to have occurred prior to stroke.

Independent hospital-level variables of interest were: hospital type (secondary or tertiary), hospital stroke volume (mean number of patients with stroke admitted and treated in hospital per month), presence of vascular surgery onsite, distance to neurosurgical facility, onsite rehabilitation service provision, presence of an early supported discharge scheme, number of full-time equivalent (fte) staff per five beds (senior doctors and junior doctors available during weekdays, healthcare associates and nurses, occupational therapists, physiotherapists and speech and language therapists), number of total beds present on the stroke unit per 100 stroke admissions, total number of hospital beds per CT scanner, number of non-stroke patients treated daily on the stroke unit per five beds, number of patients with stroke treated daily on wards outside the stroke unit per day per five beds and the mean index of multiple deprivation (IMD) of the county in which each hospital serves.

In NHS England, hospitals are either termed secondary or tertiary, depending on the level of specialist service provided. Tertiary hospitals provide more specialised care in larger, regional or national centres compared with their secondary counterparts for example, neurosurgery unit where smaller units are not viable nor practical. These more centralised hospitals are usually dedicated in providing superspecialty care beyond sub-specialty (eg, neuro-endocrine surgery is a superspecialty of neurosurgery which is a subspecialty of the specialty of surgery), and therefore have access to more advanced equipment

and expertise specific to the conditions in which it subspecialises. This does not apply to stroke directly, but it is relevant for those who have stroke and require neurosurgical intervention.

Five bed days was used as the denominator as this is how the 2016 national clinical guidelines for stroke reports the recommended staffing levels for UK stroke units, and therefore provides for a comparison.[23]

The IMD score was used as an aggregate measure of socioeconomic status in this study. This measure is based on several domains, including income, employment, education, health, crime, barriers to housing and services and the living environment, that are believed to provide an indication of deprivation. To assign an IMD score, England is subdivided into 32 844 smaller areas, with a score of 1 representing the area in England that is considered to be the most deprived and a score of 32 844 the least deprived.[24] In our study, we have taken the mean 2010 IMD scores of the areas that make up the counties of Suffolk, Norfolk and Cambridgeshire and assigned these to each of the hospitals to which they are located.[25]

We believe processes of care measures are intermediate variables that lie on the casual pathway between hospital-level factors and patient outcomes of stroke.[10] As such, we did not adjust for these covariates in the analyses. Including them in our regression model could otherwise lead to over-adjustment bias.[26 27]

## Statistical analyses

Data were available from only eight hospitals which is below the suggested critical number required to reliably estimate hospital effects through multilevel modelling.[28] Therefore, a single-level multiple linear regression model using ordinary least squares was conducted with hospital as a fixed-effect and AHLOS as the outcome. To qualify for inclusion in the multivariable model, patient-level variables had to have a p value <0.3 in univariable analysis. The standardised residuals of the model were positively skewed. However, a logarithmic transformation of AHLOS subsequently removed the skewness. Before reporting, we transformed the predicted logarithmic AHLOS values back to AHLOS, with exponentiated regression coefficients representing geometric means of AHLOS.

To explore hospital-level factors, we plotted the hospital intercept estimates of AHLOS from the regression model (mean baseline AHLOS of each hospital), against the hospital-level characteristics of interest. This is the recommended method to use on clustered data to explore hospital effects when the number of higher level units is small and hence are not interpretable in likelihood estimation.[28 29]

## Sensitivity analyses

Due to limited resources, hospital 2 failed to collect data for the full study period. Patient-level data were only collected in this hospital for October 2009 and January 2010, culminating in a small number of stroke cases for analysis (n=16). To investigate whether this small cluster

may affect our results, we performed a sensitivity analysis excluding hospital 2.

Furthermore, although we collected patient data on discharge destination, we did not include this as a covariate in our multiple regression model due to issues of multi-collinearity with discharge mRS (both had categories for inpatient death). We hypothesised that discharge mRS could more readily explain a patient's AHLOS indirectly through discharge destination (ie, more severe disability increases the risk of institutionalisation which prolongs AHLOS due to associated waiting lists) and directly through patient recovery (ie, a patient with more severe disability will likely take longer to recover than a patient with no disability, meaning it will take longer for a safe patient discharge). If we were to include discharge destination instead, AHLOS variance due to differences in disability and recovery time among patients with the same discharge placement would not be taken into account. To check the impact of excluding discharge destination on our findings, we have performed a further sensitivity analysis replacing discharge mRS with discharge destination in our multiple regression model.

## Multiple imputation

To increase power and reduce potential bias of complete case analysis, we performed multiple imputation by chained equations using the MICE package in R.[30] All the independent variables of interest, AHLOS and a number of auxiliary variables (ie, variables in our dataset that were not used in our model) (see table S1 in the online supplementary file 2) informed the imputation. Sixty-four datasets were imputed as the inclusion of auxiliary variables increased the casewise missingness to 64%. Each dataset was pooled together using Rubin's rules.[31] The distribution of sample characteristics between individuals with complete and incomplete data were compared using the appropriate hypothesis testing. Complete case analysis was also conducted so that any differences in results from the multiple imputation analysis could be reported.

All analyses were performed using R V.3.3.1 for Windows.[32]

## Patient and public involvement

The project was managed by project leader (PKM) who worked in close partnership with the project group of the study and the project steering group. The project steering group included public and patient representatives, recruited through patient and public involvement in research (PPIRes). PPIRes members were invited to attend research steering group meetings over the study duration to oversee the project.

## RESULTS
## Description of sample characteristics

Of the 2656 patients admitted consecutively to the eight NHS hospitals during the inclusion period with an initial diagnosis of stroke, 278 were excluded for the following reasons: eventually diagnosed with a condition other than stroke (n=179), transferred between hospitals (both among the eight study hospitals and from or to outside the region) (n=101), had missing data for admission and discharge dates (n=8). This left a total of 2233 patients for the study analysis (figure 1).

The median age (IQR) of our cohort was 79 (70 to 86) years, 52% were female and 83% had an ischaemic stroke (table 1). The distributions of patient characteristics appeared to vary between hospitals (see table S2 in the online supplementary file 2). Although there were low proportions of missing data for each independent variable (table 1), this compounded to 33% of patients having at least one variable missing. Hospital 4 did not collect data on prestroke mRS and 30 cases from hospital 3 had missing data on all comorbidities. Patients with complete data were less likely to have a haemorrhagic stroke, be institutionalised prior to stroke and have an inpatient death, and more likely to have had a previous stroke or TIA, have hypercholesterolaemia, hypertension, rheumatoid arthritis, have a lacunar stroke and have a discharge mRS of 6, than patients who had at least one missing variable. However, there were no significant differences in other patient characteristics such as age, sex, prestroke mRS score, brain lateralisation, inpatient complication and admission timing between the two groups (see table S3 in the online supplementary file 2).

## Hospital service characteristics

Service characteristics of each hospital are outlined in table 2, with median AHLOS.

After standardisation, by taking account of stroke admission volume, number of stroke unit beds and size of hospital, there was still extensive heterogeneity in bed capacity, staffing levels and the number of CT scanners provided at each hospital, respectively. Variations between hospitals also existed in terms of service and facility provision. For example, a number of hospitals provided rehabilitation care, neurosurgery or vascular surgery onsite, while others did not. The overall median AHLOS (IQR) was 9 (4 to 21) days and there appeared to be crude variations in this outcome between hospitals.

## Univariable linear regression

In univariable linear regression (see table S4 in the online supplementary file 2), patients who were older, female, had previous cancer, a previous stroke, had diabetes mellitus, had dementia, had a prestroke or discharge mRS score greater than 0, had a OCSP other than a lacunar infarct, had an inpatient complication, were living independently at home without formal care (compared with those who had formal care) prior to stroke, or were a winter admission had a significantly longer AHLOS (p<0.05). Patients who had a haemorrhagic stroke, hypercholesterolaemia, or showed no signs of brain lateralisation were all shown to be significantly associated with a shorter AHLOS (p<0.01).

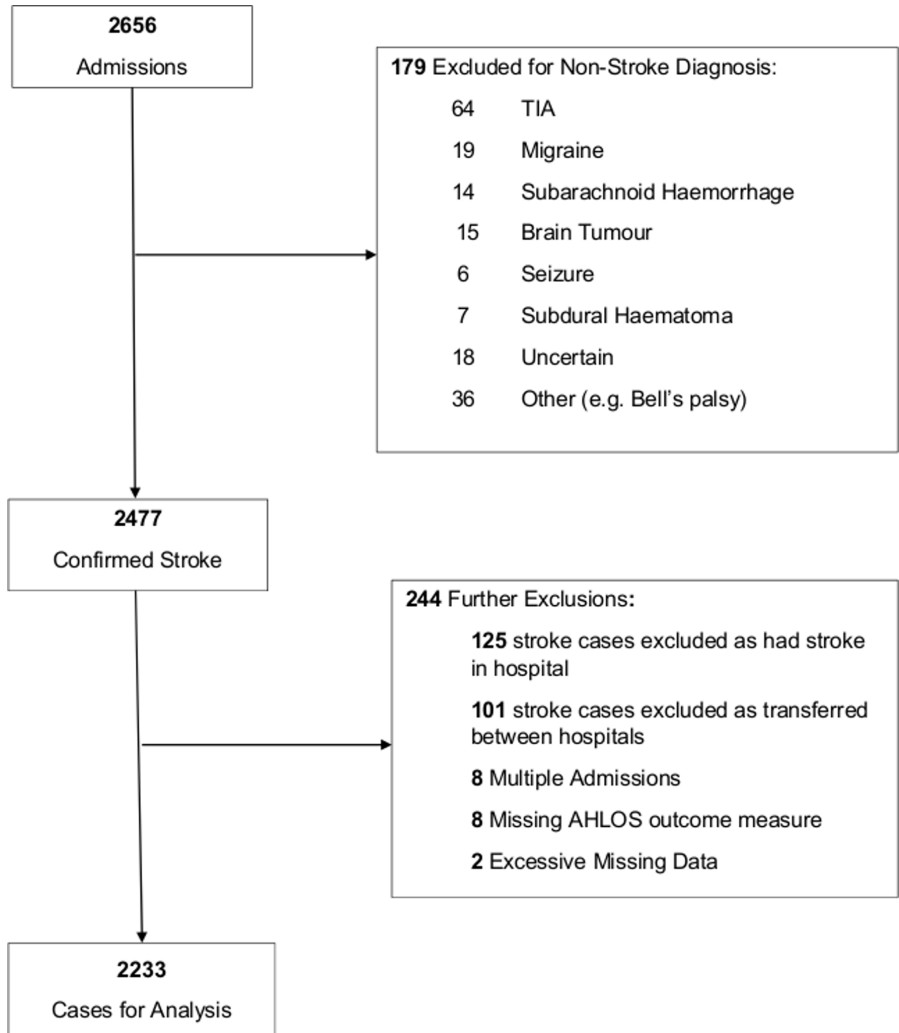

**Figure 1** Flow chart of patient participation inclusion and exclusion for study analysis. AHLOS, acute hospital length of stay; TIA, transient-ischaemic attack.

The strongest associations with AHLOS were seen for inpatients who developed a complication, who had a prestroke mRS score of 3, who were admitted to hospital 2 or who had a discharge mRS score of ≥2. Inpatient complications were associated with twice as long an AHLOS compared with those without a complication. Similarly, patients with a prestroke mRS score of 3 were 94% more likely to have a longer AHLOS than those with an mRS of 0. Patients admitted to hospital 2 had 2.69 times the AHLOS of those admitted to hospital 1. Compared with patients with a discharge mRS score, those with a score of 2, 3, 4 or 5 had over a 2, 3, 4 and 5-fold increase in AHLOS, respectively. Unsurprisingly, discharge mRS score appeared to explain the majority of AHLOS variance ($R^2$=31.1%).

Being hypertensive, having a history of a myocardial infarction or ischaemic heart disease, having previously had a TIA, having active cancer, depression, rheumatoid arthritis or chronic obstructive pulmonary disease were not shown to be significantly associated with AHLOS. Furthermore, admissions to hospitals 6 and 8 were also not shown to be significantly associated with a difference in AHLOS compared with hospital 1 admissions.

### Multiple linear regression

Multiple linear regression results for AHLOS are summarised in table 3 and shows that 42.7% of the variation in AHLOS has been explained. Sex, recurrent stroke, diabetes mellitus, hypercholesterolaemia, previous cancer, a prestroke mRS score of 1 to 3 (with reference to a score of 0) and living at home independently without formal care prior to stroke were no longer statistically associated with AHLOS in multiple regression (p>0.05). Furthermore, being admitted to hospital 3 or 4 as opposed to hospital 1 were no longer associated with a significant difference in AHLOS. No variables included from the univariable analysis with p>0.05 became statistically significant in the multivariable analysis, except for living in an institution prior to stroke which was associated with a 19% reduced AHLOS compared with those living independently without formal care. Developing an inpatient complication and having a discharge mRS score

| Table 1 | Sample characteristics of patients included in analysis (n=2233) and missing data | |
|---|---|---|
| **Patient characteristic** | **Median (IQR) or no. (%)** | **Missing data (%)** |
| Age, years | 79 (70 to 86) | 2 (0.1) |
| Sex, female | 1165 (52) | 2 (0.1) |
| Recurrent stroke* | 448 (20) | 30 (1) |
| Diabetes mellitus* | 370 (17) | 30 (1) |
| Dementia* | 207 (9) | 30 (1) |
| Hypercholesterolaemia* | 355 (16) | 30 (1) |
| Hypertensive* | 1483 (66) | 30 (1) |
| Myocardial Infarction or ischaemic heart disease* | 517 (23) | 30 (1) |
| TIA* | 340 (15) | 30 (1) |
| Previous cancer* | 195 (9) | 30 (1) |
| Active cancer* | 137 (6) | 30 (1) |
| Depression* | 137 (6) | 30 (1) |
| Rheumatoid arthritis* | 154 (7) | 30 (1) |
| COPD* | 116 (5) | 30 (1) |
| Prestroke mRS score | | 442 (20) |
| 0 | 914 (41) | |
| 1 | 335 (15) | |
| 2 | 191 (9) | |
| 3 | 184 (8) | |
| 4 and 5 | 167 (7) | |
| Prestroke residence | | 51 (2) |
| Independent living with formal care | 210 (9) | |
| Independent living without formal care | 1752 (78) | |
| Institution | 220 (10) | |
| Ischaemic stroke | 1864 (83) | 96 (4) |
| Oxford Community Stroke Project Classification | | 260 (12) |
| LACS | 503 (23) | |
| PACS | 784 (35) | |
| POCS | 279 (12) | |
| TACS | 407 (18) | |
| No brain lateralisation | 244 (12) | 167 (8) |
| Inpatient complication* | 655 (29) | 0 (0) |
| Discharge mRS score | | 50 (2) |
| 0 | 260 (12) | 329 (15) |
| 1 | 352 (16) | |
| 2 | 212 (9) | |
| 3 | 291 (13) | |
| 4 | 238 (11) | |
| 5 | 137 (6) | |
| 6 | 414 (19) | |

Continued

| Table 1 | Continued | |
|---|---|---|
| **Patient characteristic** | **Median (IQR) or no. (%)** | **Missing data (%)** |
| Winter admission | 1159 (52) | 0 (0) |
| Weekend admission | 614 (27) | 0 (0) |

*No information was assumed to indicate absence of condition or complication.
COPD, chronic obstructive pulmonary disorder; LACS, lacunar anterior circulation stroke; mRS, modified Rankin Scale; PACS, partial anterior circulation stroke; POCS, posterior circulation stroke; TACS, total anterior circulation stroke; TIA, transient ischaemic attack.

between 2 and 5 were still strongly positively related to AHLOS. After adjusting for patient covariates, AHLOS was still shown to significantly differ between hospitals, with the shortest and longest AHLOS observed for hospitals 5 and 2, respectively.

There were no obvious differences between the results using complete cases only (see tables S5–6 in the online supplementary file 2) and multiple imputation.

### Graphical exploratory analysis

Mean baseline AHLOS of each hospital (estimated from the multiple regression model) was plotted against hospital stroke volume and clustered by hospital type in figure 2. It appears that hospitals (of either type) that have larger stroke volumes have a shorter AHLOS than those with smaller stroke volumes when patient covariates are taken into account. To note also, hospital 2 deviates largely from all the other hospitals with respect to the number of patients with stroke treated daily outside the stroke unit (see figure S1 in the online supplementary file 2).

No discernible patterns were seen for mean baseline hospital AHLOS and staffing levels, surgery facilities, number of non-stroke patients treated on the stroke unit, bed numbers and IMD score (see figures S2–15 in the online supplementary file 2).

### Sensitivity analyses results

Excluding hospital 2 in our first sensitivity analysis did not alter our results (see table S7 in the online supplementary file 2). For our second sensitivity analysis, although the results were similar, the amount of variance explained reduced from an $R^2$ value of 42.7% to 40%. Furthermore, significant differences in AHLOS were shown between our reference hospital and hospitals 3 and 4, which was not shown in our main analysis (see table S8 in the online supplementary file 2).

### DISCUSSION

This multicentre cohort study has demonstrated that substantial heterogeneities exist in stroke hospital service and staff provision across three counties in the East of England. After adjusting for patient characteristics and

**Table 2** Hospital characteristics per individual hospital self-reported by clinical leads or service managers at each hospital

| Hospital characteristics | 1 | 2 | 3 | 4 | 5 | 6 | 7 | 8 |
|---|---|---|---|---|---|---|---|---|
| General characteristics | | | | | | | | |
| Catchment population | 400 000 | 160 000 | 350 000 | 230 000 | 680 000 | 300 000 | 240 000 | 275 000 |
| Hospital type | Tertiary | Secondary | Secondary | Secondary | Tertiary | Secondary | Secondary | Secondary |
| Hospital stroke volume (no. of ASCNES admissions per month) | 52 | 13 | 46 | 19 | 88 | 57 | 35 | 31 |
| Facilities and services | | | | | | | | |
| No. of hospital beds | 1000 | 304 | 800 | 500 | 1237 | 611 | 488 | 460 |
| No. of stroke unit beds (per 100 admissions) | 71 | 77 | 54 | 138 | 41 | 55 | 83 | 65 |
| No. of hospital beds per CT scanners | 500 | 304 | 400 | 250 | 518 | 306 | 244 | 230 |
| Distance to vascular surgery (miles) | 0 | 18 | 0 | 25 | 0 | 0 | 43 | 30 |
| Distance to neurosurgery (miles) | 0 | 18 | 58 | 89 | 61 | 38 | 48 | 30 |
| Rehabilitation provision | Onsite | Onsite | Offsite | Offsite | Offsite | Onsite | Offsite | Onsite |
| Early supported discharge provision | No | Yes | No | Yes | Yes | Yes | No | No |
| Stroke unit staffing levels* | | | | | | | | |
| Senior doctors† | 0.34 | 0.25 | 0.49 | 0.47 | 0.42 | 0.31 | 0.62 | 0.87 |
| Junior doctors† | 0.55 | 0.65 | 0.72 | 0.59 | 0.56 | 0.64 | 0.12 | 0.25 |
| Healthcare associates and nurses (band 5–7) | 9.2 | 8 | 6 | 7.4 | 7 | 5.3 | 6.5 | 10 |
| Physiotherapists (band 2–8) | 0.55 | 1 | 0.79 | 0.4 | 0.91 | 0.78 | 0.69 | 1 |
| Occupational therapists (band 3–8) | 0.49 | 0.5 | 1.4 | 0.59 | 0.6 | 0.58 | 0.52 | 1.1 |
| Speech and language therapists | 0.39 | 0.15 | 0.2 | 0.18 | 0.35 | 0.03 | 0.26 | 0.1 |
| No. of non-stroke patients treated daily on stroke unit (per five stroke unit beds) | 0.27 | 0 | 0.10 | 0.47 | 0.05 | 0.31 | 0.17 | 0 |
| No. of patients with stroke treated daily outside stroke unit (per five stroke unit beds) | 0.14 | 5 | 0 | 0.30 | 0.01 | 0.41 | 0 | 0 |
| Median AHLOS (IQR) | 8 (4 to 20) | 29 (24 to 42) | 11 (5 to 27) | 14 (4 to 30) | 8 (4 to 14) | 10 (5 to 22) | 11 (6 to 23) | 7 (3 to 20) |

*Number of fte staff per five stroke unit beds (weighted average for the four study periods taken). NHS banding refers to the pay scale system of healthcare staff in the UK and relates to their level of experience. Higher bands reflect higher pay and experience.
†Weekday numbers only.
AHLOS, acute hospital length of stay; ASCNES, Anglia Stroke Clinical Network Evaluation Study.

confounding factors, we have shown that AHLOS significantly differed between hospitals. This suggests that the heterogeneities we see in stroke care between hospitals have an effect on AHLOS of these patients. It also appears from our exploratory analysis that the volume of patients with stroke admitted to hospital may play a role in partially explaining these hospital-level AHLOS differences. Furthermore, the large deviation in AHLOS of hospital 2 seems to be related to the number of patients with stroke that were not being treated in their stroke unit.

In agreement with our findings, two previous studies in Japan and Denmark have shown that hospitals with larger stroke volumes are those in which AHLOS is shorter.[16 19] The reason larger volume hospitals lead to more favourable outcomes may simply be down to the fact that 'practice makes perfect' that is, the stroke physicians in these hospitals treat a greater number of patients and hence are more experienced and able to deliver higher quality care.[16 33 34] Svendsen et al, also demonstrated that patients with stroke admitted to large-volume stroke units have significantly greater odds of being treated and assessed earlier than those admitted to smaller-volume units, which could also explain their better outcomes.[19]

To translate these findings into practice may mean the centralisation of stroke services. Although this has been successfully implemented in urban centres such as Manchester and London,[35 36] this may not be feasible in more rural areas where travel times would compromise timely thrombolysis treatment.[10 37] Alternatively, a hub and spoke model of stroke care could be introduced whereby patients are first treated in their local hospital, and when stable for transfer are redirected to larger hub centres where they can gain access to more specialised care.[38] Specifically, patients with severe stroke or with complex health needs could be redirected to these better performing larger-volume centres.

Any recommendations that would lead to changes in stroke volume for the benefit of a reduced AHLOS should not compromise the quality of care. However, it has previously been reported that larger stroke volumes are independently associated with a lower risk of mortality.[10 11 39 40] Therefore, modifying this hospital factor may not only lead to a potential modest decrease in inpatient costs and more available bed days but could also be beneficial to the health outcomes of patients.

**Table 3** Multiple regression analysis for AHLOS (n=2233; $R^2$=42.7%)

| Patient characteristic | $e^{\beta}$* | 95% CI* | P value |
|---|---|---|---|
| Age, years | 1.01 | 1.00 to 1.01 | <0.001 |
| Sex, female | 1.01 | 0.94 to 1.09 | 0.79 |
| Recurrent stroke | 1.03 | 0.94 to 1.12 | 0.57 |
| Diabetes mellitus | 1.06 | 0.97 to 1.17 | 0.21 |
| Dementia | 1.28 | 1.12 to 1.46 | <0.001 |
| Hypercholesterolaemia | 0.94 | 0.85 to 1.05 | 0.27 |
| Myocardial infarction or ischaemic heart disease* | 1.00 | 0.92 to 1.09 | 0.98 |
| Previous cancer | 1.12 | 0.99 to 1.27 | 0.08 |
| COPD | 0.90 | 0.77 to 1.06 | 0.21 |
| Prestroke mRS score (reference 0) | | | <0.001 |
| 1 | 1.06 | 0.95 to 1.19 | 0.28 |
| 2 | 0.90 | 0.77 to 1.04 | 0.15 |
| 3 | 0.94 | 0.80 to 1.11 | 0.47 |
| 4 and 5 | 0.71 | 0.59 to 0.86 | <0.001 |
| Prestroke residence (reference independent living without formal care) | | | <0.001 |
| Independent living with formal care | 1.07 | 0.94 to 1.23 | 0.92 |
| Institution | 0.81 | 0.69 to 0.95 | 0.01 |
| Haemorrhagic stroke | 0.80 | 0.71 to 0.90 | <0.001 |
| Oxford Community Stroke Project Classification (reference LACS) | | | <0.001 |
| PACS | 1.30 | 1.18 to 1.42 | <0.001 |
| POCS | 1.34 | 1.18 to 1.53 | <0.001 |
| TACS | 1.29 | 1.13 to 1.48 | <0.001 |
| No brain lateralisation | 0.85 | 0.75 to 0.96 | 0.01 |
| Inpatient complication | 1.70 | 1.56 to 1.85 | <0.001 |
| Discharge mRS score (reference 0) | | | <0.001 |
| 1 | 1.15 | 1.01 to 1.31 | 0.04 |
| 2 | 1.74 | 1.50 to 2.04 | <0.001 |
| 3 | 2.70 | 2.32 to 3.13 | <0.001 |
| 4 | 3.51 | 2.98 to 4.14 | <0.001 |
| 5 | 5.07 | 4.19 to 6.14 | <0.001 |
| 6 | 1.24 | 1.05 to 1.48 | 0.01 |
| Winter admission | 1.15 | 1.08 to 1.24 | <0.001 |
| Weekend admission | 1.03 | 0.95 to 1.11 | 0.50 |
| Hospital (reference 1) | | | <0.001 |
| 2 | 2.09 | 1.38 to 3.17 | 0.001 |
| 3 | 1.07 | 0.94 to 1.22 | 0.29 |
| 4 | 1.08 | 0.90 to 1.31 | 0.40 |
| 5 | 0.78 | 0.69 to 0.87 | <0.001 |

Continued

**Table 3** Continued

| Patient characteristic | $e^{\beta}$* | 95% CI* | P value |
|---|---|---|---|
| 6 | 0.93 | 0.81 to 1.07 | 0.33 |
| 7 | 1.15 | 1.00 to 1.32 | 0.05 |
| 8 | 0.82 | 0.70 to 0.94 | 0.01 |

*β estimates and 95% CIs were calculated for predicted log AHLOS. Prior to reporting they were transformed back to AHLOS through exponentiation and represent geometric mean AHLOS.

AHLOS, acute hospital length of stay; COPD, chronic obstructive pulmonary disorder; LACS, lacunar anterior circulation stroke; mRS, modified Rankin Scale; PACS, partial anterior circulation stroke; POCS, posterior circulation stroke; TACS, total anterior circulation stroke.

The large variation in AHLOS between hospital 2 and the other hospitals in our study is also interesting to note. This coincides with a stark contrast in the number of patients with stroke that were not treated in a stroke unit in hospital 2 compared with the others. It could therefore be surmised that the large deviation in AHLOS of this hospital is driven by a lack of access to stroke unit care. This would be unsurprising given that stroke unit care has been consistently found to improve outcomes, including AHLOS, possibly due to a greater intensity of physiological monitoring, therapy and early mobilisation implemented in these discrete units.[41–44]

Other hospital-level factors that have been shown to influence a stroke patient's AHLOS include hospital size and teaching status.[12 16–18] However, these relationships were not apparent in our exploratory analysis. To investigate these and other hospital characteristics further, we require a larger sample of hospitals. This issue with sample size is also apparent when we study hospital 8 which, although has one of the lowest AHLOS, also has one of the smallest volumes of stroke patients in the study, and therefore contradicts our previous finding. Such a discrepancy is likely a reflection of the small number of hospitals assessed, as there are likely to be several competing factors playing a role in determining hospital-level AHLOS variance. For example, although hospital 8 has one of the lowest stroke volumes, it has the highest number of fte senior doctors, healthcare associates and nurses and physiotherapists per five beds, and the lowest number of hospital beds per CT scanners out of all the hospitals studied. Staffing levels may be what is responsible for this supposed contradiction as they are likely to be an important determinant of AHLOS, given that higher nurse: bed ratios have been shown to be important in reducing other stroke-related outcomes, such as mortality.[7 10]

Although not the focus of our study, we have also demonstrated several important patient variables that influence AHLOS, specifically discharge mRS, having dementia or having an inpatient complication. Other researchers have confirmed the strength of these

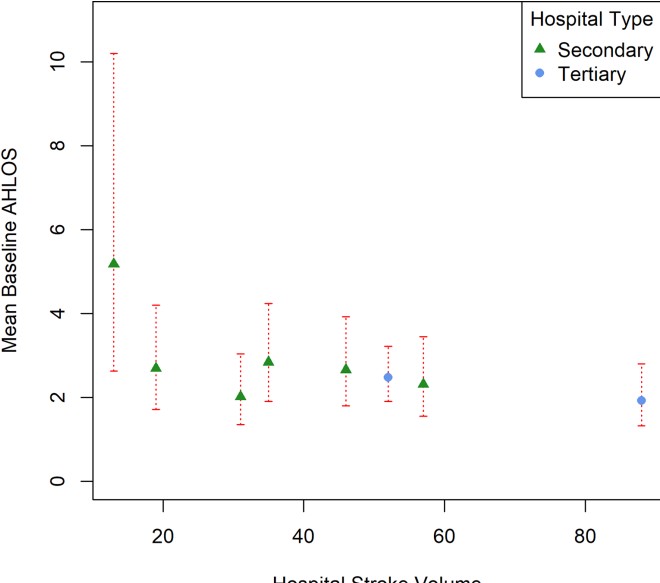

**Figure 2** Model estimates of mean baseline acute hospital length of stay (AHLOS) per hospital (in days) against hospital stroke volume and clustered by hospital type with 95% CIs. Multiple regression model was adjusted for patient covariates that had a p value <0.3 in univariable analysis.

relationships. For example, Fujinio *et al*, showed that mRS before discharge was associated with a difference in 5.77 days in AHLOS,[16] while another study showed that dementia increased AHLOS by 6.5 days.[14] Complications such as congestive heart failure, falls, UTI and pneumonia have also been shown to prolong a patient's AHLOS.[15 45 46] It is therefore important for any future studies exploring hospital-level factors to properly adjust for these patient variables, in addition to National Institute for Health Stroke Scale (NIHSS) which is another important covariate. This is especially pertinent given that the studies examining hospital-level factors and AHLOS in stroke to date have failed to adjust for these specifically. Finally, our findings in relation to other patient factors such as age, sex, stroke type and prestroke residence are in general agreement with other literature.[12–14 47 48]

The main strength of our study is its prospective design and the detailed patient-level data we obtained. This allowed us to gain a better understanding of the extent to which the variation in AHLOS exists over and above patient characteristics. We have optimised the use of available NHS data as the starting block for informing future pragmatic real-world setting randomised controlled trials by first identifying potential health service factors that could lead to important interventions. Furthermore, the findings of this study can presently be used to inform clinicians, healthcare service providers, commissioners and policy makers as to where improvements can be achieved in stroke care. The robust statistical analysis has allowed easy and quick visualisation of notable patterns in the dataset and provides a candid assessment of the research objectives by considering the limits of inference due to the small number of hospitals. Multiple imputation has

also reduced potential bias that may have otherwise been introduced from complete case analysis alone.

The major limitation of this study was the small number of hospitals that has restricted the conclusions we can make from our exploratory analysis of hospital characteristics. Furthermore, although NIHSS and a patient's discharge destination has been shown to be associated with stroke patients' AHLOS,[14 20] they were excluded as covariates from the main analysis. As NIHSS scores were only calculated for those who were potentially eligible for thrombolysis at the time of our study, the incompleteness was not missing at random and would have introduced information bias into our results. As discharge mRS and discharge destination both included a categorical factor representing inpatient death only one of these variables could be included into the analysis due to issues of multicollinearity. However, we hypothesised that discharge mRS score could more readily explain a patient's AHLOS while also serving as a proxy for discharge destination. In addition, socioeconomic status which has also been shown to relate to AHLOS in patients with stroke,[18] and differences in palliative care policies were not known. This means that any remaining difference in AHLOS between hospitals may be due to hospital-level factors as well as other unmeasured confounders. We also did not collect data on patient ethnicity although this has previously been associated with AHLOS.[49–51] While we cannot provide exact ethnic mix, the region where the study was conducted serves mainly a white British Caucasian population, with other races making up a very small minority.[52]

A further limitation of this study is that the hospital characteristics were self-reported by clinical leads or service managers at each hospital. This may have introduced information bias, especially with regard to the reported fte staffing levels, and the number of patients treated within or outside the stroke unit.

Furthermore, as this study covers eight NHS hospitals in the East of England that span both urban and rural regions, and as NHS policies are fairly standard, we believe these sites are generally representative of others across the UK. However, as we lacked an adequate number of hospitals to run a multilevel model with hospital as a random effect, our findings cannot be generalised to other healthcare settings outside the UK with differing national policies.

In summary, the heterogeneities that exist in stroke care at the regional UK level have the ability to lead to differences in stroke patient outcomes such as AHLOS. This provides a powerful message for patients, clinicians, service providers and policymakers—that there are modifiable hospital factors that may determine better outcomes in stroke. For example, a hub and spoke model of care could be advocated to increase efficiencies while also providing for more beneficial stroke health outcomes. Countries that are in the process of developing their healthcare systems can use these findings to inform their decision making in delivering optimal care.

**Author affiliations**
¹Ageing Clinical and Experimental Research Group, College of Life Sciences and Medicine, University of Aberdeen, Aberdeen, UK
²Medical Statistics Team, College of Life Sciences and Medicine, University of Aberdeen, Aberdeen, UK
³Norwich Medical School, Univeristy of East Anglia, Norwich, UK
⁴Addenbrooke's Hospital, University of Cambridge, Cambridge, UK
⁵Stroke Research Group, Norfolk and Norwich University Hospital, Norwich, UK

**Acknowledgements** We thank the stroke database team and stroke research team staff from all participating sites who contributed to data collection. We also would like to acknowledge the contribution of Anglia Stroke Clinical Network Evaluation Study (ASCNES) Group.

**Collaborators** Max O Bachmann (UEA), Garry R Barton (UEA), Fiona Cummings (ASHCN), Diana J Day (ASHCN/AH), Abraham George (JPUH), Rachel Hale (UEA), Anthony Kneale Metcalf (NNUH/UEA), Stanley D Musgrave (UEA), Phyo Kyaw Myint (UoA/UEA), Joseph Ngeh (IPH), Anne Nicholson (WSH), Peter Owusu-Agyei (PH), John F Potter (UEA/NNUH), Gill M Price (UEA).

**Contributors** MT: cleaned the data, undertook the statistical analysis and interpretation of results, reviewed the literature and prepared the manuscript. DM, PKM: supervised the study. DM: provided statistical analysis advice and assistance. MOB, EAW, JFP, PKM: conceptualised and designed the use of ASCNES and obtained funding. SDM: was ASCNES study coordinator and managed the data. All authors reviewed the manuscript and contributed in critical revision and final manuscript preparation.

**Funding** This work was supported by the National Institute for Health Research (NIHR) Research for Patient Benefit Programme (PB-PG-1208-18240). EAW receives funding support from the NIHR Biomedical Research Centre award to Cambridge. MT holds a PhD studentship funded by the College of Life Sciences and Medicine, University Aberdeen (CF10109-38).

**Disclaimer** The views expressed are those of the authors and not necessarily those of the NHS, the NIHR or the Department of Health.

**Competing interests** None declared.

**Patient consent for publication** Not required.

**Ethics approval** Ethical approval was obtained from the NRES Committee East of England—Norfolk (REC reference number 10/H0310/44).

**Provenance and peer review** Not commissioned; externally peer reviewed.

**Data sharing statement** The datasets generated and analysed during the current study are available from the ASCNES team on reasonable request.

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
