## [Reviewer comments · BMJ Open]

BMJ Open

BMJ Open is committed to open peer review. As part of this commitment we make the peer review history of every article we publish publicly available.

When an article is published we post the peer reviewers' comments and the authors' responses online. We also post the versions of the paper that were used during peer review. These are the versions that the peer review comments apply to.

The versions of the paper that follow are the versions that were submitted during the peer review process. They are not the versions of record or the final published versions. They should not be cited or distributed as the published version of this manuscript.

BMJ Open is an open access journal and the full, final, typeset and author-corrected version of record of the manuscript is available on our site with no access controls, subscription charges or pay-per-view fees (<http://bmjopen.bmj.com>).

If you have any questions on BMJ Open's open peer review process please email info.bmjopen@bmj.com

BMJ Open

Does service heterogeneity have an impact on acute hospital length of stay in stroke? A UK-based multi-centre prospective cohort study

Journal:	BMJ Open
Manuscript ID	bmjopen-2018-024506
Article Type:	Research
Date Submitted by the Author:	29-May-2018
Complete List of Authors:	Tørnes, Michelle; University of Aberdeen College of Life Sciences and Medicine, Ageing Clinical and Experimental Research Group McLernon, David; University of Aberdeen College of Life Sciences and Medicine, Medical Statistics Team Bachmann, Max; University of East Anglia, Norwich Medical School Musgrave, Stanley; University of East Anglia, Norwich Medical School Warburton, Elizabeth; University of Cambridge, Addenbrooke's Hospital Potter, John; University of East Anglia, Norwich Medical School; Norfolk and Norwich University Hospital, Stroke Research Group Myint, Phyo; University of Aberdeen College of Life Sciences and Medicine, Ageing Clinical and Experimental Research Group; Norfolk and Norwich University Hospital, Stroke Research Group
Keywords:	Acute hospital, Health Services Research, Length of Stay, Outcome, Stroke < NEUROLOGY

**Does service heterogeneity have an impact on acute hospital length of stay in stroke? A UK-based**
**multi-centre prospective cohort study**

Michelle Tørnes¹, David McLernon², Max O Bachmann³, Stanley D Musgrave³, Elizabeth A
Warburton⁴, John F Potter³, Phyo Kyaw Myint^{1,5}: On behalf of the Anglia Stroke Clinical
Network Evaluation Study (ASCNES) Group

¹Ageing Clinical and Experimental Research (ACER) Group, Institute of Applied Health
Sciences, School of Medicine, Medical Sciences and Nutrition, University of Aberdeen,
Scotland, UK; ²Medical Statistics Team, Institute of Applied Health Sciences, School of
Medicine, Medical Sciences and Nutrition, University of Aberdeen, Scotland, UK; ³Norwich
Medical School, University of East Anglia, Norwich, UK; ⁴Addenbrooke's Hospital,
Cambridge, UK; ⁵Stroke Research Group, Norfolk and Norwich University Hospital,
Norwich, UK.

**Correspondence to:**

Michelle Tørnes,

C/o: Professor P K Myint; Room 4.013, Polwarth Building | School of Medicine & Dentistry |
Division of Applied Health Sciences | Foresterhill, University of Aberdeen | Aberdeen, AB25
2ZD, UK | Tel: +44 (0) 1224 437843 | Fax: +44 (0) 1224 437911 | Mail to:

michelle.tornes@abdn.ac.uk

Word count: 3261

**Keywords:** Acute hospital, Health Services Research, Length of Stay, Outcome, Stroke

ABSTRACT

Objectives: To determine whether stroke patients' acute hospital length of stay (AHLOS) varies between hospitals, over and above cases mix differences, and to investigate the hospital-level factors driving such hospital variations in AHLOS.

Design: A multicentre prospective cohort study.

Setting: Eight National Health Service acute hospital trusts within the Anglia Stroke & Heart Clinical Network in the East of England, UK.

Participants: The study sample was systematically selected to include all consecutive patients admitted to any of the eight hospitals, diagnosed with stroke by an admitting clinician, every third month between October 2009 and September 2011.

Primary and secondary outcome measures: AHLOS was defined as the number of days between date of hospital admission and discharge or death, whichever came first. We used a multiple linear regression model to investigate the association between hospital (as a fixed-effect) and AHLOS, adjusting for a number of important patient covariates. Exploratory data analysis was utilized to gain insight into the hospital-level characteristics which may contribute to the hospital-level variance.

Results: A total of 2233 stroke admissions (52% female, median age (interquartile range (IQR)) 79 (70 to 86) years, 83% ischaemic stroke) were included in the study analysis. The overall median AHLOS (IQR) was 9 (4 to 21) days. After adjusting for patient covariates and confounding factors, AHLOS still differed significantly between hospitals ($p < 0.001$; $R^2 = 2.4\%$). Furthermore, hospitals with the longest adjusted AHLOS's were predominantly secondary and in which stroke volumes were lower.

Conclusions: We have clearly demonstrated that AHLOS varies at the hospital-level and, have highlighted the potential importance of hospital type and volume of stroke patients as the hospital factors influencing these differences.

For peer review only

ARTICLE SUMMARY

Strengths and limitations of this study

- This is a comprehensive study that has used multi-centre data to determine whether acute hospital length of stay of stroke patients varies across hospitals in the UK, after adjustment for patient-level covariates.
- With a wealth of detailed patient data we were able to adjust for the important covariates, inpatient complications and discharge destination, which previous studies have not addressed.
- Hospital-level effects were not estimated due to the limited hospital sample size of eight.
- Although National Institute for Health Stroke Scale (NIHSS) stroke patients' scores are known to be associated with acute hospital length of stay, we were unable to adjust for this as this was only calculated for patients who were potentially eligible for thrombolysis and would have introduced collection bias.

INTRODUCTION

Stroke is the second leading cause of mortality and the third leading cause of disability in the world, with a global incidence of 16.9 million in 2010.¹⁻² While acute hospitalization for stroke in the US has been estimated at a cost of \$31,667, total direct stroke-related annual medical costs are expected to triple, from \$71.6 billion in 2012 to \$184.1 billion by 2030.³⁻⁴ Considerable differences in stroke-related outcomes exist worldwide, with the highest age-standardized stroke-related mortality and disability adjusted life-years rates observed in Russia and Eastern European countries.¹ Stark regional disparities within countries are also apparent. In the UK, for example, there exists a clear north-south divide where the lowest stroke-related mortality rates are observed almost exclusively in the South of England.⁵ Such differences in outcomes likely reflect underlying stroke incidence rates and variations in exposure to relevant risk factors.⁵⁻⁶ However, we and others have demonstrated that some of the differences in post-stroke survival have also been explained by disparities in available resources and medical care.⁷⁻¹¹ Studies assessing the effect of stroke care heterogeneities have largely focused on mortality as the primary outcome.

However, it is possible that heterogeneities in stroke care also impact other important stroke-related outcomes, such as a patient's acute hospital length of stay (AHLOS). To date, studies have mainly identified patient-related determinants of AHLOS,¹²⁻¹⁵ with little exploration into hospital-level influences.

During acute hospitalization, AHLOS is the main driver of acute care costs.¹⁶ Determining the hospital-level factors influencing AHLOS therefore provides invaluable information to service providers and policymakers who can develop optimal management strategies and enhance patient care by minimizing service deficiencies, costs and bed shortages.

The aim of this study is to investigate whether there are variations in stroke patients' AHLOS which can be partly explained by heterogeneities in characteristics of stroke care between

hospitals in a UK National Health Service (NHS) setting. We also aimed to explore which
hospital-level factors drive such hospital variations in AHLOS.

For peer review only

METHODS

Study design

[revised manuscript text omitted]

Table 1 Sample characteristics of patients included in analysis (n=2333) and missing data

Patient Characteristic	Median (IQR) or No. (%)	Missing Data (%)
Age, y	79 (70 to 86)	2 (0.1)
Sex, female	1165 (52)	2 (0.1)
Recurrent Stroke*	448 (20)	30 (1)
Diabetes Mellitus*	370 (17)	30 (1)
Dementia*	207 (9)	30 (1)
Hypercholesterolemia*	355 (16)	30 (1)
Hypertensive*	1483 (66)	30 (1)
Myocardial Infarction or Ischaemic Heart Disease*	517 (23)	30 (1)
TIA*	340 (15)	30 (1)
Previous Cancer*	195 (9)	30 (1)
Active Cancer*	137 (6)	30 (1)
Depression*	137 (6)	30 (1)
Rheumatoid Arthritis*	154 (7)	30 (1)
COPD*	116 (5)	30 (1)
Pre-stroke Rankin Score		442 (20)
0	914 (41)	
1	335 (15)	
2	191 (9)	
3	184 (8)	
4 & 5	167 (7)	
Pre-Stroke Residence		51 (2)
Independent living with formal care	210 (9)	
Independent living without formal care	1752 (78)	
Institution	220 (10)	
Stroke Type		96 (4)
Ischaemic	1864 (83)	
Haemorrhagic	273 (12)	
Oxford Community Stroke Project Classification		260 (12)
LACS†	503 (23)	
PACS†	784 (35)	
POCS†	279 (12)	
TACS†	407 (18)	
No Brain Lateralisation	244 (12)	167 (8)
Inpatient Complication*	655 (29)	0 (0)
Discharge Destination		50 (2)
Death	414 (19)	
Independent living with formal care	224 (10)	
Independent living without formal care	1006 (45)	
Institution	252 (11)	
Interim/Rehab Setting	287 (13)	
Winter Admission	1159 (52)	0 (0)
Weekend Admission	614 (27)	0 (0)

*No information was assumed to indicate absence of condition or complication

† LACS= Lacunar Anterior Circulation Stroke; PACS= Partial Anterior Circulation Stroke; POCS= Posterior Circulation Stroke; TACS = Total Anterior Circulation Stroke

Hospital service characteristics

Service characteristics of each hospital are outlined in Table 2, with median AHLOS. After standardization there was still extensive heterogeneity in staffing levels, bed capacity and the provision of services and facilities. The overall median AHLOS (IQR) was 9 (4 to 21) days and there appeared to be crude variations in this outcome between hospitals.

For peer review only

Table 2 Hospital characteristics per individual hospital

Hospital Characteristics	1	2	3	4	5	6	7	8
General Characteristics								
Catchment Population	400,000	160,000	350,000	230,000	680,000	300,000	240,000	275,000
Hospital Type	Tertiary	Secondary	Secondary	Secondary	Tertiary	Secondary	Secondary	Secondary
Hospital Stroke Volume (No. of ASCNES admissions per month)	52	13	46	19	88	57	35	31
Facilities and Services								
No. of hospital beds	1000	304	800	500	1237	611	488	460
No. of stroke unit beds (per 100 admissions)	71	77	54	138	41	55	83	65
No. of hospital beds per CT scanners	500	304	400	250	518	306	244	230
Distance to Vascular Surgery (miles)	0	18	0	25	0	0	43	30
Distance to Neurosurgery (miles)	0	18	58	89	61	38	48	30
Rehabilitation Provision	Onsite	Onsite	Offsite	Offsite	Offsite	Onsite	Offsite	Onsite
Early Supported Discharge Provision	No	Yes	No	Yes	Yes	Yes	No	No
Stroke Unit Staffing Levels*								
Senior doctors †	0.34	0.25	0.49	0.47	0.42	0.31	0.62	0.87
Junior doctors †	0.55	0.65	0.72	0.59	0.56	0.64	0.12	0.25
Health care associates and nurses (band 5-7)	9.2	8	6	7.4	7	5.3	6.5	10
Physiotherapists (band 2-8)	0.55	1	0.79	0.4	0.91	0.78	0.69	1
Occupational Therapists (band 3-8)	0.49	0.5	1.4	0.59	0.6	0.58	0.52	1.1
Speech and Language Therapists	0.39	0.15	0.2	0.18	0.35	0.03	0.26	0.1
No. of non-stroke patients treated daily on stroke unit (per five stroke unit beds)	0.27	0	0.10	0.47	0.05	0.31	0.17	0
No. of stroke patients treated daily outside stroke unit (per five stroke unit beds)	0.14	5	0	0.30	0.01	0.41	0	0
Median AHLOS (IQR)	8 (4 to 20)	29 (24 to 42)	11 (5 to 27)	14 (4 to 30)	8 (4 to 14)	10 (5 to 22)	11 (6 to 23)	7 (3 to 20)

*Number of fte staff per five stroke unit beds (weighted average for the four study periods taken)

† Weekday numbers only

Univariable linear regression

In univariable linear regression (Table S4 in the online supplementary document 2), patients who were older, female, had previous cancer, a previous stroke, had diabetes mellitus, had dementia or were a winter admission had a significantly longer AHLOS ($p < 0.05$). Patients who had a haemorrhagic stroke, hypercholesterolemia, pre-stroke Rankin score of 0, no signs of brain lateralisation, lacunar stroke and who lived independently at home without formal care (compared to those who had formal care) prior to stroke were all shown to be significantly associated with a shorter AHLOS ($p < 0.01$).

The strongest associations with AHLOS were seen for inpatients who developed a complication, who were admitted to hospital 2 and who were institutionalized after discharge. Inpatient complications were associated with twice as long an AHLOS compared to those without a complication. Patients admitted to hospital 2 had triple the AHLOS of those admitted to hospital 1. Institutionalization quadrupled a patients' AHLOS compared to those who were discharged to home without formal care.

Finally, compared to being admitted to hospital 1 of our study, admission to hospitals 3, 4 and 7 were also significantly associated with an increasing AHLOS, whereas hospital 5 was associated with a decreasing AHLOS ($p < 0.05$; $R^2 = 2.4$).

Multiple linear regression

Multiple linear regression results for AHLOS are summarized in Table 3 and shows that 40% of the variation in AHLOS has been explained. Sex, recurrent stroke and dementia mellitus were no longer statistically associated ($p < 0.05$) with AHLOS in multiple regression. No variables included from the univariable analysis with $p > 0.05$ became statistically significant in the multivariable analysis. Developing an inpatient complication and being institutionalized were still strongly positively related to AHLOS. After adjusting for patient

covariates and confounding factors, AHLOS was still shown to significantly differ between
hospitals, with the shortest and longest AHLOS observed for hospitals 5 and 2, respectively.

There were no obvious differences between the results using complete cases only (Tables S5-
6 in the online supplementary document 2) and multiple imputation.

For peer review only

Table 3 Multiple regression analysis for AHLOS (n=2233; R²=40%)

Patient Characteristic	e ^β *	95% CI*	P
Age, y	1.01	1.00 to 1.01	<0.001
Sex, female	0.99	0.92 to 1.07	0.80
Recurrent Stroke	1.00	0.91 to 1.10	1.00
Diabetes Mellitus	1.08	0.98 to 1.19	0.12
Dementia	1.20	1.05 to 1.38	0.01
Hypercholesterolemia	0.94	0.85 to 1.04	0.25
Myocardial Infarction or Ischaemic Heart	1.01	0.93 to 1.10	0.83
Previous Cancer	1.17	1.03 to 1.33	0.01
COPD	0.91	0.77 to 1.07	0.23
Pre-stroke Rankin Score (reference 0)			<0.001
1	1.15	1.03 to 1.28	0.02
2	1.15	1.00 to 1.33	0.05
3	1.33	1.13 to 1.56	<0.001
4 & 5	1.15	0.96 to 1.38	0.12
Pre-Stroke Residence (reference Independent living without formal care)			<0.001
Independent living with formal care	0.86	0.75 to 0.99	0.04
Institution	0.52	0.44 to 0.62	<0.001
Haemorrhagic Stroke	0.84	0.75 to 0.95	<0.001
Oxford Community Stroke Project Classification (reference LACS)			<0.001
PACS	1.34	1.22 to 1.48	<0.001
POCS	1.44	1.26 to 1.63	<0.001
TACS	1.49	1.31 to 1.70	<0.001
No Brain Lateralisation	0.82	0.73 to 0.93	<0.001
Inpatient Complication	1.72	1.58 to 1.87	<0.001
Discharge Destination (reference Independent living without formal care)			<0.001
Independent living with formal care	1.99	1.74 to 2.27	<0.001
Institution	3.58	3.09 to 4.15	<0.001
Interim/Rehab Setting	2.18	1.94 to 2.46	<0.001
Death	0.85	0.74 to 0.97	0.02
Winter Admission	1.15	1.07 to 1.24	<0.001
Weekend Admission	1.04	0.96 to 1.13	0.30
Hospital (reference 1)			<0.001
2.76	1.80 to 4.22	<0.001
1.24	1.09 to 1.42	<0.001
1.36	1.15 to 1.61	<0.001
0.85	0.75 to 0.95	0.01
1.06	0.92 to 1.22	0.42
1.19	1.03 to 1.37	0.02
0.99	0.85 to 1.14	0.84

*β estimates and 95% confidence intervals were calculated for predicted log AHLOS. Prior to reporting they were transformed back to AHLOS through exponentiation and represent geometric mean AHLOS

Graphical exploratory analysis

Mean baseline AHLOS of each hospital (estimated from the multiple regression model) was plotted against hospital stroke volume and clustered by hospital type in Figure 2. It appears that hospitals (of either type) that have that have higher stroke volumes have a shorter AHLOS than those with lower stroke volumes. In addition, it also appears that secondary hospitals have longer AHLOS in general than tertiary hospitals when all patient covariates are taken into account.

No discernible patterns were seen for mean baseline hospital AHLOS and staffing levels, surgery facilities, number of stroke patients treated outside the stroke unit, number of non-stroke patients treated on the stroke unit, and bed numbers (Figures S1-13 in the online supplementary document 2).

DISCUSSION

This multi-centre cohort study has demonstrated that substantial heterogeneities exist in stroke hospital service and staff provision across three counties in the East of England. After adjusting for patient characteristics and confounding factors, we have shown that AHLOS significantly differed between hospitals. This suggests that the heterogeneities we see in stroke care between hospitals are having an effect on AHLOS of these patients. It also appears from our exploratory analysis that the volume of stroke patients admitted to hospital and the type of hospital may play a role in partially explaining these hospital-level AHLOS differences.

A number of other studies have demonstrated hospital-level variation in AHLOS due to hospital factors, such as stroke unit volume and hospital size.^{12, 23} However, these studies did not account for important covariates such as inpatient complications and discharge destination. Indeed, our analysis has shown how strongly these two factors are associated with a patient's AHLOS, and so by adjusting for them in our model, our study has been able to establish that any remaining differences in this outcome between hospitals is due to hospital-level factors. We have therefore shown that, in addition to stroke-related mortality,⁷⁻¹¹ other important patient outcomes are determined by hospital heterogeneities. This finding is particularly relevant given its strong correlation with inpatient costs and the variation in AHLOS seen both nationally and regionally.^{16, 24}

Our exploration of hospital characteristics indicates that being admitted to a hospital that has a higher stroke volume compared to one that has a lower stroke volume may be responsible for shortening AHLOS. This has also been demonstrated for high-volume stroke units in a previous study.²³ This latter study, however, did not adjust for inpatient complications and discharge destination. However, in other studies that have adequately adjusted for these

patient characteristics, higher stroke volumes have also been shown to be significantly
associated with lowered risk of mortality which implies that this hospital-level characteristic
has an impact on outcomes in stroke.^{11, 25-26}

In addition to hospital stroke volume, hospital type is another hospital-level characteristic that
appears to play a role in influencing AHLOS. We have shown that tertiary hospitals (also
referred to as academic hospitals) generally have a lower AHLOS compared to secondary
hospitals. This finding is in agreement with a previous multicentre study in Argentina which
demonstrated that unadjusted median AHLOS was shorter for academic hospitals.²⁷

It may be that the quality of care and accessibility to resources and sub-specialists is better in
these higher volume or academic hospitals, and this is leading to more favourable
outcomes.²⁸ For example, it has been shown that stroke patients admitted to high-volume
stroke units have significantly greater odds of being treated and assessed earlier than those
admitted to lower-volume units.²³ This apparent increase in efficiency likely results from the
greater pressure on beds these higher-volume units experience which requires them to have a
faster throughput of patients. Furthermore, it has been shown that admission rates to stroke
units are significantly higher in academic hospitals, and pneumonia rates are lower.²⁷ The
reason for this is not yet clear but is likely to play a role in determining AHLOS.

Hospital 8, however appears to contradict the above findings in that although it has one of the
lowest AHLOS, it is a secondary hospital and it also has one of the smallest volume of stroke
patients in the study. Such a discrepancy is likely to be a reflection of the small number of
hospitals assessed, as there are likely to be a number of competing factors playing a role in
determining hospital-level AHLOS variance. Although hospital 8 has one of the lowest stroke
volumes and is a secondary hospital, it has the highest number of fte senior doctors, health
care associates and nurses, and physiotherapists per five beds, and the lowest number of

hospital beds per CT scanners out of all the hospitals studied. Staffing levels are likely to be
an important determinant of AHLOS, given that higher nurse: bed ratios have been shown to
be important in reducing other stroke-related outcomes, such as mortality.⁸⁻⁹

The main strength of our study is its prospective design and the detailed patient-level data we
obtained. This allowed us to gain a better understanding of the extent to which the variation
in AHLOS exists over and above patient characteristics. A gold standard randomized
controlled trial would be unethical and ineffective at exploring these hospital-level variations,
thus our observational study design is the best approach to answer these important questions.
The robust statistical analysis has allowed easy and quick visualization of notable patterns in
the dataset and provides a candid assessment of the research objectives by considering the
limits of inference due to the small number of hospitals. Multiple imputation has also reduced
potential bias that may have otherwise been introduced from complete case analysis alone.

The major limitation of this study was the small number of hospitals that has restricted the
conclusions we can make from our exploratory analysis of hospital characteristics. A number
of competing factors may be playing a role in determining AHLOS, but due to this small
sample size and large heterogeneities between the hospitals and their stroke units, we are
unable to disentangle any definitive relationships. Furthermore, although National Institute
for Health Stroke Scale (NIHSS) and a patient's mRS at discharge has been shown to be
associated with stroke patients' AHLOS,^{14, 16, 18, 29} they were excluded as covariates from the
main analysis. As NIHSS scores were only calculated for those who were potentially eligible
for thrombolysis at the time of our study, the incompleteness was not missing at random and
would have introduced collection bias into our results. As discharge mRS and discharge
destination both included a categorical factor representing inpatient death only one of these
variables could be included into the analysis due to issues of multi-collinearity. However, we
hypothesized that discharge destination could more readily explain a patient's AHLOS due to

waiting times associated with institutionalization. Furthermore, as this study only covers
eight NHS hospitals in the East of England, the findings may not be generalisable to
healthcare settings outside the UK.

Although some studies have shown a link between a number of hospital characteristics and
AHLOS, no study has yet addressed the issue of clustering. A study with an adequate number
of hospitals, robust statistical techniques (such as multi-level modelling) and high-quality
data is therefore required in order to identify the types of services and staffing levels required
for lowering AHLOS.

In summary, the heterogeneities that exist in stroke care at the regional UK level have the
ability to lead to differences in stroke-patient outcomes such as, AHLOS. This provides a
powerful message for patients, clinicians, service providers and policymakers – that there are
modifiable hospital factors that can determine better outcomes in stroke. For example, low
volume hospitals could consider reducing their stroke bed numbers as a means to increase
their efficiency. Countries that are in the process of developing their healthcare systems can
use these findings to inform their decision making in delivering optimal care.

**Acknowledgements:** We thank the stroke database team and stroke research team staff from
all participating sites who contributed to data collection. We also would like to acknowledge
the contribution of Anglia Stroke Clinical Network Evaluation Study (ASCNES) Group.

**Author Contributions:** M.T cleaned the data, undertook the statistical analysis and
interpretation of results, reviewed the literature and, prepared the manuscript. D.M and
P.K.M supervised the study. D.M provided statistical analysis advice and assistance. M.O.B,
E.A.W, J.F.P and P.K.M conceptualized and designed the use of ASCNES and obtained
funding. S.D.M was ASCNES study coordinator and managed the data. All authors reviewed
the manuscript and contributed in critical revision and final manuscript preparation.

**Funding:** This work was supported by the National Institute for Health Research (NIHR)
Research for Patient Benefit Programme (PB-PG-1208-18240). EAW receives funding
support from the NIHR Biomedical Research Centre award to Cambridge. MT holds a PhD
studentship funded by the College of Life Sciences & Medicine, University Aberdeen
(CF10109-38).

**Disclaimer:** The views expressed are those of the authors and not necessarily those of the
NHS, the NIHR or the Department of Health.

**Competing interests:** The authors declare no competing interests.

**Patient consent:** Not required.

**Ethics approval:** Ethical approval was obtained from the NRES Committee East of England
– Norfolk (REC Reference number 10/H0310/44).

**Data sharing statement:** The datasets generated and analysed during the current study are
available from the ASCNES team on reasonable request.

REFERENCES

1. Feigin VL, Norrving B, Mensag GA. Global Burden of Stroke. *Circ Res* 2017;120:439-448. doi: 10.1161/CIRCRESAHA.116.308413.
2. Krishnamurthi RV, Feigin VL, Forouzanfar MH, et al on behalf of the Global Burden of Diseases, Injuries, and Risk Factors Study 2010 (GBD 2010) and the GBD Stroke Experts Group. Global and Regional burden of first-ever ischaemic and haemorrhagic stroke during 1990-2010: findings from the Global Burden of Disease Study 2010. *Lancet Glob Health* 2013;1:e259-e281. doi: [http://dx.doi.org/10.1016/S0140-6736\(13\)61953-4](http://dx.doi.org/10.1016/S0140-6736(13)61953-4).
3. Mu F, Hurley D, Betts KA, et al. Real-world costs of ischemic stroke by discharge status. *Curr Med Res Opin* 2016;33:371-378. doi: <https://doi.org/10.1080/03007995.2016.1257979>.
4. Ovbiagele B, Goldstein LB, Higashida RT, et al on behalf of the American Heart Association Advocacy Coordinating Committee and Stroke Council. Forecasting the Future of Stroke in the United States: A Policy Statement From the American Heart Association and American Stroke Association. *Stroke* 2010;44:2361-2375. doi: 10.1161/STR.0b013e31829734f2.
5. Townsend N, Bhatnagar P, Wilkins E, et al. Cardiovascular disease statistics 2015. London: British Heart Foundation, 2015.
6. Marshall IJ, Wang Y, Crichton S, et al. The effects of socioeconomic status on stroke risk and outcomes. *Lancet Neurol* 2015;14:1206-1218. doi: 10.1016/S1474-4422(15)00200-8.
7. Bray BD, Ayis S, Campbell J, et al. Associations between the organisation of stroke services, process of care, and mortality in England: prospective cohort study. *BMJ* 2013;346. doi: 10.1136/bmj.f2827.

8. Bray BD, Ayis S, Campbell J, et al. Associations between stroke mortality and weekend
working by stroke specialist physicians and registered nurses: prospective multicentre cohort
study. *PLoS Med* 2014;11:e1001705. doi: 10.1371/journal.pmed.1001705.
9. Myint PK, Bachmann MO, Loke YK, et al. Important factors in predicting mortality
outcome from stroke: findings from the Anglia Stroke Clinical Network Evaluation
Study. *Age Ageing* 2017;46:83-90. doi: 10.1093/ageing/afw175.
10. Rudd AG, Jenkinson D, Grant RL, et al. Staffing levels and patient dependence in
English stroke units. *Clin Med (Lond)* 2009;9:110-5. doi: 10.7861/clinmedicine.9-2-110.
11. Fonarow GC, Smith EE, Reeves MJ, et al for the Get With The Guidelines Steering
Committee and Hospitals. Hospital-Level Variation in Mortality and Rehospitalization for
Medicare Beneficiaries with Acute Ischemic Stroke. *Stroke* 2011;42:159-166. doi:
10.1161/STROKEAHA.110.601831.
12. Kim SM, Hwang SW, Oh E, et al. Determinants of the Length of Stay in Stroke Patients.
*Osong Public Health and Research Perspectives* 2013;4:329-341. doi:
http://dx.doi.org/10.1016/j.phrp.2013.10.008.
13. Kwok CS, Clark A, Ford G, et al. Association Between Prestroke Disability and Inpatient
Mortality and Length of Acute Stay After Acute Stroke. *J Am Geriatr Soc* 2012;60:726-732.
doi: 10.1111/j.1532-5415.2011.03889.x.
14. Appelros P. Prediction of length of stay for stroke patients. *Acta Neurol Scand*
2007;116:15-19. doi: 10.1111/j.1600-0404.2006.00756.x.

15. Arboix A, Massons J, Garcia-Eroles L, et al. Clinical Predictors of Prolonged Hospital
Stay after Acute Stroke: Relevance of Medical Complications. *International Journal of*
*Clinical Medicine* 2012;3:502-507. doi: <http://dx.doi.org/10.4236/ijcm.2012.36090>.
16. Jorgensen HS, Nakayama H, Raaschou HO, et al. Acute stroke care and rehabilitation: an
analysis of the direct cost and its clinical and social determinants. The Copenhagen Stroke
Study. *Stroke* 1997;28:1138-1141. doi: <https://doi.org/10.1161/01.STR.28.6.1138>.
17. Myint PK, Potter JF, Price GM, et al. Evaluation of stroke service in Anglia stroke
clinical network to examine the variation in acute services and stroke outcomes. 2011. *BMC*
*Health Serv Res* 2011;11:50. doi: 10.1186/1472-6963-11-50.
18. Kwok CS, Musgrave SD, Price GM, et al, and On behalf of the Anglia Stroke Clinical
Network Evaluation Study (ASCNES) Group. Similarity of patient characteristics and
outcomes in consecutive data collection on stroke admissions over one month compared to
longer periods. *BMC Res Notes* 2014;7:342. doi: 10.1186/1756-0500-7-342.
19. Bryan ML, Jenkins SP. Multilevel Modelling of Country Effects: A Cautionary Tale.
*European Sociological Review* 2016;32:3-22. doi: 10.1093/esr/jcv059.
20. Bowers J, Drake KW. EDA for HLM: Visualization when Probabilistic Inference Fails.
*Political Analysis* 2005;13:301-326. doi: 10.1093/pan/mpi031.
21. van Burren S, Groothuis-Oudshoorn K. MICE: Multivariate Imputation by Chained
Equations in R. *Journal of Statistical Software* 2011;45:1-67. doi: 10.18637/jss.v045.i03.
22. R Development Core Team (2017). R: A language and environment for statistical
computing. R Foundation for Statistical Computing, Vienna, Austria. ISBN 3-900051-07-0,
URL <http://www.R-project.org>.

23. Svendsen ML, Ehlers LH, Ingeman A, et al. Higher Stroke Unit Volume Associated With
Improved Quality of Early Stroke Care and Reduced Length of Stay. *Stroke* 2012;43:3041-
3045. doi: 10.1161/STROKEAHA.111.645184.
24. Peltola M, Seppälä TT, Malmivaara A, et al on behalf of the Eurohopes Study Group.
Individual and Regional-level Factors Contributing to Variation in Length of Stay after
Cerebral Infarction in Six European Countries. *Health Econ* 2015;24:38-52. doi:
10.1002/hec.3264.
25. Matsui H, Fushimi K, Yasunaga H. Variation in Risk-Standardized Mortality of Stroke
among Hospitals in Japan. *PLoS One* 2015;10:e0139216. doi:
:10.1371/journal.pone.0139216.
26. Tsugawa Y, Kumamaru H, Yasunaga H, et al. The Association of Hospital Volume With
Mortality and Costs of Care for Stroke in Japan. *Med Care* 2013;51:782-788. doi:
10.1097/MLR.0b013e31829c8b70.
27. Sposato LA, Esnaola MM, Zamora R, et al, on behalf of ReNACer Investigators and the
Argentinian Neurological Society. Quality of Ischemic Stroke Care in Emerging Countries:
The Argentinian National Stroke Registry (ReNACer). *Stroke* 2008; 39:3036-3041. doi: DOI:
10.1161/STROKEAHA.108.521062.
28. Kimball MM, Neal D, Waters MF, et al. Race and Income Disparity in Ischemic Stroke
Care: Nationwide Inpatient Sample Database 2002 to 2008. *J Stroke Cerebrovasc Dis*
2014;23:17-24. doi: <https://doi.org/10.1016/j.jstrokecerebrovasdis.2012.06.004>.
29. Chang K, Tseng M, Weng H, et al. Prediction of Length of Stay of First-Ever Ischemic
Stroke. *Stroke* 2002;33:2670-2674. doi: 10.1161/01.STR.0000034396.68980.39.

**Figure 1** Flow chart of patient participation inclusion and exclusion for study analysis

**Figure 2** Model estimates of mean baseline AHLOS per hospital against hospital stroke
volume and clustered by hospital type with 95% confidence intervals. Multiple regression
model was adjusted for patient covariates.

For peer review only

Figure 1 Flow chart of patient participation inclusion and exclusion for study analysis

Figure 2 Model estimates of mean baseline AHLOS per hospital against hospital stroke volume and clustered by hospital type with 95% confidence intervals. Multiple regression model was adjusted for patient covariates.

152x152mm (300 x 300 DPI)

STUDY PROTOCOL

Open Access

Evaluation of stroke services in Anglia stroke clinical network to examine the variation in acute services and stroke outcomes

Phyo K Myint^{1,2*}, John F Potter^{1,2}, Gill M Price¹, Garry R Barton¹, Anthony K Metcalf², Rachel Hale², Genevieve Dalton³, Stanley D Musgrave¹, Abraham George⁴, Raj Shekhar⁵, Peter Owusu-Agyei⁶, Kevin Walsh⁷, Joseph Ngeh⁸, Anne Nicholson⁹, Diana J Day¹⁰, Elizabeth A Warburton¹⁰, Max O Bachmann¹

Abstract

Background: Stroke is the third leading cause of death in developed countries and the leading cause of long-term disability worldwide. A series of national stroke audits in the UK highlighted the differences in stroke care between hospitals. The study aims to describe variation in outcomes following stroke and to identify the characteristics of services that are associated with better outcomes, after accounting for case mix differences and individual prognostic factors.

Methods/Design: We will conduct a cohort study in eight acute NHS trusts within East of England, with at least one year of follow-up after stroke. The study population will be a systematically selected representative sample of patients admitted with stroke during the study period, recruited within each hospital. We will collect individual patient data on prognostic characteristics, health care received, outcomes and costs of care and we will also record relevant characteristics of each provider organisation. The determinants of one year outcome including patient reported outcome will be assessed statistically with proportional hazards regression models. Self (or proxy) completed EuroQoL (EQ-5D) questionnaires will measure quality of life at baseline and follow-up for cost utility analyses.

Discussion: This study will provide observational data about health service factors associated with variations in patient outcomes and health care costs following hospital admission for acute stroke. This will form the basis for future RCTs by identifying promising health service interventions, assessing the feasibility of recruiting and following up trial patients, and provide evidence about frequency and variances in outcomes, and intra-cluster correlation of outcomes, for sample size calculations. The results will inform clinicians, public, service providers, commissioners and policy makers to drive further improvement in health services which will bring direct benefit to the patients.

Background

Stroke is the third leading cause of mortality and the number one cause of long-term disability in the UK. More than 150,000 people suffer a stroke in the UK each year [1]. It costs the NHS approximately £ 7 billion per annum [2]. Stroke incidence rises sharply with age and despite better primary and secondary preventative measures, the total number of strokes is set to rise in

the UK [3]. Nevertheless, stroke care in UK is far from ideal: patients having a worse outcome in terms of death and dependency than many other European countries [4-6], at least in part due to differences in care provided [7]. There is also variation in outcome between different localities within the UK [8-11], these local differences being highlighted in the most recent publication of the National Sentinel Stroke Audit in 2009 [12]. These differences probably arise as a result of substantial variations in how the stroke services are provided across the UK. Examples of such differences are access to neurovascular/neurosurgical service, early supported

* Correspondence: phyo.k.myint@uea.ac.uk

¹Norwich Medical School, Faculty of Medicine & Health Sciences, Norwich, UK

Full list of author information is available at the end of the article

discharge, and stroke specialist on call rota for thrombolysis. The presence or absence of variations in stroke outcomes as a result of variation in care and how much the observed variations in patients' outcomes including patient reported outcome measure (PROM) are determined by the differences in service delivery have not been examined previously.

We hypothesise that variation in patient outcomes including mortality, length of stay, institutionalisation rate, and patient reported outcomes between care providers can partly be explained by the different ways in which stroke services are delivered. The main objectives of the study are (1) to describe variation in outcomes following stroke and to identify the characteristics of services that are associated with better outcomes after accounting for case mix differences and individual prognostic factors, and (2) to obtain preliminary data to identify sample size and inform future pragmatic real world setting RCTs in the area of health service delivery in stroke.

25 **Methods/Design**

A prospective cohort study will be conducted to identify characteristics of services that are associated with the best outcomes including patient reported outcomes, taking into account case-mix and patients' prognostic features. The study will consist of two components (1) consecutive stroke admissions in selected months (a total of 8 months) and (2) a prospective study of patient reported outcome in some of these selected months.

**Sample Population**

For the first component, the sample population will be stroke patients who are admitted to any of the hospitals within the Anglia region of Stroke & Heart Clinical Network between October 2009 and September 2011. Baseline data are already recorded, prior to the study commencement, as part of routine clinical data collection by Anglia Stroke Clinical Network (as described in detail below). The study sample will be a systematically selected sample (every third month) rather than a consecutive cohort of patients admitted to eight acute NHS hospital trusts. Therefore, this is not a consecutive case study; instead it seeks to be representative of the catchment population of the hospital and has taken into account the seasonal variation in stroke incidence and outcome [13].

For the patient reported outcome component of the study the following inclusion and exclusion criteria will be used. Inclusion criteria are (1) age \geq 18 years, (2) admitted to hospital with stroke (diagnosed by stroke physicians) during the study months, (3) able to provide informed consent or patient's personal consultee agrees to study participation. Exclusion criteria include (1) age

<18 years, (2) patients with pre-existing diagnosis of dementia (for PROM component only).

The Anglia Stroke Network was funded through the NHS Improvement Programme, following the publication of the National Stroke Strategy in December 2007. The Network was established in April 2008 to support the development of stroke services in Norfolk, Suffolk and Cambridgeshire regions. Since its inception, the Network regularly collected data to capture clinical service activities of the eight acute hospital trusts in the Network for the purpose of monitoring of services benchmarked by National targets and guidance from National Institute of Health & Clinical Excellence (NICE) in England and Wales. Data collection commenced in January 2009 and involves the individual trusts collecting clinical data which is fed back to the network by monthly reports. The total number of strokes admitted to the 8 acute trusts within the Network is approximately 4,000 per annum in 2009. The stroke cases were identified prospectively data were collected by the clinical team who looked after the patients and anonymised raw clinical data were sent to the network on monthly basis. The network collates and analyses the data for above mentioned purposes.

**Sample size**

Since this is an exploratory study designed to provide information for further analytic research, sample size will be determined partly pragmatically rather than on particular hypothesis tests. For illustration purposes, a total sample of 2264 patients would provide 80% power to detect a constant Hazard ratio (HR) of 0.76 for one-year mortality between two groups of roughly equal size, based on the log-rank test. This assumes a 20% one-year mortality rate in the reference group, no loss to follow-up before one year and 2-sided type I error of 5%. If one-year mortality is 30%, then 2264 patients would provide 76% power to detect a HR of 0.81.

Plan of investigation

The study will have a cohort design. We will follow up a cohort of patients systematically selected from each trust. For pragmatic purposes we will sample all patients who are admitted every third month, starting from October 2009. Over one calendar month, there will be ~ 300-350 stroke cases entered into the Network Clinical Data. Between October 2009 and September 2011, the Clinical Network would have collected a total of eight 3-monthly datasets per trust (i.e. 8 study months in total: Oct 2009, Jan 2010, April 2010, July 2010, October 2010, Jan 2011, April 2011, July 2011). Therefore, the estimated total cohort size with baseline clinical data will be ~ 2,400 stroke cases

during this exercise (30% of 4000 patients admitted annually in 8 trusts = 1200 × 2 yrs).

We will collect patient data by hospital trusts and conduct a questionnaire survey of patients' outcomes. Due to the nature of the study we would need 100% follow-up in randomly selected populations. Because we will be using a partially historical cohort, to avoid selection bias for mortality outcome, informed consent from all eligible participants will not be feasible. Therefore, it is most appropriate for the clinical team to collect the outcome data to comply with current ethical guidance in the UK. Therefore, the identifiable patient data will only be held at the local NHS trusts.

Neither the network nor the investigators will have access to any identifiable patient information (e.g. name, address). For outcome data we will utilise death certificate and hospital episode data from the Patient Administrative System (PAS) as described previously [14,15]. This approach will be used in conjunction with telephone and postal follow-up for questionnaire surveys such as EQ-5 D, and Stroke Impact Scale. These data will be counter-checked using discharge coding records, which record each hospital episode.

The clinical teams will retrieve case records to collect (1) baseline measures which were not recorded in baseline Network surveys and (2) outcome measures including mortality and hospital length of stay. At study commencement (October 2010) one year follow up data can be collected immediately for October 2009 cohorts (follow up complete at end September 2010). The follow up will be completed in September 2012 as the stroke patients included in the last survey for the study conducted by the Network in July 2011 will complete one year follow-up in June 2012 and data collection of the study will be completed by July-August 2012 with the view of final cohort data arrival to research team by the end of December 2012.

Due to multi-centre nature of the study the individual sites are expected to join the study at different time points (after their respective NHS Research & Development Committees' approval). We will collect characteristics of stroke services, patient related factors, prognostic indicators, treatment options and trial/study participation. Missing prognostic data will be imputed statistically, to ensure that all eligible patients are included in the primary analysis (see also Statistical Methods).

The service characteristics of interest include:

At hospital level

- staffing (including junior doctors and therapists (whole time equivalent), physicians characteristics
 - university or district general hospital
 - distance from tertiary referral centre

- availability of vascular surgery on site, neuro-surgery and neuro ITU on site
- monitoring beds
- physician on call rota
- compliance with NICE guidelines

At patient level

- provision of thrombolysis and CT
- medication

Outcome measurements

Primary outcome of the study will be one year mortality comparison between services with different characteristics. The secondary outcomes will include (1) final discharge destination (good or poor outcome) [16], (2) length of acute hospital stay, (3) length of stay in rehabilitation, (4) complications during acute and rehab-hospital stay and significant procedures (e.g. aspiration pneumonia, myocardial infarction), (5) readmissions, (6) composite cardiovascular events (recurrent TIA/Stroke/Acute Coronary Syndrome, Myocardial infarction).

Patient Reported Outcome Measures (PROM)

PROM will consist of (1) Stroke Impact Scale, (2) health related quality of life: EQ-5 D at one year in those who completed questionnaire at the baseline, (3) modified RANKIN, (4) Barthel score and (5) health service use.

Statistical analysis

Quantitative data will be analysed by multivariate Cox-proportional hazards to examine the relationships between different aspects of health services and time to death, adjusting for prognostic characteristics. Multiple logistic or linear regression models will be constructed as appropriate for dichotomised and continuous outcome variables respectively. T tests for normally distributed data and Mann-Whitney U tests for non-normally distributed data will be used to compare continuous outcomes. Volume-outcome relationships will be investigated. Missing prognostic and EQ-5 D data will be imputed, based on each patient's other prognostic characteristics. Clustering of data by hospital trust will be investigated and, if necessary, taken into account, and intra-class correlation coefficients calculated to inform future research.

Economic evaluation

Health care resources are scarce and it is therefore important to ensure that evaluations are undertaken in order to ensure that services provided by the NHS constitute value for money. Within this study we will thereby seek to estimate the cost-effectiveness of different stroke service deliveries.

Costs will first be calculated from the perspective of the NHS and personal social services (PSS). Thus, levels

of resources use will be recorded during the follow-up period, including the length of original hospital stay, input by the multi-disciplinary team, other investigations (e.g. x-ray) and any complications (including details of any further hospital admissions). Unit costs will subsequently be assigned to each of these resource items, enabling both the total mean cost in participants and the incremental cost between two different service deliveries (chosen to compare the cost effectiveness, e.g. traditional on call rota vs. telemedicine) to be calculated after adjusting for other factors. The main measure of effectiveness to be used in the economic analysis will be the EQ-5 D [17], where responses will be sought at baseline, and at 12 month as mentioned above. This will enable the overall effect of each mode of service delivery, and the incremental effect of services to be estimated.

Outcome

As the National Institute of Health and Clinical Excellence [18] recommends use of the EQ-5 D [17] within cost-effectiveness analysis this will be our primary measure within the economic analyses. EQ-5 D data will be collected at two University Hospitals and two district general hospitals within the clinical network. We will use “mapping” strategy to estimate the cost-effectiveness analyses across the region. The use of mapping, where scores from a condition-specific (non preference-based) measure are ‘converted’ into a utility (preference-based) score using a pre-defined formulae, has been advocated (in certain instances) by the UK National Institute of Health and Clinical Excellence (NICE) [18], and has been used to estimate the utility scores, and in turn cost-effectiveness, of a number of health care interventions [19]. Mapping presents the possibility of not asking all participants to complete the EQ-5 D. In this study we propose to take advantage of this by developing a mapping algorithm based on the response from participants participating in this component to predict the EQ-5 D for participants in retrospective cohorts and those who did not participate in PROM component.

Because the quality of life measure (EQ-5D) which can be used to estimate health utility and calculate QALYs (Quality Adjusted Life Years) for economic evaluation is outside the remit of routine data collection and cannot be done retrospectively, we will collect EQ-5 D data in only the second year of the study (October 2010 and January, April and July 2011 cohorts and one year follow up data to be collected September and December 2011, and March and June 2012) in those who provide informed consent to the study (we estimate that the sample will be approximately 15-20% of the whole sample after excluding the one year pre-study period (between October 2009-September 2010) and after

taking into account of refusal rate (estimated ~ 30%) in trusts with Stroke or Comprehensive Local Research Network Research Nurses.

Economic Analysis

In the Economic analysis if one option is shown to be less costly and more effective than another option (for example, telemedicine vs. on call system) then that option will ‘dominate’ the other and be deemed cost-effective. Alternatively, the incremental cost-effectiveness ratio (ICER) associated with a particular option will be estimated and assessed in relation to a range of cost-effectiveness thresholds. The associated level of uncertainty will also be characterised by e.g. estimating the cost-effectiveness acceptability curve (CEAC) for each intervention and conducting value of information analysis [20]. Sensitivity analysis will also be undertaken to assess the robustness of conclusions to key assumptions. We will also seek to identify what resource items should be monitored in a future study (i.e. what are the big cost drivers which are likely to be affected by the intervention) and how these items should be identified.

The study is funded by the NIHR Research for Patient Benefit Programme (PB-PG-1208-18240) and obtained ethical approval from the Norfolk Research Ethics Committee.

Discussion

In this study we specifically aim to identify services that are associated with the best clinical outcomes including mortality and hospital length of stay including patient reported outcome adjusting for patient prognostic factors and potential confounders. Our study will be able to provide useful information in stroke service provision in UK and beyond. Furthermore, inclusion of patient reported outcome is novel and exciting component of our study.

Studies which have examined the delivery of specific services such as rapid imaging, have shown improvement in patients’ outcome in stroke [21]. A recent report from Germany suggested that a telestroke network may be a useful strategy to implement in their non-urban stroke services [22]. Lees et al (2008) [23] highlighted that there is room for improvement in terms of acute services for stroke. Interestingly, one of the observations was that centres with higher workload performed better. There is also existing evidence in Cancer literature that centres with higher surgical caseload have better outcomes [24]. There has also been a recent evaluation of the impact on stroke outcome by evidence-based practice in an Australian setting [25]. Examples of service delivery that are associated with better outcomes include organised stroke unit care [26], thrombolysis treatment and appropriate secondary prevention [27], and early supported discharge

in selected patients [28,29]. However, the cost-effectiveness of such services has yet to be fully examined.

Rodgers et al [30] highlighted the need for improvement in hospital-based stroke services e.g. stroke unit staffing levels were lower than was available in RCTs. The accumulating body of evidence has been a major driving force behind the UK Government's strategy to improve stroke care (National Stroke Strategy, 2007) [31]. A key strand of the strategy was to set up stroke networks to deliver stroke service development across geographically defined areas. The stroke networks have worked to agree minimum standards for stroke care and they have worked with commissioners to assist the commissioning process for stroke services. The acute stroke services are currently delivered by different NHS trusts and there is therefore a wide range of inequality in service availability and provision with differing structure and local support systems.

This research aims to utilise NHS data in the most meaningful and innovative way and we aim to maximize the benefit with minimum investment to produce best research output for patient care by collaborating with clinical teams and the network in providing excellent value for money. This observational study seeks to identify areas of clinical practice which merit future randomised controlled trials (RCTs) to identify best practice in improving stroke care which will be of maximum benefit to patients. We also aim to obtain preliminary data to estimate sample sizes and conduct value of information analyses to design future pragmatic RCTs of innovative ways of delivering stroke care.

As we include eight diverse NHS trusts, the findings are likely to be generalisable in the UK setting and beyond. This study will provide observational data about health service factors associated with variations in patient outcomes and health care costs following hospital admission for acute stroke. This will form the basis for future RCTs by identifying promising health service interventions, assessing the feasibility of recruiting and following up trial patients, and provide evidence about frequency and variances in outcomes, and intra-cluster correlation of outcomes, for sample size calculations. The results will also inform clinicians, public, service providers, commissioners and policy makers to drive further improvement in health services and bring direct benefit to patients.

The study will describe the variation in outcomes between different stroke services, and identify the characteristics of services associated with better outcomes after accounting for case-mix. We will also estimate the relative costs of and health gain estimated as Quality Adjusted Life Year (QALY) gain that may be demonstrated by different services. The commissioners of services will be informed as to which service delivery

structures are likely to provide value for money to make purchasing decisions. They will also be better informed about the types of service associated with better patient reported outcome. Hospital trusts will be able to evaluate their services systematically and plan their care appropriately to meet local and regional needs and demands based on our study findings. Professionals will be able to reflect on the impact of services they are delivering to help improve their performance and the way services are organised by adopting the most effective and cost effective approaches. As an observational study, the study limitations include inability to control for unknown confounders and residual confounding effect of known confounders which are adjusted for. The causal relationship cannot be implied but as we stated the findings will provide knowledge about areas that requires further evaluation in clinical trial setting.

There is very little work which assesses service provision robustly against patients' own reported outcomes. This exciting study may lead to a clearer drive for patients to define what makes a good service. We hope that the best clinical practices are adopted to suit the local populations' needs and demand. As we included eight diverse NHS trusts, the findings will be generalisable in the UK setting and likely to be applicable in international setting. All these will become drivers of improvement in stroke services for the benefit of stroke sufferers.

Acknowledgements

We would like to thank the participants of the study. We gratefully acknowledge the contribution of Stroke Research Nurses and the study steering committee members including representatives from the Regional Stroke Association, and Patient and Public Involvement in Research Panel. We would like to thank our colleagues from all participating trusts, site data co-ordinator and staff from Anglia Stroke & Heart Network for their assistance. Norfolk and Norwich University Hospital NHS Foundation Trust sponsors the study.

Manuscripts that are under submission based on this protocol
None.

Author details

¹Norwich Medical School, Faculty of Medicine & Health Sciences, Norwich, UK. ²Norfolk & Norwich University Hospital, Norwich, UK. ³Anglia Stroke & Heart Clinical Network, Cambridge, UK. ⁴James Paget University Hospital, Lowestoft, UK. ⁵Queen Elizabeth Hospital, King's Lynn, UK. ⁶Peterborough City Hospital, Peterborough, UK. ⁷Hinchingbrooke Hospital, Huntingdon, UK. ⁸Ipswich Hospital, Ipswich, UK. ⁹West Suffolk Hospital, Bury St Edmund, UK. ¹⁰Addenbrooke's University Hospital, Cambridge, UK.

Authors' contributions

PKM, DJD, MOB designed the outline of the study. PKM, JFP, MOB, EAW, GMP, GAB and AKM obtained the funding for the study. SDM & RH contributed in protocol preparation. All authors contributed in writing of the paper. All authors read and approved the final manuscript. PKM is the guarantor.

Competing interests

The authors declare that they have no competing interests.

Received: 21 January 2011 Accepted: 28 February 2011
Published: 28 February 2011

References

- National Institute for Health and Clinical Excellence: **STROKE: National clinical guideline for diagnosis and initial management of acute stroke and transient ischaemic attack (TIA). The National Collaborative Centre for Chronic Conditions.** *Royal College of Physicians of London* 2008 [<http://www.nice.org.uk/nicemedia/pdf/CG68FullGuideline.pdf>], ISBN 978-1-86016-339-5.
- National Audit Office: **Reducing brain damage: faster access to better stroke care.** *London: Stationery Office, 2005* National Audit Office Report; 2005.
- Rothwell PM, Coull AJ, Silver LE, Fairhead JF, Giles MF, Lovelock CE, Redgrave JN, Bull LM, Welch SJ, Cuthbertson FC, Binney LE, Gutnikov SA, Anslow P, Banning AP, Mant D, Mehta Z, Oxford Vascular Study: **Population-based study of event-rate, incidence, case fatality and mortality for all acute vascular events in all arterial territories (Oxford Vascular Study).** *Lancet* 2005, **366**:1773-83.
- Grieve R, Hutton J, Bhalla A, Rastenyte D, Ryglewicz D, Sarti C, Lamassa M, Giroud M, Dundas R, Wolfe CD: **A comparison of the costs and survival of hospital-admitted stroke patients across Europe.** *Stroke* 2001, **32**:1684-91.
- Weir NU, Sandercock PA, Lewis SC, Signorini DF, Warlow CP: **Variations between countries in outcome after stroke in the International Stroke Trial (IST).** *Stroke* 2001, **32**:1370-7.
- Gray LJ, Sprigg N, Bath PM, Sørensen P, Lindenstrøm E, Boysen G, De Deyn PP, Friis P, Leys D, Marttila R, Olsson JE, O'Neill D, Ringelstein B, van der Sande JJ, Turpie AG, TAIST Investigators: **Significant variation in mortality and functional outcome after acute ischaemic stroke between Western countries: data from the tinzaparin in acute ischaemic stroke trial (TAIST).** *J Neurol Neurosurg Psychiatry* 2006, **77**:327-33.
- Markus H: **Improving the outcome of stroke.** *BMJ* 2007, **335**:359-60.
- Rudd AG, Irwin P, Rutledge Z, Lowe D, Wade DT, Pearson M: **Regional variations in stroke care in England, Wales and Northern Ireland: results from the National Sentinel Audit of Stroke.** *Royal College of Physicians Intercollegiate Stroke Working Party.* *Clin Rehabil* 2001, **15**:562-72.
- Rudd AG, Lowe D, Hoffman A, Irwin P, Pearson M: **Secondary prevention for stroke in the United Kingdom: results from the National Sentinel Audit of Stroke.** *Age Ageing* 2004, **33**:280-6.
- Rudd AG, Hoffman A, Irwin P, Pearson M, Lowe D, Intercollegiate Working Party for Stroke: **Stroke units: research and reality. Results from the National Sentinel Audit of Stroke.** *Qual Saf Health Care* 2005, **14**:7-12.
- Howell E, Graham C, Hoffman A, Lowe D, McKeivitt C, Reeves R, Rudd AG: **Comparison of patients' assessments of the quality of stroke care with audit findings.** *Qual Saf Health Care* 2007, **16**:450-5.
- National Sentinel Stroke Audit: **Phase II (clinical audit), 2008. Report for England, Wales and Northern Ireland.** *Intercollegiate Stroke Working Party.* *Clinical Effectiveness and Evaluation Unit, Royal College of Physicians of London* 2009.
- Myint PK, Vowler SL, Woodhouse PR, Redmayne O, Fulcher RA: **Winter excess in hospital admissions, in-patient mortality and length of acute hospital stay in stroke: a hospital database study over six seasonal years in Norfolk, UK.** *Neuroepidemiology* 2007, **28**:79-85.
- Myint PK, Kamath AV, Vowler SL, Maisey DN, Harrison BD: **The CURB (confusion, urea, respiratory rate and blood pressure) criteria in community-acquired pneumonia (CAP) in hospitalised elderly patients aged 65 years and over: a prospective observational cohort study.** *Age Ageing* 2005, **34**:75-7.
- Myint PK, Vowler SL, Redmayne O, Fulcher RA: **Utilisation of diagnostic computerised tomography imaging and immediate clinical outcomes in older people with stroke before and after introduction of the National Service Framework for older people. A comparative study of hospital-based stroke registry data (1997-2003): Norfolk experience.** *Age Ageing* 2006, **35**:399-403.
- Myint PK, Vowler SL, Redmayne O, Fulcher RA: **Cognition, continence and transfer status at the time of discharge from an acute hospital setting and their associations with an unfavourable discharge outcome after stroke.** *Gerontology* 2008, **54**:202-9.
- Brooks R: **EuroQol: the current state of play.** *Health Policy* 1996, **37**:53-72.
- National Institute for Clinical Excellence: 2004 [http://www.nice.org.uk/aboutnice/howwework/devicetech/technologyappraisalprocessguides/guide_to_the_methods_of_technology_appraisal_reference_n0515.jsp], (reference N0515). ISBN: 1-84257-595-3.
- Barton GR, Sach TH, Jenkinson C, Avery AJ, Doherty M, Muir KR: **Do estimates of cost-utility based on the EQ-5 D differ from those based on the mapping of utility scores?** *Health Qual Life Outcomes* 2008, **6**:51.
- Barton GR, Briggs AH, Fenwick EA: **Optimal cost-effective decisions: the role of the cost-effectiveness acceptability curve (CEAC), cost-effectiveness acceptability frontier (CEAF) and expected value of perfect information (EVPI).** *Value Health* 2008, **11**:886-897.
- Wardlaw JM, Seymour J, Cairns J, Keir S, Lewis S, Sandercock P: **Immediate computed tomography scanning of acute stroke is cost-effective and improves quality of life.** *Stroke* 2004, **35**:2477-83.
- Audebert HJ, Kukla C, Vatankeh B, Gotzler B, Schenkel J, Hofer S, Fürst A, Haberl RL: **Comparison of tissue plasminogen activator administration management between Telestroke Network hospitals and academic stroke centers: the Telemedical Pilot Project for Integrative Stroke Care in Bavaria/Germany.** *Stroke* 2006, **37**:1822-7.
- Lees KR, Ford GA, Muir KW, Ahmed N, Dyker AG, Atula S, Kalra L, Warburton EA, Baron JC, Jenkinson DF, Wahlgren NG, Walters MR, SITS-UK Group: **Thrombolytic therapy for acute stroke in the United Kingdom: experience from the safe implementation of thrombolysis in stroke (SITS) register.** *QJM* 2008, **101**:863-9.
- Bachmann MO, Alderson D, Edwards D, Wotton S, Bedford C, Peters TJ, Harvey IM: **Cohort study in South and West England of the influence of specialization on the management and outcome of patients with oesophageal and gastric cancers.** *Br J Surg* 2002, **89**:914-22.
- Gattellari M, Worthington J, Jalaludin B, Mohsin M: **Stroke unit care in a real-life setting: can results from randomized controlled trials be translated into every-day clinical practice? An observational study of hospital data in a large Australian population.** *Stroke* 2009, **40**:10-7.
- Stroke Unit Trialists' Collaboration: **Organised inpatient (stroke unit) care for stroke.** *Cochrane Database Syst Rev* 2007, **4**: CD000197, Review.
- Higgins P, Lees KR: **Advances in emerging therapies.** *Stroke* 2009, **40**(5): e292-4.
- Langhorne P, Taylor G, Murray G, Dennis M, Anderson C, Bautz-Holter E, Dey P, Indredavik B, Mayo N, Power M, Rodgers H, Ronning OM, Rudd A, Suwanwela N, Widen-Holmqvist L, Wolfe C: **Early supported discharge services for stroke patients: a meta-analysis of individual patients' data.** *Lancet* 2005, **365**:501-6.
- Early Supported Discharge Trialists: **Services for reducing duration of hospital care for acute stroke patients.** *Cochrane Database Syst Rev* 2005, **2**: CD000443.
- Rodgers H, Dennis M, Cohen D, Rudd A, British Association of Stroke Physicians: **British Association of Stroke Physicians: benchmarking survey of stroke services.** *Age Ageing* 2003, **32**:211-7.
- National Stroke Strategy: **Department of Health. 2007** [http://www.dh.gov.uk/en/Publicationsandstatistics/Publications/PublicationsPolicyandGuidance/DH_081062].

Pre-publication history

The pre-publication history for this paper can be accessed here:
<http://www.biomedcentral.com/1472-6963/11/50/prepub>

doi:10.1186/1472-6963-11-50

Cite this article as: Myint *et al.*: Evaluation of stroke services in Anglia stroke clinical network to examine the variation in acute services and stroke outcomes. *BMC Health Services Research* 2011 **11**:50.

Submit your next manuscript to BioMed Central and take full advantage of:

- Convenient online submission
- Thorough peer review
- No space constraints or color figure charges
- Immediate publication on acceptance
- Inclusion in PubMed, CAS, Scopus and Google Scholar
- Research which is freely available for redistribution

Submit your manuscript at
www.biomedcentral.com/submit

Supplementary document to:

Does service heterogeneity have an impact on acute hospital length of stay in stroke? A UK-based multi-centre prospective cohort study

Michelle Tørnes, David McLernon, Max O Bachmann, Stanley D Musgrave, Elizabeth A Warburton, John F Potter, Phyo Kyaw Myint: On behalf of the Anglia Stroke Clinical Network Evaluation Study (ASCNES) Group

For peer review only

CONTENT

Table S1 Variables used to inform multiple imputation of missing data

Table S2 Sample characteristics distribution of the 2333 patients included in analysis per individual hospital (n (%)) unless otherwise stated)

Table S3 Sample characteristics of cases with complete data and those with at least one variable missing

Table S4 Univariable regression analysis for multiply imputed dataset (n=2233) for AHLOS

Table S5 Univariable linear regression complete case analysis for AHLOS

Table S6 Multiple linear regression complete case analysis for AHLOS (n=1554, R²=43%)

Figure S1 Model estimates of mean baseline AHLOS per hospital and presence of vascular surgery onsite with 95% confidence intervals

Figure S2 Model estimates of mean baseline AHLOS per hospital and distance to neurosurgical facility with 95% confidence intervals

Figure S3 Model estimates of mean baseline AHLOS per hospital AHLOS and number of fte senior doctors per five beds available during weekdays with 95% confidence intervals

Figure S4 Model estimates of mean baseline AHLOS per hospital and number of fte junior doctors per five beds available during weekdays with 95% confidence intervals

Figure S5 Model estimates of mean baseline AHLOS per hospital and number of fte health care associates and nurses per five beds with 95% confidence intervals

Figure S6 Model estimates of mean baseline AHLOS per hospital and number of fte occupational therapists per five beds with 95% confidence intervals

**Figure S7** Model estimates of mean baseline AHLOS per hospital and number of fte
physiotherapists per five beds with 95% confidence intervals

**Figure S8** Model estimates of mean baseline AHLOS per hospital and number of fte speech
and language therapists per five beds with 95% confidence intervals

**Figure S9** Model estimates of mean baseline AHLOS per hospital and number of total beds
present on stroke unit with 95% confidence intervals

**Figure S10** Model estimates of mean baseline AHLOS per hospital and number of hospital
beds per CT scanner with 95% confidence intervals

**Figure S11** Model estimates of mean baseline AHLOS per hospital and provision of onsite
rehabilitation service with 95% confidence intervals

**Figure S12** Model estimates of mean baseline AHLOS per hospital and presence of early
supported discharge scheme with 95% confidence intervals

**Figure S13** Model estimates of mean baseline AHLOS per hospital and number of non-
stroke patients present on the stroke unit per day per five stroke unit beds with 95%
confidence intervals

**Figure S14** Model estimates of mean baseline AHLOS per hospital and number of stroke
patients treated outside the stroke unit per day per five stroke unit beds with 95% confidence
intervals

Table S1 Variables used to inform multiple imputation of missing data

Variable	Measure
I. Independent Variables	
Trust	0=Trust 1 1 =Trust 2 2 =Trust 3 3=Trust 4 4=Trust 5 4=Trust 6 5=Trust 7 6=Trust 8
Sex	0=Male 1=Female
Age	Continuous, years
Recurrent Stroke	0=No 1=Yes
Diabetes Mellitus	0=No 1=Yes
Dementia	0=No 1=Yes
Hypercholesterolemia	0=No 1=Yes
Myocardial Infarction or Ischaemic Heart Disease	0=No 1=Yes
Transient Ischaemic Attack	0=No 1=Yes
Previous Cancer	0=No 1=Yes
Active Cancer	0=No 1=Yes
Depression	0=No 1=Yes
Rheumatoid Arthritis	0=No 1=Yes
Chronic Obstructive Pulmonary Disease	0=No 1=Yes
Pre-Stroke Rankin Score (mRS)	0=0 1=1 2=2 3=3 4=4 & 5
Pre-Stroke Residence	0=Independent living without formal care 1=Independent living with formal care 2=Institutional care
Stroke Type	0=Ischaemic 1=Haemorrhagic
Oxfordshire Community Stroke Classification	0=LACS 1=PACS 2=POCS 3=TACS
Brain Lateralisation	0=Yes 1=No
Inpatient Complication	0=No 1=Yes
Discharge Destination	0=Independent living without formal care 1=Independent living with formal care 2=Institutional care 3=Interim or rehabilitation setting 4=Death
Season of Admission	0=Summer 1=Winter
Day of Admission	0=Weekday 1=Weekend
II. Dependent Variable	
Logarithmic acute hospital LOS	Continuous, days
III. Auxiliary Variables	
Discharge Rankin Score (mRS)	0=0 1=1 2=2 3=3 4=4 5=5 6=6
Atrial Fibrillation	0=No 1=Yes
Baseline Systolic Blood Pressure	Continuous, mmHg
Baseline Diastolic Blood Pressure	Continuous, mmHg
Glucose Concentration on Admission	Continuous, mmol/L
Weight	Continuous, kg
Heart Rate	Continuous, beats per minute
Temperature	Continuous, °C
Oxygen Saturation	Continuous, %
ITU or HDU admission	0. No 1. Yes
Systolic Blood Pressure at Discharge	Continuous, mmHg
Diastolic Blood Pressure at Discharge	Continuous, mm Hg

Table S2 Sample characteristics of the 2333 patients included in analysis per individual hospital (n (%) unless otherwise stated)

Variables	Hospital 1	Hospital 2	Hospital 3	Hospital 4	Hospital 5	Hospital 6	Hospital 7	Hospital 8
	350 (16)	16 (1)	350 (16)	143 (6)	618 (28)	281 (13)	252 (11)	223 (10)
Age, y, median (IQR)	78 (68 to 85)	87 (81 to 92)	79 (72 to 86)	79 (70 to 86)	79 (71 to 85)	78 (71 to 85)	80 (68 to 85)	80 (71 to 87)
Sex, female	180 (52)	9 (56)	197 (56)	76 (53)	309 (50)	155 (55)	116 (46)	123 (55)
Recurrent Stroke	50 (14)	5 (31)	61 (17)	19 (17)	143 (23)	62 (22)	66 (26)	42 (19)
Diabetes Mellitus	48 (14)	1 (6)	59 (17)	17 (15)	92 (15)	66 (23)	44 (17)	43 (19)
Dementia	26 (7)	1 (6)	35 (10)	10 (9)	58 (9)	29 (10)	23 (9)	25 (11)
Hypercholesterolemia	48 (14)	3 (19)	24 (7)	7 (6)	61 (10)	80 (28)	38 (15)	94 (42)
Hypertensive	225 (64)	8 (50)	202 (58)	56 (50)	446 (72)	200 (71)	187 (74)	159 (71)
Myocardial Infarction or Ischaemic Heart Disease	45 (13)	3 (19)	87 (25)	30 (27)	142 (23)	80 (28)	49 (19)	81 (36)
Transient Ischaemic Attack	32 (9)	3 (19)	58 (17)	17 (15)	113 (18)	40 (14)	47 (19)	30 (13)
Previous Cancer	33 (9)	1 (6)	38 (11)	12 (11)	41 (7)	18 (6)	21 (8)	31 (14)
Active Cancer	24 (7)	2 (12)	8 (2)	10 (9)	49 (8)	9 (3)	20 (8)	15 (7)
Depression	13 (4)	0 (0)	17 (5)	8 (7)	33 (5)	11 (4)	18 (7)	17 (8)
Rheumatoid Arthritis	11 (3)	1 (6)	43 (12)	3 (3)	83 (13)	2 (1)	7 (3)	4 (2)
COPD	15 (4)	1 (6)	20 (6)	6 (5)	26 (4)	20 (7)	11 (4)	17 (8)
Pre-stroke Rankin Score								
0	84 (43)	3 (19)	117 (36)	-	330 (56)	126 (64)	136 (56)	118 (53)
1	60(31)	3 (19)	75 (23)	-	87 (15)	16 (8)	61 (25)	33 (15)
2	24 (12)	3 (19)	51 (16)	-	56 (9)	17 (9)	16 (7)	24 (11)
3	21 (11)	2 (12)	38 (12)	-	60 (10)	20 (10)	15 (6)	28 (13)
4 & 5	7 (4)	5 (31)	44 (14)	-	57 (10)	18 (9)	16 (7)	20 (9)
Pre-Stroke Residence								
Independent living with formal care	21 (6)	4 (25)	23 (7)	15 (14)	62 (10)	30 (11)	34 (13)	21 (10)
Independent living w/o formal care	292 (86)	9 (56)	285 (82)	86 (77)	493 (80)	215 (77)	193 (77)	179 (82)
Institution	28 (8)	3 (19)	40 (11)	10 (9)	63 (10)	35 (12)	23 (9)	18 (8)

Variables	Hospital 1	Hospital 2	Hospital 3	Hospital 4	Hospital 5	Hospital 6	Hospital 7	Hospital 8
	350 (16)	16 (1)	350 (16)	143 (6)	618 (28)	281 (13)	252 (11)	223 (10)
Stroke Type								
Ischaemic	293 (85)	14 (100)	286 (87)	90 (91)	541 (88)	233 (85)	213 (87)	194 (88)
Haemorrhagic	50 (15)	0 (0)	43 (13)	9 (9)	73 (12)	40 (15)	32 (13)	26(12)
Oxford Community Stroke Project Classification								
LACS	64 (24)	1 (7)	95 (29)	20 (28)	149 (25)	51 (18)	39 (19)	84 (39)
PACS	117 (43)	11 (79)	109 (33)	38 (54)	216 (37)	147 (53)	80 (39)	66 (30)
POCS	51 (19)	-	29 (9)	3 (4)	117 (20)	21 (8)	33 (16)	25 (12)
TACS	38 (14)	2 (14)	99 (30)	10 (14)	107 (18)	57 (21)	52 (25)	42 (19)
No Brain Lateralisation	50 (15)	2 (13)	14 (4)	9 (9)	129 (21)	1 (0.4)	30 (12)	9 (4)
Inpatient Complication	108 (31)	4 (25)	34 (10)	36 (25)	229 (37)	109 (39)	83 (33)	52 (23)
Discharge Destination								
Death	53 (17)	5 (31)	77 (22)	29 (21)	110 (18)	58 (21)	47 (19)	35 (16)
Independent living with formal care	54 (17)	2 (12)	24 (7)	3 (2)	34 (6)	42 (15)	44 (18)	21 (10)
Independent living w/o formal care	147 (48)	7 (44)	141 (40)	78 (55)	272 (44)	112 (40)	119 (47)	130 (59)
Institution	29 (9)	2 (12)	57 (16)	13 (9)	50 (8)	33 (12)	34 (14)	34 (15)
Interim/rehab Setting	26 (8)	0 (0)	51 (15)	18 (13)	152 (25)	32 (12)	7 (3)	1 (0)
Winter Admission	172 (49)	16 (100)	181 (52)	73 (51)	332 (54)	140 (50)	131 (52)	114 (51)
Weekend Admission	113 (32)	3 (19)	98 (28)	43 (30)	177 (29)	74 (26)	55 (22)	51 (23)

Table S3 Sample characteristics of complete cases and those with at least one variable missing

Patient Characteristic	Complete Cases (n=1486)	Cases with at least one missing variable (n=747)
	Median (IQR) or No. (%)	
Age, y	79 (71 to 86)	79 (69 to 86)
Sex, female	777 (52)	388 (52)
Recurrent Stroke	327 (22)	121 (17)
Diabetes Mellitus	257 (17)	113 (16)
Dementia	138 (9)	69 (10)
Hypercholesterolemia	262 (18)	93 (13)
Hypertensive	1047 (70)	436 (61)
Myocardial Infarction or Ischaemic Heart Disease	361 (24)	156 (22)
TIA	248 (17)	92 (13)
Previous Cancer	140 (9)	55 (8)
Active Cancer	91 (6)	46 (6)
Depression	78 (5)	39 (5)
Rheumatoid Arthritis	129 (9)	25 (3)
COPD	76 (5)	40 (6)
Pre-stroke Rankin Score		
0	758 (51)	156 (51)
1	281 (19)	54 (18)
2	167 (11)	24 (8)
3	149 (10)	35 (11)
4 & 5	131 (9)	36 (12)
Pre-stroke Residence		
Independent living with formal care	145 (10)	65 (9)
Independent living without formal care	1205 (81)	547 (79)
Institution	136 (9)	84 (12)
Haemorrhagic Stroke	138 (9)	135 (21)
Oxford Community Stroke Project Classification		
LACS	405 (27)	98 (20)
PACS	569 (38)	215 (44)
POCS	211 (14)	68 (14)
TACS	301 (20)	106 (22)
No Brain Lateralisation	173 (12)	71 (12)
Inpatient Complication	419 (28)	236 (32)
Discharge Destination		
Death	233 (16)	181 (26)
Independent living with formal care	145 (10)	79 (11)
Independent living without formal care	708 (48)	298 (43)
Institution	181 (12)	71 (10)
Interim/Rehab Setting	219 (15)	68 (10)
Winter Admission	766 (52)	393 (53)
Weekend Admission	401 (27)	213 (29)

Table S4 Univariable regression analysis for multiply imputed dataset for AHLOS (n=2233).

Patient Characteristic	β	95% CI	P	R ²
Age, y	1.02	1.02 to 1.02	<0.001	4.8
Sex, female	1.20	1.10 to 1.31	<0.001	0.7
Recurrent Stroke	1.17	1.05 to 1.31	0.01	0.4
Diabetes Mellitus	1.16	1.03 to 1.31	0.02	0.3
Dementia	1.46	1.25 to 1.70	<0.001	1.1
Hypercholesterolemia	0.84	0.75 to 0.95	0.01	0.3
Hypertensive	1.02	0.93 to 1.12	0.66	0
Myocardial Infarction/ Ischaemic Heart Disease*	1.07	0.96 to 1.19	0.23	0.1
TIA	1.07	0.94 to 1.21	0.30	0.1
Previous Cancer	1.23	1.05 to 1.44	0.01	0.3
Active Cancer	0.97	0.80 to 1.16	0.72	0
Depression	1.06	0.86 to 1.29	0.59	0
Rheumatoid Arthritis	1.10	0.92 to 1.31	0.31	0.1
COPD	0.86	0.71 to 1.06	0.15	0.1
Pre-stroke Rankin Score (reference 0)			<0.001	5.5
1	1.57	1.38 to 1.79	<0.001	
2	1.63	1.39 to 1.91	<0.001	
3	1.94	1.65 to 2.28	<0.001	
4 & 5	1.32	1.13 to 1.55	<0.001	
Pre-stroke Residence (reference Independent living w/o formal care)			<0.001	1.4
Independent living with formal care	1.52	1.31 to 1.77	<0.001	
Institution	1.13	0.97 to 1.31	0.11	
Haemorrhagic Stroke	0.83	0.73 to 0.96	0.01	0.3
Oxford Community Stroke Project Classification (reference LACS)			<0.001	4.0
PACS	1.62	1.44 to 1.82	<0.001	
POCS	1.22	1.05 to 1.42	0.01	
TACS	1.66	1.45 to 1.90	<0.001	
Brain Lateralisation	0.69	0.60 to 0.80	<0.001	1.2
Inpatient Complication	2.13	1.94 to 2.34	<0.001	10.3
Discharge Destination (reference Independent living w/o formal care)			<0.001	25.7
Independent living with formal care	2.56	2.24 to 2.93	<0.001	
Institution	4.44	3.91 to 5.05	<0.001	
Interim/Rehab Setting	2.61	2.31 to 2.94	<0.001	
Death	1.15	1.03 to 1.28	0.01	
Summer Admission	1.20	1.09 to 1.31	<0.001	0.7
Weekday Admission	1.08	0.98 to 1.20	0.12	0.1
Hospital (reference 1)			<0.001	2.4
2.69	1.58 to 4.58	<0.001
1.19	1.02 to 1.39	0.03
1.24	1.01 to 1.53	0.04
0.86	0.75 to 0.99	0.03
1.11	0.94 to 1.31	0.22
1.18	1.00 to 1.41	0.05
0.86	0.72 to 1.03	0.11

Table S5 Univariable linear regression complete case analysis for AHLOS

Patient Characteristic	N	β	95% CI	P	% R ²
Age, y	2231	1.02	1.02 to 1.02	<0.001	4.7
Sex, female	1165 v. 1066	1.20	1.10 to 1.31	<0.001	0.7
Recurrent Stroke	448 v. 1755	1.17	1.05 to 1.31	0.005	0.3
Diabetes Mellitus	370 v. 1833	1.16	1.03 to 1.31	0.02	0.2
Dementia	207 v. 1996	1.46	1.25 to 1.70	<0.001	1.0
Hypercholesterolemia	355 v. 1848	0.85	0.75 to 0.95	0.01	0.3
Hypertensive	1483 v. 720	1.03	0.93 to 1.13	0.57	0
Myocardial Infarction or Ischaemic Heart Disease*	517 v. 1686	1.07	0.96 to 1.19	0.23	0
TIA	340 v. 1863	1.06	0.94 to 1.20	0.32	0
Previous Cancer	195 v. 2008	1.23	1.05 to 1.44	0.01	0.3
Active Cancer	137 v. 2066	0.96	0.80 to 1.15	0.65	0
Depression	117 v. 2086	1.05	0.86 to 1.28	0.65	0
Rheumatoid Arthritis	154 v. 2049	1.10	0.92 to 1.31	0.31	0
COPD	116 v. 2087	0.86	0.70 to 1.05	0.14	0.1
Pre-stroke Rankin Score (reference 0)				<0.001	5.8
1	335 v. 914	1.58	1.39 to 1.80	<0.001	
2	191 v. 914	1.62	1.38 to 1.90	<0.001	
3	184 v. 914	1.97	1.67 to 2.31	<0.001	
4 & 5	167 v. 914	1.45	1.22 to 1.71	<0.001	
Pre-stroke Residence (reference Independent living without formal care)				<0.001	1.3
Independent living with formal care	210 v. 1752	1.52	1.31 to 1.77	<0.001	
Institution	220 v. 1752	1.14	0.98 to 1.32	0.09	
Haemorrhagic Stroke	273 v. 1864	0.85	0.74 to 0.97	0.02	0.2
Oxford Community Stroke Project Classification (reference LACS)				<0.001	4.3
PACS	784 v. 503	1.62	1.44 to 1.82	<0.001	
POCS	279 v. 503	1.24	1.06 to 1.44	0.01	
TACS	407 v. 503	1.75	1.53 to 2.01	<0.001	
No Brain Lateralisation	244 v. 1822	0.68	0.59 to 0.79	<0.001	1.3
Inpatient Complication	655 v. 1578	2.13	1.94 to 2.34	<0.001	10
Discharge Destination (reference Independent living without formal care)				<0.001	25
Independent living with formal care	414 v. 1006	2.56	2.24 to 2.92	<0.001	
Institution	224 v. 1006	4.38	3.86 to 4.97	<0.001	
Interim/Rehab Setting	252 v. 1006	2.61	2.31 to 2.94	<0.001	
Death	287 v. 1006	1.13	1.02 to 1.26	0.02	
Winter Admission	1159 v. 1074	1.20	1.09 to 1.31	<0.001	0.6
Weekend Admission	614 v. 1619	1.08	0.98 to 1.20	0.12	0.1
Hospital (reference 1)				<0.001	2.1
16 v. 350	2.69	1.58 to 4.58	<0.001
350 v. 350	1.19	1.02 to 1.39	0.03
v. 350	1.24	1.01 to 1.53	0.04
618 v. 350	0.86	0.75 to 0.99	0.03
281 v. 350	1.11	0.94 to 1.31	0.22
252 v. 350	1.18	1.00 to 1.41	0.05
223 v. 350	0.86	0.72 to 1.03	0.11

Table S6 Multiple Linear Regression Complete Case Analysis for AHLOS (n=1554, R²=43%).

Patient Characteristic	N	β	95% CI	P
Age, y	1554	1.01	1.00 to 1.01	<0.001
Sex, female	816 v. 738	0.96	0.88 to 1.04	0.32
Recurrent Stroke	335 v. 1219	1.00	0.91 to 1.11	0.98
Diabetes Mellitus	265 v. 1289	1.00	0.90 to 1.12	0.94
Dementia	142 v. 1412	1.26	1.08 to 1.46	0.003
Hypercholesterolemia	271 v. 1283	0.93	0.84 to 1.04	0.22
Myocardial Infarction or Ischaemic Heart	377 v. 1177	1.00	0.91 to 1.10	0.98
Previous Cancer	146 v. 1408	1.19	1.04 to 1.37	0.01
COPD	83 v. 1471	0.91	0.77 to 1.09	0.32
Pre-stroke Rankin Score (reference 0)				<0.001
1	294 v. 799	1.11	1.00 to 1.24	0.06
2	173 v. 799	1.16	1.01 to 1.33	0.04
3	155 v. 799	1.34	1.14 to 1.58	<0.001
4 & 5	133 v. 799	1.27	1.05 to 1.53	0.01
Pre-Stroke Residence (reference Independent living without formal care)				<0.001
Independent living with formal care	154 v. 1262	0.86	0.74 to 1.00	0.05
Institution	138 v. 1262	0.49	0.40 to 0.59	<0.001
Haemorrhagic Stroke	141 v. 1413	0.90	0.78 to 1.04	0.15
Oxford Community Stroke Project Classification				<0.001
PACS	602 v. 425	1.28	1.16 to 1.42	<0.001
POCS	220 v. 425	1.39	1.22 to 1.59	<0.001
TACS	307 v. 425	1.56	1.37 to 1.79	<0.001
No Brain Lateralisation	178 v. 1376	0.90	0.79 to 1.03	0.11
Inpatient Complication	442 v. 1112	1.65	1.50 to 1.82	<0.001
Discharge Destination (reference Independent living without formal care)				<0.001
Independent living with formal care	156 v. 743	1.97	1.70 to 2.29	<0.001
Institution	187 v. 743	3.73	3.17 to 4.39	<0.001
Interim/Rehab Setting	235 v. 743	2.17	1.91 to 2.46	<0.001
Death	233 v. 743	0.98	0.84 to 1.15	0.82
Winter Admission	798 v. 756	1.14	1.05 to 1.23	0.001
Weekend Admission	425 v. 1129	1.04	0.95 to 1.13	0.39
Hospital (reference 1)				<0.001
2	14 v. 134	2.80	1.81 to 4.34	<0.001
3	384 v. 134	1.39	1.18 to 1.65	<0.001
4	-	-	-	-
5	568 v. 134	0.90	0.77 to 1.04	0.16
6	159 v. 134	1.23	1.02 to 1.49	0.03
7	194 v. 134	1.37	1.14 to 1.63	<0.001
8	201 v. 134	1.11	0.93 to 1.33	0.23

Figure S1 Model estimates of mean baseline AHLOS per hospital and presence of vascular surgery onsite with 95% confidence intervals

Figure S2 Model estimates of mean baseline AHLOS per hospital and distance to neurosurgical facility with 95% confidence intervals

Figure S3 Model estimates of mean baseline AHLOS per hospital and number of fte senior doctors per five beds available during weekdays with 95% confidence intervals

Figure S4 Model estimates of mean baseline AHLOS per hospital and number of fte junior doctors per five beds available during weekdays with 95% confidence intervals

Figure S5 Model estimates of mean baseline AHLOS per hospital and number of fte health care associates and nurses per five beds with 95% confidence intervals

Figure S6 Model estimates of mean baseline AHLOS per hospital and number of full-time occupational therapists per five beds with 95% confidence intervals

Figure S7 Model estimates of mean baseline AHLOS per hospital and number of fte physiotherapists per five beds with 95% confidence intervals

Figure S8 Model estimates of mean baseline AHLOS per hospital and number of full-time speech and language therapists per five beds with 95% confidence intervals

Figure S9 Model estimates of mean baseline AHLOS per hospital and number of total beds present on stroke unit with 95% confidence intervals

Figure S10 Model estimates of mean baseline AHLOS per hospital and number of hospital beds per CT scanner with 95% confidence intervals

Figure S11 Model estimates of mean baseline AHLOS per hospital and provision of onsite rehabilitation service with 95% confidence intervals

Figure S12 Model estimates of mean baseline AHLOS per hospital and presence of early supported discharge scheme with 95% confidence intervals

Figure S13 Model estimates of mean baseline AHLOS per hospital and number of non-stroke patients present on the stroke unit per day per five stroke unit beds with 95% confidence intervals

Figure S14 Model estimates of mean baseline AHLOS per hospital and number of stroke patients treated outside the stroke unit per day per five stroke unit beds with 95% confidence intervals

STROBE 2007 (v4) Statement—Checklist of items that should be included in reports of cohort studies

Section/Topic	Item #	Recommendation	Reported on page #
Title and abstract	1	(a) Indicate the study’s design with a commonly used term in the title or the abstract	1 & 2
		(b) Provide in the abstract an informative and balanced summary of what was done and what was found	2
Introduction			
Background/rationale	2	Explain the scientific background and rationale for the investigation being reported	4
Objectives	3	State specific objectives, including any prespecified hypotheses	4 & 5
Methods			
Study design	4	Present key elements of study design early in the paper	6
Setting	5	Describe the setting, locations, and relevant dates, including periods of recruitment, exposure, follow-up, and data collection	6
Participants	6	(a) Give the eligibility criteria, and the sources and methods of selection of participants. Describe methods of follow-up	6
		(b) For matched studies, give matching criteria and number of exposed and unexposed	NA
Variables	7	Clearly define all outcomes, exposures, predictors, potential confounders, and effect modifiers. Give diagnostic criteria, if applicable	7 & 8
Data sources/ measurement	8*	For each variable of interest, give sources of data and details of methods of assessment (measurement). Describe comparability of assessment methods if there is more than one group	6
Bias	9	Describe any efforts to address potential sources of bias	8
Study size	10	Explain how the study size was arrived at	6
Quantitative variables	11	Explain how quantitative variables were handled in the analyses. If applicable, describe which groupings were chosen and why	7
Statistical methods	12	(a) Describe all statistical methods, including those used to control for confounding	8 & 9
		(b) Describe any methods used to examine subgroups and interactions	NA
		(c) Explain how missing data were addressed	8 & 9
		(d) If applicable, explain how loss to follow-up was addressed	NA
		(e) Describe any sensitivity analyses	NA
Results			

Participants	13*	(a) Report numbers of individuals at each stage of study—eg numbers potentially eligible, examined for eligibility, confirmed eligible, included in the study, completing follow-up, and analysed	10
		(b) Give reasons for non-participation at each stage	10
		(c) Consider use of a flow diagram	
Descriptive data	14*	(a) Give characteristics of study participants (eg demographic, clinical, social) and information on exposures and potential confounders	10 & 11
		(b) Indicate number of participants with missing data for each variable of interest	10 & 11
		(c) Summarise follow-up time (eg, average and total amount)	NA
Outcome data	15*	Report numbers of outcome events or summary measures over time	12 & 13
Main results	16	(a) Give unadjusted estimates and, if applicable, confounder-adjusted estimates and their precision (eg, 95% confidence interval). Make clear which confounders were adjusted for and why they were included	14 - 16
		(b) Report category boundaries when continuous variables were categorized	NA
		(c) If relevant, consider translating estimates of relative risk into absolute risk for a meaningful time period	NA
Other analyses	17	Report other analyses done—eg analyses of subgroups and interactions, and sensitivity analyses	15 & 17
Discussion			
Key results	18	Summarise key results with reference to study objectives	18
Limitations			
Interpretation	20	Give a cautious overall interpretation of results considering objectives, limitations, multiplicity of analyses, results from similar studies, and other relevant evidence	18 - 21
Generalisability	21	Discuss the generalisability (external validity) of the study results	21
Other information			
Funding	22	Give the source of funding and the role of the funders for the present study and, if applicable, for the original study on which the present article is based	22

*Give information separately for cases and controls in case-control studies and, if applicable, for exposed and unexposed groups in cohort and cross-sectional studies.

Note: An Explanation and Elaboration article discusses each checklist item and gives methodological background and published examples of transparent reporting. The STROBE checklist is best used in conjunction with this article (freely available on the Web sites of PLoS Medicine at <http://www.plosmedicine.org/>, Annals of Internal Medicine at <http://www.annals.org/>, and Epidemiology at <http://www.epidem.com/>). Information on the STROBE Initiative is available at www.strobe-statement.org.

BMJ Open

Does service heterogeneity have an impact on acute hospital length of stay in stroke? A UK-based multi-centre prospective cohort study

Journal:	BMJ Open
Manuscript ID	bmjopen-2018-024506.R1
Article Type:	Research
Date Submitted by the Author:	12-Dec-2018
Complete List of Authors:	Tørnes, Michelle; University of Aberdeen College of Life Sciences and Medicine, Ageing Clinical and Experimental Research Group McLernon, David; University of Aberdeen College of Life Sciences and Medicine, Medical Statistics Team Bachmann, Max; University of East Anglia, Norwich Medical School Musgrave, Stanley; University of East Anglia, Norwich Medical School Warburton, Elizabeth; University of Cambridge, Addenbrooke's Hospital Potter, John; University of East Anglia, Norwich Medical School; Norfolk and Norwich University Hospital, Stroke Research Group Myint, Phyo; University of Aberdeen College of Life Sciences and Medicine, Ageing Clinical and Experimental Research Group; Norfolk and Norwich University Hospital, Stroke Research Group
Primary Subject Heading:	Epidemiology
Secondary Subject Heading:	Health services research, Neurology, Geriatric medicine
Keywords:	Acute hospital, Health Services Research, Length of Stay, Outcome, Stroke < NEUROLOGY

**Does service heterogeneity have an impact on acute hospital length of stay in stroke? A UK-based**
**multi-centre prospective cohort study**

Michelle Tørnes¹, David McLernon², Max O Bachmann³, Stanley D Musgrave³, Elizabeth A
Warburton⁴, John F Potter³, Phyo Kyaw Myint^{1,5}: On behalf of the Anglia Stroke Clinical
Network Evaluation Study (ASCNES) Group

¹Ageing Clinical and Experimental Research (ACER) Group, Institute of Applied Health
Sciences, School of Medicine, Medical Sciences and Nutrition, University of Aberdeen,
Scotland, UK; ²Medical Statistics Team, Institute of Applied Health Sciences, School of
Medicine, Medical Sciences and Nutrition, University of Aberdeen, Scotland, UK; ³Norwich
Medical School, University of East Anglia, Norwich, UK; ⁴Addenbrooke's Hospital,
Cambridge, UK; ⁵Stroke Research Group, Norfolk and Norwich University Hospital,
Norwich, UK.

**Correspondence to:**

Michelle Tørnes,

C/o: Professor P K Myint; Room 4.013, Polwarth Building | School of Medicine & Dentistry |
Division of Applied Health Sciences | Foresterhill, University of Aberdeen | Aberdeen, AB25
2ZD, UK | Tel: +44 (0) 1224 437843 | Fax: +44 (0) 1224 437911 | Mail to:

michelle.tornes@abdn.ac.uk

Word count: 4821

**Keywords:** Acute hospital, Health Services Research, Length of Stay, Outcome, Stroke

ABSTRACT

Objectives: To determine whether stroke patients' acute hospital length of stay (AHLOS) varies between hospitals, over and above cases mix differences, and to investigate the hospital-level explanatory factors.

Design: A multicentre prospective cohort study.

Setting: Eight National Health Service acute hospital trusts within the Anglia Stroke & Heart Clinical Network in the East of England, UK.

Participants: The study sample was systematically selected to include all consecutive patients admitted within a month to any of the eight hospitals, diagnosed with stroke by an accredited stroke physician every third month between October 2009 and September 2011.

Primary and secondary outcome measures: AHLOS was defined as the number of days between date of hospital admission and discharge or death, whichever came first. We used a multiple linear regression model to investigate the association between hospital (as a fixed-effect) and AHLOS, adjusting for a number of important patient covariates, such as age, sex, stroke type, residence prior to stroke, Modified Rankin Scale score, comorbidities, and inpatient complications. Exploratory data analysis was utilized to gain insight into the hospital-level characteristics which may contribute to the hospital-level variance. These included hospital type, stroke monthly case volume, service provisions (i.e. onsite rehabilitation), and staffing levels.

Results: A total of 2233 stroke admissions (52% female, median age (interquartile range (IQR)) 79 (70 to 86) years, 83% ischaemic stroke) were included. The overall median AHLOS (IQR) was 9 (4 to 21) days. After adjusting for patient covariates, AHLOS still differed significantly between hospitals ($p < 0.001$). Furthermore, hospitals with the longest adjusted AHLOS's had predominantly lower stroke volumes.

1
2
3 **Conclusions:** We have clearly demonstrated that AHLOS varies between different hospitals.

[revised manuscript text omitted]

0	260 (12)	329 (15)
352 (16)
212 (9)
291 (13)
238 (11)
137 (6)
414 (19)
Winter Admission	1159 (52)	0 (0)
Weekend Admission	614 (27)	0 (0)

IQR, Interquartile Range; TIA, Transient Ischaemic Attack; COPD, Chronic Obstructive Pulmonary Disorder;
mRS, modified Rankin Scale; LACS, Lacunar Anterior Circulation Stroke; PACS, Partial Anterior Circulation
Stroke; POCS, Posterior Circulation Stroke; TACS, Total Anterior Circulation Stroke.

*No information was assumed to indicate absence of condition or complication

**Hospital service characteristics**

Service characteristics of each hospital are outlined in Table 2, with median AHLOS.

After standardization, by taking account of stroke admission volume, number of stroke unit
beds, and size of hospital, there was still extensive heterogeneity in bed capacity, staffing
levels, and the number of CT scanners provided at each hospital, respectively. Variations
between hospitals also existed in terms of service and facility provision. For example, a
number of hospitals provided rehabilitation care, neurosurgery or vascular surgery onsite,
whilst others did not. The overall median AHLOS (IQR) was 9 (4 to 21) days and there
appeared to be crude variations in this outcome between hospitals.

Table 2 Hospital characteristics per individual hospital self-reported by clinical leads or service managers at each hospital

Hospital Characteristics	1	2	3	4	5	6	7	8
General Characteristics								
Catchment Population	400,000	160,000	350,000	230,000	680,000	300,000	240,000	275,000
Hospital Type	Tertiary	Secondary	Secondary	Secondary	Tertiary	Secondary	Secondary	Secondary
Hospital Stroke Volume (No. of ASCNES admissions per month)	52	13	46	19	88	57	35	31
Facilities and Services								
No. of hospital beds	1000	304	800	500	1237	611	488	460
No. of stroke unit beds (per 100 admissions)	71	77	54	138	41	55	83	65
No. of hospital beds per CT scanners	500	304	400	250	518	306	244	230
Distance to Vascular Surgery (miles)	0	18	0	25	0	0	43	30
Distance to Neurosurgery (miles)	0	18	58	89	61	38	48	30
Rehabilitation Provision	Onsite	Onsite	Offsite	Offsite	Offsite	Onsite	Offsite	Onsite
Early Supported Discharge Provision	No	Yes	No	Yes	Yes	Yes	No	No
Stroke Unit Staffing Levels*								
Senior doctors †	0.34	0.25	0.49	0.47	0.42	0.31	0.62	0.87
Junior doctors †	0.55	0.65	0.72	0.59	0.56	0.64	0.12	0.25
Health care associates and nurses (band 5-7)	9.2	8	6	7.4	7	5.3	6.5	10
Physiotherapists (band 2-8)	0.55	1	0.79	0.4	0.91	0.78	0.69	1
Occupational Therapists (band 3-8)	0.49	0.5	1.4	0.59	0.6	0.58	0.52	1.1
Speech and Language Therapists	0.39	0.15	0.2	0.18	0.35	0.03	0.26	0.1
No. of non-stroke patients treated daily on stroke unit (per five stroke unit beds)	0.27	0	0.10	0.47	0.05	0.31	0.17	0
No. of stroke patients treated daily outside stroke unit (per five stroke unit beds)	0.14	5	0	0.30	0.01	0.41	0	0
Median AHLOS (IQR)	8 (4 to 20)	29 (24 to 42)	11 (5 to 27)	14 (4 to 30)	8 (4 to 14)	10 (5 to 22)	11 (6 to 23)	7 (3 to 20)

ASCNES, Anglia Stroke Clinical Network Evaluation Study; CT, Computerised Tomography; AHLOS, Acute Hospital Length of Stay; IQR, Interquartile Range.

*Number of fte staff per five stroke unit beds (weighted average for the four study periods taken). NHS banding refers to the pay scale system of healthcare staff in the UK
and relates to their level of experience. Higher bands reflect higher pay and experience.

† Weekday numbers only

For peer review only

Univariable linear regression

In univariable linear regression (Table S4 in the online supplementary document 2), patients who were older, female, had previous cancer, a previous stroke, had diabetes mellitus, had dementia, had a pre-stroke or discharge mRS score greater than 0, had a OCSP other than a lacunar infarct, had an inpatient complication, were living independently at home without formal care (compared to those who had formal care) prior to stroke, or were a winter admission had a significantly longer AHLOS ($p < 0.05$). Patients who had a haemorrhagic stroke, hypercholesterolemia, or showed no signs of brain lateralisation were all shown to be significantly associated with a shorter AHLOS ($p < 0.01$).

The strongest associations with AHLOS were seen for inpatients who developed a complication, who had a pre-stroke mRS score of 3, who were admitted to Hospital 2 or who had a discharge mRS score of ≥ 2 . Inpatient complications were associated with twice as long an AHLOS compared to those without a complication. Similarly, patients with a pre-stroke mRS score of 3 were 94% more likely to have a longer AHLOS than those with an mRS of 0. Patients admitted to Hospital 2 had 2.69 times the AHLOS of those admitted to Hospital 1. Compared to patients with a discharge mRS score, those with a score of 2, 3, 4 or 5 had over a 2, 3, 4, and 5-fold increase in AHLOS, respectively. Not unsurprisingly, discharge mRS score appeared to explain the majority of AHLOS variance ($R^2 = 31.1\%$).

Being hypertensive, having a history of a myocardial infarction or ischaemic heart disease, having previously had a TIA, having active cancer, depression, rheumatoid arthritis or chronic obstructive pulmonary disease were not shown to be significantly associated with AHLOS. Furthermore, admissions to Hospitals 6 and 8 were also not shown to be significantly associated with a difference in AHLOS compared to Hospital 1 admissions.

Multiple linear regression

Multiple linear regression results for AHLOS are summarized in Table 3 and shows that
42.7% of the variation in AHLOS has been explained. Sex, recurrent stroke, diabetes
mellitus, hypercholesterolemia, previous cancer, a pre-stroke mRS score of 1 to 3 (with
reference to a score of 0) and living at home independently without formal care prior to
stroke were no longer statistically associated with AHLOS in multiple regression ($p>0.05$).
Furthermore, being admitted to Hospital 3 or 4 as opposed to Hospital 1 were no longer
associated with a significant difference in AHLOS. No variables included from the
univariable analysis with $p>0.05$ became statistically significant in the multivariable analysis,
except for living in an institution prior to stroke which was associated with a 19% reduced
AHLOS compared to those living independently without formal care. Developing an
inpatient complication and having a discharge mRS score between 2 and 5 were still strongly
positively related to AHLOS. After adjusting for patient covariates, AHLOS was still shown
to significantly differ between hospitals, with the shortest and longest AHLOS observed for
Hospitals 5 and 2, respectively.

There were no obvious differences between the results using complete cases only (Tables S5-
6 in the online supplementary document 2) and multiple imputation.

Table 3 Multiple regression analysis for AHLOS (n=2233; R²=42.7%)

Patient Characteristic	e ^β *	95% CI*	P
Age, y	1.01	1.00 to 1.01	<0.001
Sex, female	1.01	0.94 to 1.09	0.79
Recurrent Stroke	1.03	0.94 to 1.12	0.57
Diabetes Mellitus	1.06	0.97 to 1.17	0.21
Dementia	1.28	1.12 to 1.46	<0.001
Hypercholesterolemia	0.94	0.85 to 1.05	0.27
Myocardial Infarction or Ischaemic Heart	1.00	0.92 to 1.09	0.98
Previous Cancer	1.12	0.99 to 1.27	0.08
COPD	0.90	0.77 to 1.06	0.21
Pre-stroke mRS Score (reference 0)			<0.001
1	1.06	0.95 to 1.19	0.28
2	0.90	0.77 to 1.04	0.15
3	0.94	0.80 to 1.11	0.47
4 & 5	0.71	0.59 to 0.86	<0.001
Pre-Stroke Residence (reference Independent living without formal care)			<0.001
Independent living with formal care	1.07	0.94 to 1.23	0.92
Institution	0.81	0.69 to 0.95	0.01
Haemorrhagic Stroke	0.80	0.71 to 0.90	<0.001
Oxford Community Stroke Project Classification (reference LACS)			<0.001
PACS	1.30	1.18 to 1.42	<0.001
POCS	1.34	1.18 to 1.53	<0.001
TACS	1.29	1.13 to 1.48	<0.001
No Brain Lateralisation	0.85	0.75 to 0.96	0.01
Inpatient Complication	1.70	1.56 to 1.85	<0.001
Discharge mRS Score (reference 0)			<0.001
1.15	1.01 to 1.31	0.04
1.74	1.50 to 2.04	<0.001
2.70	2.32 to 3.13	<0.001
3.51	2.98 to 4.14	<0.001
5.07	4.19 to 6.14	<0.001
1.24	1.05 to 1.48	0.01
Winter Admission	1.15	1.08 to 1.24	<0.001
Weekend Admission	1.03	0.95 to 1.11	0.50
Hospital (reference 1)			<0.001
2.09	1.38 to 3.17	0.001
1.07	0.94 to 1.22	0.29
1.08	0.90 to 1.31	0.40
0.78	0.69 to 0.87	<0.001
0.93	0.81 to 1.07	0.33
1.15	1.00 to 1.32	0.05
0.82	0.70 to 0.94	0.01

1
2
3 AHLOS, Acute Hospital Length of Stay; CI, Confidence Intervals; COPD, Chronic Obstructive Pulmonary
4 Disorder; mRS, modified Rankin Scale; LACS, Lacunar Anterior Circulation Stroke; 
[revised manuscript text omitted]

P.K.M supervised the study. D.M provided statistical analysis advice and assistance. M.O.B,
E.A.W, J.F.P and P.K.M conceptualized and designed the use of ASCNES and obtained
funding. S.D.M was ASCNES study coordinator and managed the data. All authors reviewed
the manuscript and contributed in critical revision and final manuscript preparation.

**Funding:** This work was supported by the National Institute for Health Research (NIHR)
Research for Patient Benefit Programme (PB-PG-1208-18240). EAW receives funding
support from the NIHR Biomedical Research Centre award to Cambridge. MT holds a PhD
studentship funded by the College of Life Sciences & Medicine, University Aberdeen
(CF10109-38).

**Disclaimer:** The views expressed are those of the authors and not necessarily those of the
NHS, the NIHR or the Department of Health.

**Competing interests:** The authors declare no competing interests.

**Patient consent:** Not required.

**Ethics approval:** Ethical approval was obtained from the NRES Committee East of England
– Norfolk (REC Reference number 10/H0310/44).

**Data sharing statement:** The datasets generated and analysed during the current study are
available from the ASCNES team on reasonable request.

REFERENCES

1. Feigin VL, Norrving B, Mensah GA. Global burden of stroke. *Circ Res* 2017;120:439-48. doi:10.1161/CIRCRESAHA.116.308413.
2. Krishnamurthi RV, Feigin VL, Forouzanfar MH, et al on behalf of the Global Burden of Diseases, Injuries, and Risk Factors Study 2010 (GBD 2010) and the GBD Stroke Experts Group. Global and regional burden of first-ever ischaemic and haemorrhagic stroke during 1990-2010: findings from the Global Burden of Disease Study 2010. *The Lancet Glob health*. 2013;1:e259-e281. doi:10.1016/S2214-109X(13)70089-5.
3. Mu F, Hurley D, Betts KA, et al. Real-world costs of ischemic stroke by discharge status. *Curr Med Res Opin* 2016;33:371-8. doi:10.1080/03007995.2016.1257979.
4. Ovbiagele B, Goldstein LB, Higashida RT, et al on behalf of the American Heart Association Advocacy Coordinating Committee and Stroke Council. Forecasting the future of stroke in the United States: A policy statement from the American Heart Association and American Stroke Association. *Stroke* 2010;44:2361-75. doi:10.1161/STR.0b013e31829734f2.
5. Townsend N, Bhatnagar P, Wilkins E et al. Cardiovascular disease statistics 2015. London: British Heart Foundation, 2015.
6. Marshall IJ, Wang Y, Crichton S, et al. The effects of socioeconomic status on stroke risk and outcomes. *Lancet Neurol* 2015;14:1206-18. doi:10.1016/S1474-4422(15)00200-8.
7. Bray BD, Ayis S, Campbell J, et al. Associations between stroke mortality and weekend working by stroke specialist physicians and registered nurses: prospective multicentre cohort study. *PLoS Med* 2014;11:e1001705. doi:10.1371/journal.pmed.1001705.

8. Bray BD, Ayis S, Campbell J, et al. Associations between the organisation of stroke
services, process of care, and mortality in England: prospective cohort study. *BMJ* 2013;346.
doi:10.1136/bmj.f2827.
9. Rudd AG, Jenkinson D, Grant RL, et al. Staffing levels and patient dependence in English
stroke units. *Clin Med (Lond)* 2009;9:110-5. doi:10.7861/clinmedicine.9-2-110.
10. Myint PK, Bachmann MO, Loke YK, et al. Important factors in predicting mortality
outcome from stroke: findings from the Anglia Stroke Clinical Network Evaluation Study.
*Age Ageing* 2017;46:83-90. doi:10.1093/ageing/afw175.
11. Fonarow GC, Smith EE, Reeves MJ, et al for the Get With The Guidelines Steering
Committee and Hospitals. Hospital-level variation in mortality and rehospitalization for
medicare beneficiaries with acute ischemic stroke. *Stroke* 2011;42:159-66.
doi:10.1161/STROKEAHA.110.601831.
12. Kim SM, Hwang SW, Oh E, et al. Determinants of the length of stay in stroke patients.
*Osong Public Health and Research Perspectives* 2013;4:329-41.
13. Kwok CS, Clark A, Ford GA, et al. Association between prestroke disability and inpatient
mortality and length of acute hospital stay after acute stroke. *J Am Geriatr Soc* 2012;60:726-
32. doi:10.1111/j.1532-5415.2011.03889.x.
14. Appelros P. Prediction of length of stay for stroke patients. *Acta Neurol Scand*
2007;116:15-9. doi:10.1111/j.1600-0404.2006.00756.x.

[revised manuscript text omitted]

38. Elrod JK, Fortenberry Jr JL. The hub-and-spoke organization design revisited: a lifeline
for rural hospitals. *BMC Health Serv Res* 2017;17:795. doi:10.1186/s12913-017-2755-5.
39. Matsui H, Fushimi K, Yasunaga H. Variation in risk-standardized mortality of stroke
among hospitals in Japan. *PLoS One* 2015;10:e0139216. doi:10.1371/journal.pone.0139216.
40. Tsugawa Y, Kumamaru H, Yasunaga H, et al. The association of hospital volume with
mortality and costs of care for stroke in Japan. *Med Care* 2013;51:782-8.
doi:10.1097/MLR.0b013e31829c8b70.
41. Stroke Unit Trialists' Collaboration. Organised inpatient (stroke unit) care for stroke
(Review). *Cochrane Database Syst Rev* 2013;9. doi: 10.1002/14651858.CD000197.pub3.
42. Chan DKY, Cordato D, O'Rourke F, et al. Comprehensive stroke units: a review of
comparative evidence and experience. *Int J Stroke* 2013;8:260-4. doi:10.1111/j.1747-
4949.2012.00850.x.
43. Chiu A, Shen Q, Cheuk G, et al. Establishment of a stroke unit in a district hospital:
review of experience. *Intern Med J* 2007;37:73-8. doi:10.1111/j.1445-5994.2007.01235.x
44. Tamm A, Siddiqui M, Shuaib A, et al. Impact of stroke care unit on patient outcomes in a
community hospital. *Stroke* 2014;45:211-6. doi:10.1161/STROKEAHA.113.002504.
45. Ingeman A, Andersen G, Hundborg HH, et al. In-hospital medical complications, length
of stay, and mortality among stroke unit patients. *Stroke* 2011;42:3214-8.
doi:10.1161/STROKEAHA.110.610881.

46. Tirschwell DL, Kukull WA, Longstreth WT. Medical complications of ischemic stroke
and length of hospital stay: experience in Seattle, Washington. *J Stroke Cerebrovasc Dis*
1999;8:336-43. doi:[10.1016/S1052-3057\(99\)80008-1](https://doi.org/10.1016/S1052-3057(99)80008-1).
47. Myint PK, Vowler SL, Woodhouse PR, et al. Winter excess in hospital admissions, in-
patient mortality and length of acute hospital stay in stroke: a hospital database study over six
seasonal years in Norfolk, UK. *Neuroepidemiology* 2007;28:79-85. doi:10.1159/000098550.
48. Nuyen J, Spreuwenberg PM, Groenewegen PP, et al. Impact of preexisting depression
on length of stay and discharge destination among patients hospitalized for acute stroke:
linked register-based study. *Stroke* 2008;39:132-8. doi:[10.1161/STROKEAHA.107.490565](https://doi.org/10.1161/STROKEAHA.107.490565).
49. Jia H, Damush TM, Qin H, et al. The impact of poststroke depression on healthcare use
by veterans with acute stroke. *Stroke* 2006;37:2796-801.
50. Schwamm LH, Reeves MJ, Pan W, et al. Race/ethnicity, quality of care, and outcomes in
ischemic stroke. *Circulation* 2010;121:1492-1501.
doi:10.1161/CIRCULATIONAHA.109.881490.
51. Ng YS, Tan KH, Chen C, et al. Predictors of acute, rehabilitation and total length of
stay in acute stroke: a prospective cohort study. *Ann Acad Med Singapore* 2016;45:394-403.
52. Office of National Statistics. *Dataset(s): 2011 census: Key statistics and quick statistics
for local authorities in the united kingdom - part 1. KS201UK ethnic group, local authorities
in the united kingdom*. London: ONS, 2013.
<https://www.ons.gov.uk/peoplepopulationandcommunity/populationandmigration/population>

estimates/datasets/2011censuskeystatisticsandquickstatisticsforlocalauthoritiesintheunitedkin
gdompart1 (accessed 12 Dec 2018).

For peer review only

**Figure 1** Flow chart of patient participation inclusion and exclusion for study analysis

**Figure 2** Model estimates of mean baseline acute hospital length of stay (AHLOS) per
hospital (in days) against hospital stroke volume and clustered by hospital type with 95%
confidence intervals. Multiple regression model was adjusted for patient covariates that had a
p-value<0.3 in univariable analysis.

For peer review only

Figure 1 Flow chart of patient participation inclusion and exclusion for study analysis

Figure 2 Model estimates of mean baseline acute hospital length of stay (AHLOS) per hospital (in days) against hospital stroke volume and clustered by hospital type with 95% confidence intervals. Multiple regression model was adjusted for patient covariates that had a p-value<0.3 in univariable analysis.

152x152mm (300 x 300 DPI)

STUDY PROTOCOL

Open Access

Evaluation of stroke services in Anglia stroke clinical network to examine the variation in acute services and stroke outcomes

Phyo K Myint^{1,2*}, John F Potter^{1,2}, Gill M Price¹, Garry R Barton¹, Anthony K Metcalf², Rachel Hale², Genevieve Dalton³, Stanley D Musgrave¹, Abraham George⁴, Raj Shekhar⁵, Peter Owusu-Agyei⁶, Kevin Walsh⁷, Joseph Ngeh⁸, Anne Nicholson⁹, Diana J Day¹⁰, Elizabeth A Warburton¹⁰, Max O Bachmann¹

Abstract

Background: Stroke is the third leading cause of death in developed countries and the leading cause of long-term disability worldwide. A series of national stroke audits in the UK highlighted the differences in stroke care between hospitals. The study aims to describe variation in outcomes following stroke and to identify the characteristics of services that are associated with better outcomes, after accounting for case mix differences and individual prognostic factors.

Methods/Design: We will conduct a cohort study in eight acute NHS trusts within East of England, with at least one year of follow-up after stroke. The study population will be a systematically selected representative sample of patients admitted with stroke during the study period, recruited within each hospital. We will collect individual patient data on prognostic characteristics, health care received, outcomes and costs of care and we will also record relevant characteristics of each provider organisation. The determinants of one year outcome including patient reported outcome will be assessed statistically with proportional hazards regression models. Self (or proxy) completed EuroQoL (EQ-5D) questionnaires will measure quality of life at baseline and follow-up for cost utility analyses.

Discussion: This study will provide observational data about health service factors associated with variations in patient outcomes and health care costs following hospital admission for acute stroke. This will form the basis for future RCTs by identifying promising health service interventions, assessing the feasibility of recruiting and following up trial patients, and provide evidence about frequency and variances in outcomes, and intra-cluster correlation of outcomes, for sample size calculations. The results will inform clinicians, public, service providers, commissioners and policy makers to drive further improvement in health services which will bring direct benefit to the patients.

Background

Stroke is the third leading cause of mortality and the number one cause of long-term disability in the UK. More than 150,000 people suffer a stroke in the UK each year [1]. It costs the NHS approximately £ 7 billion per annum [2]. Stroke incidence rises sharply with age and despite better primary and secondary preventative measures, the total number of strokes is set to rise in

the UK [3]. Nevertheless, stroke care in UK is far from ideal: patients having a worse outcome in terms of death and dependency than many other European countries [4-6], at least in part due to differences in care provided [7]. There is also variation in outcome between different localities within the UK [8-11], these local differences being highlighted in the most recent publication of the National Sentinel Stroke Audit in 2009 [12]. These differences probably arise as a result of substantial variations in how the stroke services are provided across the UK. Examples of such differences are access to neurovascular/neurosurgical service, early supported

* Correspondence: phyo.k.myint@uea.ac.uk

¹Norwich Medical School, Faculty of Medicine & Health Sciences, Norwich, UK

Full list of author information is available at the end of the article

discharge, and stroke specialist on call rota for thrombolysis. The presence or absence of variations in stroke outcomes as a result of variation in care and how much the observed variations in patients' outcomes including patient reported outcome measure (PROM) are determined by the differences in service delivery have not been examined previously.

We hypothesise that variation in patient outcomes including mortality, length of stay, institutionalisation rate, and patient reported outcomes between care providers can partly be explained by the different ways in which stroke services are delivered. The main objectives of the study are (1) to describe variation in outcomes following stroke and to identify the characteristics of services that are associated with better outcomes after accounting for case mix differences and individual prognostic factors, and (2) to obtain preliminary data to identify sample size and inform future pragmatic real world setting RCTs in the area of health service delivery in stroke.

25 **Methods/Design**

A prospective cohort study will be conducted to identify characteristics of services that are associated with the best outcomes including patient reported outcomes, taking into account case-mix and patients' prognostic features. The study will consist of two components (1) consecutive stroke admissions in selected months (a total of 8 months) and (2) a prospective study of patient reported outcome in some of these selected months.

**Sample Population**

For the first component, the sample population will be stroke patients who are admitted to any of the hospitals within the Anglia region of Stroke & Heart Clinical Network between October 2009 and September 2011. Baseline data are already recorded, prior to the study commencement, as part of routine clinical data collection by Anglia Stroke Clinical Network (as described in detail below). The study sample will be a systematically selected sample (every third month) rather than a consecutive cohort of patients admitted to eight acute NHS hospital trusts. Therefore, this is not a consecutive case study; instead it seeks to be representative of the catchment population of the hospital and has taken into account the seasonal variation in stroke incidence and outcome [13].

For the patient reported outcome component of the study the following inclusion and exclusion criteria will be used. Inclusion criteria are (1) age \geq 18 years, (2) admitted to hospital with stroke (diagnosed by stroke physicians) during the study months, (3) able to provide informed consent or patient's personal consultee agrees to study participation. Exclusion criteria include (1) age

<18 years, (2) patients with pre-existing diagnosis of dementia (for PROM component only).

The Anglia Stroke Network was funded through the NHS Improvement Programme, following the publication of the National Stroke Strategy in December 2007. The Network was established in April 2008 to support the development of stroke services in Norfolk, Suffolk and Cambridgeshire regions. Since its inception, the Network regularly collected data to capture clinical service activities of the eight acute hospital trusts in the Network for the purpose of monitoring of services benchmarked by National targets and guidance from National Institute of Health & Clinical Excellence (NICE) in England and Wales. Data collection commenced in January 2009 and involves the individual trusts collecting clinical data which is fed back to the network by monthly reports. The total number of strokes admitted to the 8 acute trusts within the Network is approximately 4,000 per annum in 2009. The stroke cases were identified prospectively data were collected by the clinical team who looked after the patients and anonymised raw clinical data were sent to the network on monthly basis. The network collates and analyses the data for above mentioned purposes.

**Sample size**

Since this is an exploratory study designed to provide information for further analytic research, sample size will be determined partly pragmatically rather than on particular hypothesis tests. For illustration purposes, a total sample of 2264 patients would provide 80% power to detect a constant Hazard ratio (HR) of 0.76 for one-year mortality between two groups of roughly equal size, based on the log-rank test. This assumes a 20% one-year mortality rate in the reference group, no loss to follow-up before one year and 2-sided type I error of 5%. If one-year mortality is 30%, then 2264 patients would provide 76% power to detect a HR of 0.81.

Plan of investigation

The study will have a cohort design. We will follow up a cohort of patients systematically selected from each trust. For pragmatic purposes we will sample all patients who are admitted every third month, starting from October 2009. Over one calendar month, there will be ~ 300-350 stroke cases entered into the Network Clinical Data. Between October 2009 and September 2011, the Clinical Network would have collected a total of eight 3-monthly datasets per trust (i.e. 8 study months in total: Oct 2009, Jan 2010, April 2010, July 2010, October 2010, Jan 2011, April 2011, July 2011). Therefore, the estimated total cohort size with baseline clinical data will be ~ 2,400 stroke cases

during this exercise (30% of 4000 patients admitted annually in 8 trusts = 1200 × 2 yrs).

We will collect patient data by hospital trusts and conduct a questionnaire survey of patients' outcomes. Due to the nature of the study we would need 100% follow-up in randomly selected populations. Because we will be using a partially historical cohort, to avoid selection bias for mortality outcome, informed consent from all eligible participants will not be feasible. Therefore, it is most appropriate for the clinical team to collect the outcome data to comply with current ethical guidance in the UK. Therefore, the identifiable patient data will only be held at the local NHS trusts.

Neither the network nor the investigators will have access to any identifiable patient information (e.g. name, address). For outcome data we will utilise death certificate and hospital episode data from the Patient Administrative System (PAS) as described previously [14,15]. This approach will be used in conjunction with telephone and postal follow-up for questionnaire surveys such as EQ-5 D, and Stroke Impact Scale. These data will be counter-checked using discharge coding records, which record each hospital episode.

The clinical teams will retrieve case records to collect (1) baseline measures which were not recorded in baseline Network surveys and (2) outcome measures including mortality and hospital length of stay. At study commencement (October 2010) one year follow up data can be collected immediately for October 2009 cohorts (follow up complete at end September 2010). The follow up will be completed in September 2012 as the stroke patients included in the last survey for the study conducted by the Network in July 2011 will complete one year follow-up in June 2012 and data collection of the study will be completed by July-August 2012 with the view of final cohort data arrival to research team by the end of December 2012.

Due to multi-centre nature of the study the individual sites are expected to join the study at different time points (after their respective NHS Research & Development Committees' approval). We will collect characteristics of stroke services, patient related factors, prognostic indicators, treatment options and trial/study participation. Missing prognostic data will be imputed statistically, to ensure that all eligible patients are included in the primary analysis (see also Statistical Methods).

The service characteristics of interest include:

At hospital level

- staffing (including junior doctors and therapists (whole time equivalent), physicians characteristics
 - university or district general hospital
 - distance from tertiary referral centre

- availability of vascular surgery on site, neuro-surgery and neuro ITU on site
- monitoring beds
- physician on call rota
- compliance with NICE guidelines

At patient level

- provision of thrombolysis and CT
- medication

Outcome measurements

Primary outcome of the study will be one year mortality comparison between services with different characteristics. The secondary outcomes will include (1) final discharge destination (good or poor outcome) [16], (2) length of acute hospital stay, (3) length of stay in rehabilitation, (4) complications during acute and rehab-hospital stay and significant procedures (e.g. aspiration pneumonia, myocardial infarction), (5) readmissions, (6) composite cardiovascular events (recurrent TIA/Stroke/Acute Coronary Syndrome, Myocardial infarction).

Patient Reported Outcome Measures (PROM)

PROM will consist of (1) Stroke Impact Scale, (2) health related quality of life: EQ-5 D at one year in those who completed questionnaire at the baseline, (3) modified RANKIN, (4) Barthel score and (5) health service use.

Statistical analysis

Quantitative data will be analysed by multivariate Cox-proportional hazards to examine the relationships between different aspects of health services and time to death, adjusting for prognostic characteristics. Multiple logistic or linear regression models will be constructed as appropriate for dichotomised and continuous outcome variables respectively. T tests for normally distributed data and Mann-Whitney U tests for non-normally distributed data will be used to compare continuous outcomes. Volume-outcome relationships will be investigated. Missing prognostic and EQ-5 D data will be imputed, based on each patient's other prognostic characteristics. Clustering of data by hospital trust will be investigated and, if necessary, taken into account, and intra-class correlation coefficients calculated to inform future research.

Economic evaluation

Health care resources are scarce and it is therefore important to ensure that evaluations are undertaken in order to ensure that services provided by the NHS constitute value for money. Within this study we will thereby seek to estimate the cost-effectiveness of different stroke service deliveries.

Costs will first be calculated from the perspective of the NHS and personal social services (PSS). Thus, levels

of resources use will be recorded during the follow-up period, including the length of original hospital stay, input by the multi-disciplinary team, other investigations (e.g. x-ray) and any complications (including details of any further hospital admissions). Unit costs will subsequently be assigned to each of these resource items, enabling both the total mean cost in participants and the incremental cost between two different service deliveries (chosen to compare the cost effectiveness, e.g. traditional on call rota vs. telemedicine) to be calculated after adjusting for other factors. The main measure of effectiveness to be used in the economic analysis will be the EQ-5 D [17], where responses will be sought at baseline, and at 12 month as mentioned above. This will enable the overall effect of each mode of service delivery, and the incremental effect of services to be estimated.

Outcome

As the National Institute of Health and Clinical Excellence [18] recommends use of the EQ-5 D [17] within cost-effectiveness analysis this will be our primary measure within the economic analyses. EQ-5 D data will be collected at two University Hospitals and two district general hospitals within the clinical network. We will use “mapping” strategy to estimate the cost-effectiveness analyses across the region. The use of mapping, where scores from a condition-specific (non preference-based) measure are ‘converted’ into a utility (preference-based) score using a pre-defined formulae, has been advocated (in certain instances) by the UK National Institute of Health and Clinical Excellence (NICE) [18], and has been used to estimate the utility scores, and in turn cost-effectiveness, of a number of health care interventions [19]. Mapping presents the possibility of not asking all participants to complete the EQ-5 D. In this study we propose to take advantage of this by developing a mapping algorithm based on the response from participants participating in this component to predict the EQ-5 D for participants in retrospective cohorts and those who did not participate in PROM component.

Because the quality of life measure (EQ-5D) which can be used to estimate health utility and calculate QALYs (Quality Adjusted Life Years) for economic evaluation is outside the remit of routine data collection and cannot be done retrospectively, we will collect EQ-5 D data in only the second year of the study (October 2010 and January, April and July 2011 cohorts and one year follow up data to be collected September and December 2011, and March and June 2012) in those who provide informed consent to the study (we estimate that the sample will be approximately 15-20% of the whole sample after excluding the one year pre-study period (between October 2009-September 2010) and after

taking into account of refusal rate (estimated ~ 30%) in trusts with Stroke or Comprehensive Local Research Network Research Nurses.

Economic Analysis

In the Economic analysis if one option is shown to be less costly and more effective than another option (for example, telemedicine vs. on call system) then that option will ‘dominate’ the other and be deemed cost-effective. Alternatively, the incremental cost-effectiveness ratio (ICER) associated with a particular option will be estimated and assessed in relation to a range of cost-effectiveness thresholds. The associated level of uncertainty will also be characterised by e.g. estimating the cost-effectiveness acceptability curve (CEAC) for each intervention and conducting value of information analysis [20]. Sensitivity analysis will also be undertaken to assess the robustness of conclusions to key assumptions. We will also seek to identify what resource items should be monitored in a future study (i.e. what are the big cost drivers which are likely to be affected by the intervention) and how these items should be identified.

The study is funded by the NIHR Research for Patient Benefit Programme (PB-PG-1208-18240) and obtained ethical approval from the Norfolk Research Ethics Committee.

Discussion

In this study we specifically aim to identify services that are associated with the best clinical outcomes including mortality and hospital length of stay including patient reported outcome adjusting for patient prognostic factors and potential confounders. Our study will be able to provide useful information in stroke service provision in UK and beyond. Furthermore, inclusion of patient reported outcome is novel and exciting component of our study.

Studies which have examined the delivery of specific services such as rapid imaging, have shown improvement in patients’ outcome in stroke [21]. A recent report from Germany suggested that a telestroke network may be a useful strategy to implement in their non-urban stroke services [22]. Lees *et al* (2008) [23] highlighted that there is room for improvement in terms of acute services for stroke. Interestingly, one of the observations was that centres with higher workload performed better. There is also existing evidence in Cancer literature that centres with higher surgical caseload have better outcomes [24]. There has also been a recent evaluation of the impact on stroke outcome by evidence-based practice in an Australian setting [25]. Examples of service delivery that are associated with better outcomes include organised stroke unit care [26], thrombolysis treatment and appropriate secondary prevention [27], and early supported discharge

in selected patients [28,29]. However, the cost-effectiveness of such services has yet to be fully examined.

Rodgers *et al* [30] highlighted the need for improvement in hospital-based stroke services e.g. stroke unit staffing levels were lower than was available in RCTs. The accumulating body of evidence has been a major driving force behind the UK Government's strategy to improve stroke care (National Stroke Strategy, 2007) [31]. A key strand of the strategy was to set up stroke networks to deliver stroke service development across geographically defined areas. The stroke networks have worked to agree minimum standards for stroke care and they have worked with commissioners to assist the commissioning process for stroke services. The acute stroke services are currently delivered by different NHS trusts and there is therefore a wide range of inequality in service availability and provision with differing structure and local support systems.

This research aims to utilise NHS data in the most meaningful and innovative way and we aim to maximize the benefit with minimum investment to produce best research output for patient care by collaborating with clinical teams and the network in providing excellent value for money. This observational study seeks to identify areas of clinical practice which merit future randomised controlled trials (RCTs) to identify best practice in improving stroke care which will be of maximum benefit to patients. We also aim to obtain preliminary data to estimate sample sizes and conduct value of information analyses to design future pragmatic RCTs of innovative ways of delivering stroke care.

As we include eight diverse NHS trusts, the findings are likely to be generalisable in the UK setting and beyond. This study will provide observational data about health service factors associated with variations in patient outcomes and health care costs following hospital admission for acute stroke. This will form the basis for future RCTs by identifying promising health service interventions, assessing the feasibility of recruiting and following up trial patients, and provide evidence about frequency and variances in outcomes, and intra-cluster correlation of outcomes, for sample size calculations. The results will also inform clinicians, public, service providers, commissioners and policy makers to drive further improvement in health services and bring direct benefit to patients.

The study will describe the variation in outcomes between different stroke services, and identify the characteristics of services associated with better outcomes after accounting for case-mix. We will also estimate the relative costs of and health gain estimated as Quality Adjusted Life Year (QALY) gain that may be demonstrated by different services. The commissioners of services will be informed as to which service delivery

structures are likely to provide value for money to make purchasing decisions. They will also be better informed about the types of service associated with better patient reported outcome. Hospital trusts will be able to evaluate their services systematically and plan their care appropriately to meet local and regional needs and demands based on our study findings. Professionals will be able to reflect on the impact of services they are delivering to help improve their performance and the way services are organised by adopting the most effective and cost effective approaches. As an observational study, the study limitations include inability to control for unknown confounders and residual confounding effect of known confounders which are adjusted for. The causal relationship cannot be implied but as we stated the findings will provide knowledge about areas that requires further evaluation in clinical trial setting.

There is very little work which assesses service provision robustly against patients' own reported outcomes. This exciting study may lead to a clearer drive for patients to define what makes a good service. We hope that the best clinical practices are adopted to suit the local populations' needs and demand. As we included eight diverse NHS trusts, the findings will be generalisable in the UK setting and likely to be applicable in international setting. All these will become drivers of improvement in stroke services for the benefit of stroke sufferers.

Acknowledgements

We would like to thank the participants of the study. We gratefully acknowledge the contribution of Stroke Research Nurses and the study steering committee members including representatives from the Regional Stroke Association, and Patient and Public Involvement in Research Panel. We would like to thank our colleagues from all participating trusts, site data co-ordinator and staff from Anglia Stroke & Heart Network for their assistance. Norfolk and Norwich University Hospital NHS Foundation Trust sponsors the study.

Manuscripts that are under submission based on this protocol

None.

Author details

¹Norwich Medical School, Faculty of Medicine & Health Sciences, Norwich, UK. ²Norfolk & Norwich University Hospital, Norwich, UK. ³Anglia Stroke & Heart Clinical Network, Cambridge, UK. ⁴James Paget University Hospital, Lowestoft, UK. ⁵Queen Elizabeth Hospital, King's Lynn, UK. ⁶Peterborough City Hospital, Peterborough, UK. ⁷Hinchingbrooke Hospital, Huntingdon, UK. ⁸Ipswich Hospital, Ipswich, UK. ⁹West Suffolk Hospital, Bury St Edmund, UK. ¹⁰Addenbrooke's University Hospital, Cambridge, UK.

Authors' contributions

PKM, DJD, MOB designed the outline of the study. PKM, JFP, MOB, EAW, GMP, GAB and AKM obtained the funding for the study. SDM & RH contributed in protocol preparation. All authors contributed in writing of the paper. All authors read and approved the final manuscript. PKM is the guarantor.

Competing interests

The authors declare that they have no competing interests.

Received: 21 January 2011 Accepted: 28 February 2011

Published: 28 February 2011

References

- National Institute for Health and Clinical Excellence: **STROKE: National clinical guideline for diagnosis and initial management of acute stroke and transient ischaemic attack (TIA). The National Collaborative Centre for Chronic Conditions.** *Royal College of Physicians of London* 2008 [<http://www.nice.org.uk/nicemedia/pdf/CG68FullGuideline.pdf>], ISBN 978-1-86016-339-5.
- National Audit Office: **Reducing brain damage: faster access to better stroke care.** *London: Stationery Office, 2005* National Audit Office Report; 2005.
- Rothwell PM, Coull AJ, Silver LE, Fairhead JF, Giles MF, Lovelock CE, Redgrave JN, Bull LM, Welch SJ, Cuthbertson FC, Binney LE, Gutnikov SA, Anslow P, Banning AP, Mant D, Mehta Z, Oxford Vascular Study: **Population-based study of event-rate, incidence, case fatality and mortality for all acute vascular events in all arterial territories (Oxford Vascular Study).** *Lancet* 2005, **366**:1773-83.
- Grieve R, Hutton J, Bhalla A, Rastenyte D, Ryglewicz D, Sarti C, Lamassa M, Giroud M, Dundas R, Wolfe CD: **A comparison of the costs and survival of hospital-admitted stroke patients across Europe.** *Stroke* 2001, **32**:1684-91.
- Weir NU, Sandercock PA, Lewis SC, Signorini DF, Warlow CP: **Variations between countries in outcome after stroke in the International Stroke Trial (IST).** *Stroke* 2001, **32**:1370-7.
- Gray LJ, Sprigg N, Bath PM, Sørensen P, Lindenstrøm E, Boysen G, De Deyn PP, Friis P, Leys D, Marttila R, Olsson JE, O'Neill D, Ringelstein B, van der Sande JJ, Turpie AG, TAIST Investigators: **Significant variation in mortality and functional outcome after acute ischaemic stroke between Western countries: data from the tinzaparin in acute ischaemic stroke trial (TAIST).** *J Neurol Neurosurg Psychiatry* 2006, **77**:327-33.
- Markus H: **Improving the outcome of stroke.** *BMJ* 2007, **335**:359-60.
- Rudd AG, Irwin P, Rutledge Z, Lowe D, Wade DT, Pearson M: **Regional variations in stroke care in England, Wales and Northern Ireland: results from the National Sentinel Audit of Stroke.** *Royal College of Physicians Intercollegiate Stroke Working Party.* *Clin Rehabil* 2001, **15**:562-72.
- Rudd AG, Lowe D, Hoffman A, Irwin P, Pearson M: **Secondary prevention for stroke in the United Kingdom: results from the National Sentinel Audit of Stroke.** *Age Ageing* 2004, **33**:280-6.
- Rudd AG, Hoffman A, Irwin P, Pearson M, Lowe D, Intercollegiate Working Party for Stroke: **Stroke units: research and reality. Results from the National Sentinel Audit of Stroke.** *Qual Saf Health Care* 2005, **14**:7-12.
- Howell E, Graham C, Hoffman A, Lowe D, McKeivitt C, Reeves R, Rudd AG: **Comparison of patients' assessments of the quality of stroke care with audit findings.** *Qual Saf Health Care* 2007, **16**:450-5.
- National Sentinel Stroke Audit: **Phase II (clinical audit), 2008. Report for England, Wales and Northern Ireland.** *Intercollegiate Stroke Working Party.* *Clinical Effectiveness and Evaluation Unit, Royal College of Physicians of London* 2009.
- Myint PK, Vowler SL, Woodhouse PR, Redmayne O, Fulcher RA: **Winter excess in hospital admissions, in-patient mortality and length of acute hospital stay in stroke: a hospital database study over six seasonal years in Norfolk, UK.** *Neuroepidemiology* 2007, **28**:79-85.
- Myint PK, Kamath AV, Vowler SL, Maisey DN, Harrison BD: **The CURB (confusion, urea, respiratory rate and blood pressure) criteria in community-acquired pneumonia (CAP) in hospitalised elderly patients aged 65 years and over: a prospective observational cohort study.** *Age Ageing* 2005, **34**:75-7.
- Myint PK, Vowler SL, Redmayne O, Fulcher RA: **Utilisation of diagnostic computerised tomography imaging and immediate clinical outcomes in older people with stroke before and after introduction of the National Service Framework for older people. A comparative study of hospital-based stroke registry data (1997-2003): Norfolk experience.** *Age Ageing* 2006, **35**:399-403.
- Myint PK, Vowler SL, Redmayne O, Fulcher RA: **Cognition, continence and transfer status at the time of discharge from an acute hospital setting and their associations with an unfavourable discharge outcome after stroke.** *Gerontology* 2008, **54**:202-9.
- Brooks R: **EuroQol: the current state of play.** *Health Policy* 1996, **37**:53-72.
- National Institute for Clinical Excellence: 2004 [http://www.nice.org.uk/aboutnice/howwework/devicetech/technologyappraisalprocessguides/guide_to_the_methods_of_technology_appraisal_reference_n0515.jsp], (reference N0515). ISBN: 1-84257-595-3.
- Barton GR, Sach TH, Jenkinson C, Avery AJ, Doherty M, Muir KR: **Do estimates of cost-utility based on the EQ-5 D differ from those based on the mapping of utility scores?** *Health Qual Life Outcomes* 2008, **6**:51.
- Barton GR, Briggs AH, Fenwick EA: **Optimal cost-effective decisions: the role of the cost-effectiveness acceptability curve (CEAC), cost-effectiveness acceptability frontier (CEAF) and expected value of perfect information (EVPI).** *Value Health* 2008, **11**:886-897.
- Wardlaw JM, Seymour J, Cairns J, Keir S, Lewis S, Sandercock P: **Immediate computed tomography scanning of acute stroke is cost-effective and improves quality of life.** *Stroke* 2004, **35**:2477-83.
- Audebert HJ, Kukla C, Vatankeh B, Gotzler B, Schenkel J, Hofer S, Fürst A, Haberl RL: **Comparison of tissue plasminogen activator administration management between Telestroke Network hospitals and academic stroke centers: the Telemedical Pilot Project for Integrative Stroke Care in Bavaria/Germany.** *Stroke* 2006, **37**:1822-7.
- Lees KR, Ford GA, Muir KW, Ahmed N, Dyker AG, Atula S, Kalra L, Warburton EA, Baron JC, Jenkinson DF, Wahlgren NG, Walters MR, SITS-UK Group: **Thrombolytic therapy for acute stroke in the United Kingdom: experience from the safe implementation of thrombolysis in stroke (SITS) register.** *QJM* 2008, **101**:863-9.
- Bachmann MO, Alderson D, Edwards D, Wotton S, Bedford C, Peters TJ, Harvey IM: **Cohort study in South and West England of the influence of specialization on the management and outcome of patients with oesophageal and gastric cancers.** *Br J Surg* 2002, **89**:914-22.
- Gattellari M, Worthington J, Jalaludin B, Mohsin M: **Stroke unit care in a real-life setting: can results from randomized controlled trials be translated into every-day clinical practice? An observational study of hospital data in a large Australian population.** *Stroke* 2009, **40**:10-7.
- Stroke Unit Trialists' Collaboration: **Organised inpatient (stroke unit) care for stroke.** *Cochrane Database Syst Rev* 2007, **4**: CD000197, Review.
- Higgins P, Lees KR: **Advances in emerging therapies.** *Stroke* 2009, **40**(5): e292-4.
- Langhorne P, Taylor G, Murray G, Dennis M, Anderson C, Bautz-Holter E, Dey P, Indredavik B, Mayo N, Power M, Rodgers H, Ronning OM, Rudd A, Suwanwela N, Widen-Holmqvist L, Wolfe C: **Early supported discharge services for stroke patients: a meta-analysis of individual patients' data.** *Lancet* 2005, **365**:501-6.
- Early Supported Discharge Trialists: **Services for reducing duration of hospital care for acute stroke patients.** *Cochrane Database Syst Rev* 2005, **2**: CD000443.
- Rodgers H, Dennis M, Cohen D, Rudd A, British Association of Stroke Physicians: **British Association of Stroke Physicians: benchmarking survey of stroke services.** *Age Ageing* 2003, **32**:211-7.
- National Stroke Strategy: **Department of Health. 2007** [http://www.dh.gov.uk/en/Publicationsandstatistics/Publications/PublicationsPolicyandGuidance/DH_081062].

Pre-publication history

The pre-publication history for this paper can be accessed here:
<http://www.biomedcentral.com/1472-6963/11/50/prepub>

doi:10.1186/1472-6963-11-50

Cite this article as: Myint *et al.*: Evaluation of stroke services in Anglia stroke clinical network to examine the variation in acute services and stroke outcomes. *BMC Health Services Research* 2011 **11**:50.

Submit your next manuscript to BioMed Central and take full advantage of:

- Convenient online submission
- Thorough peer review
- No space constraints or color figure charges
- Immediate publication on acceptance
- Inclusion in PubMed, CAS, Scopus and Google Scholar
- Research which is freely available for redistribution

Submit your manuscript at
www.biomedcentral.com/submit

Supplementary document to:

**Does service heterogeneity have an impact on acute hospital length of stay in stroke? A UK-based**
**multi-centre prospective cohort study**

Michelle Tørnes, David McLernon, Max O Bachmann, Stanley D Musgrave, Elizabeth A
Warburton, John F Potter, Phyo Kyaw Myint: On behalf of the Anglia Stroke Clinical
Network Evaluation Study (ASCNES) Group

For peer review only

CONTENT

Table S1 Variables used to inform multiple imputation of missing data

Table S2 Sample characteristics distribution of the 2333 patients included in analysis per individual hospital (n (%)) unless otherwise stated)

Table S3 Sample characteristics of complete cases and those with at least one variable missing

Table S4 Univariable regression analysis for multiple imputed dataset for AHLOS (n=2233)

Table S5 Univariable linear regression complete case analysis for AHLOS

Table S6 Multiple linear regression complete case analysis for AHLOS (n=1496, $R^2=44.7\%$)

Table S7 Multiple linear regression sensitivity analysis for AHLOS, excluding Hospital 2 using multiple imputed dataset (n=2217, $R^2=44.7\%$)

Table S8 Multiple linear regression sensitivity analysis for AHLOS, including discharge destination using multiple imputed dataset (n=2233, $R^2=40\%$)

Figure S1 Model estimates of mean baseline acute hospital length of stay (AHLOS) per hospital (in days) and number of stroke patients treated outside the stroke unit per day per five stroke unit beds with 95% confidence intervals. Multiple regression model was adjusted for patient covariates that had a p-value<0.3 in univariable analysis.

Figure S2 Model estimates of mean baseline acute hospital length of stay (AHLOS) per hospital (in days) and presence of vascular surgery onsite with 95% confidence intervals. Multiple regression model was adjusted for patient covariates that had a p-value<0.3 in univariable analysis.

**Figure S3** Model estimates of mean baseline acute hospital length of stay (AHLOS) per
hospital (in days) and distance to neurosurgical facility with 95% confidence intervals.
Multiple regression model was adjusted for patient covariates that had a p-value<0.3 in
univariable analysis.

**Figure S4** Model estimates of mean baseline acute hospital length of stay (AHLOS) per
hospital (in days) and number of fte senior doctors per five beds available during weekdays
with 95% confidence intervals. Multiple regression model was adjusted for patient covariates
that had a p-value<0.3 in univariable analysis.

**Figure S5** Model estimates of mean baseline acute hospital length of stay (AHLOS) per
hospital (in days) and number of fte junior doctors per five beds available during weekdays
with 95% confidence intervals. Multiple regression model was adjusted for patient covariates
that had a p-value<0.3 in univariable analysis.

**Figure S6** Model estimates of mean baseline acute hospital length of stay (AHLOS) per
hospital (in days) and number of fte health care associates and nurses per five beds with 95%
confidence intervals. Multiple regression model was adjusted for patient covariates that had a
p-value<0.3 in univariable analysis.

**Figure S7** Model estimates of mean baseline acute hospital length of stay (AHLOS) per
hospital (in days) and number of fte occupational therapists per five beds with 95%
confidence intervals. Multiple regression model was adjusted for patient covariates that had a
p-value<0.3 in univariable analysis.

**Figure S8** Model estimates of mean baseline acute hospital length of stay (AHLOS) per
hospital (in days) and number of fte physiotherapists per five beds with 95% confidence
intervals. Multiple regression model was adjusted for patient covariates that had a p-
value<0.3 in univariable analysis.

**Figure S9** Model estimates of mean baseline acute hospital length of stay (AHLOS) per
hospital (in days) and number of fte speech and language therapists per five beds with 95%
confidence intervals. Multiple regression model was adjusted for patient covariates that had a
p-value<0.3 in univariable analysis.

**Figure S10** Model estimates of mean baseline acute hospital length of stay (AHLOS) per
hospital (in days) and number of total beds present on stroke unit per 100 admissions with
95% confidence intervals. Multiple regression model was adjusted for patient covariates that
had a p-value<0.3 in univariable analysis.

**Figure S11** Model estimates of mean baseline acute hospital length of stay (AHLOS) per
hospital (in days) and number of hospital beds per computed tomography (CT) scanner with
95% confidence intervals. Multiple regression model was adjusted for patient covariates that
had a p-value<0.3 in univariable analysis.

**Figure S12** Model estimates of mean baseline acute hospital length of stay (AHLOS) per
hospital (in days) and provision of onsite rehabilitation service with 95% confidence
intervals. Multiple regression model was adjusted for patient covariates that had a p-
value<0.3 in univariable analysis.

**Figure S13** Model estimates of mean baseline acute hospital length of stay (AHLOS) per
hospital (in days) and presence of early supported discharge scheme with 95% confidence
intervals. Multiple regression model was adjusted for patient covariates that had a p-
value<0.3 in univariable analysis.

**Figure S14** Model estimates of mean baseline acute hospital length of stay (AHLOS) per
hospital (in days) and number of non-stroke patients present on the stroke unit per day per
five stroke unit beds with 95% confidence intervals. Multiple regression model was adjusted
for patient covariates that had a p-value<0.3 in univariable analysis.

**Figure S15** Model estimates of mean baseline acute hospital length of stay (AHLOS) per
hospital (in days) and mean Index of Multiple Deprivation (IMD) score of the counties in
which the hospital serves with 95% confidence intervals. Multiple regression model was
adjusted for patient covariates that had a p-value<0.3 in univariable analysis.

For peer review only

Table S1 Variables used to inform multiple imputation of missing data

Variable	Measure
I. Independent Variables	
Trust	0=Trust 1 1=Trust 2 2=Trust 3 3=Trust 4 4=Trust 5 4=Trust 6 5=Trust 7 6=Trust 8
Sex	0=Male 1=Female
Age	Continuous, years
Recurrent Stroke	0=No 1=Yes
Diabetes Mellitus	0=No 1=Yes
Dementia	0=No 1=Yes
Hypercholesterolemia	0=No 1=Yes
Myocardial Infarction or Ischaemic Heart Disease	0=No 1=Yes
Transient Ischaemic Attack	0=No 1=Yes
Previous Cancer	0=No 1=Yes
Active Cancer	0=No 1=Yes
Depression	0=No 1=Yes
Rheumatoid Arthritis	0=No 1=Yes
Chronic Obstructive Pulmonary Disease	0=No 1=Yes
Pre-Stroke modified Rankin Score (mRS)	0=0 1=1 2=2 3=3 4=4 & 5
Pre-Stroke Residence	0=Independent living without formal care 1=Independent living with formal care 2=Institutional care
Stroke Type	0=Ischaemic 1=Haemorrhagic
Oxfordshire Community Stroke Classification	0=LACS 1=PACS 2=POCS 3=TACS
Brain Lateralisation	0=Yes 1=No
Inpatient Complication	0=No 1=Yes
Discharge modified Rankin Score (mRS)	0=0 1=1 2=2 3=3 4=4 5=5 6=6
Season of Admission	0=Summer 1=Winter
Day of Admission	0=Weekday 1=Weekend
II. Dependent Variable	
Logarithmic acute hospital LOS	Continuous, days
III. Auxiliary Variables	
Discharge Destination	0=Independent living without formal care 1=Independent living with formal care 2=Institutional care 3=Interim or rehabilitation setting 4=Death
Atrial Fibrillation	0=No 1=Yes
Baseline Systolic Blood Pressure	Continuous, mmHg
Baseline Diastolic Blood Pressure	Continuous, mmHg
Glucose Concentration on Admission	Continuous, mmol/L
Weight	Continuous, kg
Heart Rate	Continuous, beats per minute
Temperature	Continuous, °C
Oxygen Saturation	Continuous, %
ITU or HDU admission	0. No 1. Yes
Systolic Blood Pressure at Discharge	Continuous, mmHg
Diastolic Blood Pressure at Discharge	Continuous, mm Hg

LOS, Length of Stay; ITU, Intensive Care Unit; HDU, High Dependency Unit.

Table S2 Sample characteristics of the 2333 patients included in analysis per individual hospital (n (%)) unless otherwise stated)

Variables	Hospital 1	Hospital 2	Hospital 3	Hospital 4	Hospital 5	Hospital 6	Hospital 7	Hospital 8
	350 (16)	16 (1)	350 (16)	143 (6)	618 (28)	281 (13)	252 (11)	223 (10)
Age, y, median (IQR)	78 (68 to 85)	87 (81 to 92)	79 (72 to 86)	79 (70 to 86)	79 (71 to 85)	78 (71 to 85)	80 (68 to 85)	80 (71 to 87)
Sex, female	180 (52)	9 (56)	197 (56)	76 (53)	309 (50)	155 (55)	116 (46)	123 (55)
Recurrent Stroke	50 (14)	5 (31)	61 (17)	19 (17)	143 (23)	62 (22)	66 (26)	42 (19)
Diabetes Mellitus	48 (14)	1 (6)	59 (17)	17 (15)	92 (15)	66 (23)	44 (17)	43 (19)
Dementia	26 (7)	1 (6)	35 (10)	10 (9)	58 (9)	29 (10)	23 (9)	25 (11)
Hypercholesterolemia	48 (14)	3 (19)	24 (7)	7 (6)	61 (10)	80 (28)	38 (15)	94 (42)
Hypertensive	225 (64)	8 (50)	202 (58)	56 (50)	446 (72)	200 (71)	187 (74)	159 (71)
Myocardial Infarction or Ischaemic Heart Disease	45 (13)	3 (19)	87 (25)	30 (27)	142 (23)	80 (28)	49 (19)	81 (36)
Transient Ischaemic Attack	32 (9)	3 (19)	58 (17)	17 (15)	113 (18)	40 (14)	47 (19)	30 (13)
Previous Cancer	33 (9)	1 (6)	38 (11)	12 (11)	41 (7)	18 (6)	21 (8)	31 (14)
Active Cancer	24 (7)	2 (12)	8 (2)	10 (9)	49 (8)	9 (3)	20 (8)	15 (7)
Depression	13 (4)	0 (0)	17 (5)	8 (7)	33 (5)	11 (4)	18 (7)	17 (8)
Rheumatoid Arthritis	11 (3)	1 (6)	43 (12)	3 (3)	83 (13)	2 (1)	7 (3)	4 (2)
COPD	15 (4)	1 (6)	20 (6)	6 (5)	26 (4)	20 (7)	11 (4)	17 (8)
Pre-stroke mRS Score								
0	84 (43)	3 (19)	117 (36)	-	330 (56)	126 (64)	136 (56)	118 (53)
1	60(31)	3 (19)	75 (23)	-	87 (15)	16 (8)	61 (25)	33 (15)
2	24 (12)	3 (19)	51 (16)	-	56 (9)	17 (9)	16 (7)	24 (11)
3	21 (11)	2 (12)	38 (12)	-	60 (10)	20 (10)	15 (6)	28 (13)
4 & 5	7 (4)	5 (31)	44 (14)	-	57 (10)	18 (9)	16 (7)	20 (9)
Pre-Stroke Residence								
Independent living with formal care	21 (6)	4 (25)	23 (7)	15 (14)	62 (10)	30 (11)	34 (13)	21 (10)
Independent living w/o formal care	292 (86)	9 (56)	285 (82)	86 (77)	493 (80)	215 (77)	193 (77)	179 (82)
Institution	28 (8)	3 (19)	40 (11)	10 (9)	63 (10)	35 (12)	23 (9)	18 (8)

Variables	Hospital 1	Hospital 2	Hospital 3	Hospital 4	Hospital 5	Hospital 6	Hospital 7	Hospital 8
	350 (16)	16 (1)	350 (16)	143 (6)	618 (28)	281 (13)	252 (11)	223 (10)
Stroke Type								
Ischaemic	293 (85)	14 (100)	286 (87)	90 (91)	541 (88)	233 (85)	213 (87)	194 (88)
Haemorrhagic	50 (15)	0 (0)	43 (13)	9 (9)	73 (12)	40 (15)	32 (13)	26 (12)
Oxford Community Stroke Project Classification								
LACS	64 (24)	1 (7)	95 (29)	20 (28)	149 (25)	51 (18)	39 (19)	84 (39)
PACS	117 (43)	11 (79)	109 (33)	38 (54)	216 (37)	147 (53)	80 (39)	66 (30)
POCS	51 (19)	-	29 (9)	3 (4)	117 (20)	21 (8)	33 (16)	25 (12)
TACS	38 (14)	2 (14)	99 (30)	10 (14)	107 (18)	57 (21)	52 (25)	42 (19)
No Brain Lateralisation	50 (15)	2 (13)	14 (4)	9 (9)	129 (21)	1 (0.4)	30 (12)	9 (4)
Inpatient Complication	108 (31)	4 (25)	34 (10)	36 (25)	229 (37)	109 (39)	83 (33)	52 (23)
Discharge mRS Score								
0	37 (15)	0 (0)	11 (3)	0 (0)	114 (19)	34 (16)	42 (17)	22 (10)
1	65 (25)	2 (12)	55 (17)	0 (0)	97 (16)	25 (12)	55 (23)	53 (24)
2	36 (14)	1 (6)	46 (14)	0 (0)	57 (10)	20 (9)	33 (14)	19 (9)
3	41 (16)	4 (25)	40 (12)	0 (0)	87 (15)	36 (17)	34 (14)	49 (22)
4	19 (7)	3 (19)	57 (17)	0 (0)	89 (15)	25 (12)	16 (7)	29 (13)
5	4 (2)	1 (6)	47 (14)	0 (0)	40 (7)	14 (7)	16 (7)	15 (7)
6	53 (21)	5 (31)	77 (23)	29 (100)	110 (19)	58 (27)	47 (19)	35 (16)
Winter Admission	172 (49)	16 (100)	181 (52)	73 (51)	332 (54)	140 (50)	131 (52)	114 (51)
Weekend Admission	113 (32)	3 (19)	98 (28)	43 (30)	177 (29)	74 (26)	55 (22)	51 (23)

IQR, Interquartile Range; COPD, Chronic Obstructive Pulmonary Disease; mRS, modified Rankin Scale; LACS, Lacunar Anterior Circulation Stroke; PACS, Partial Anterior Circulation Stroke; POCS, Posterior Circulation Stroke; TACS, Total Anterior Circulation Stroke.

Table S3 Sample characteristics of complete cases and those with at least one variable missing

Patient Characteristic	Complete Cases (n=1496)	Cases with at least one missing variable	P
	Median (IQR) or No. (%)		
Age, y*	79 (71 to 86)	79 (70 to 86)	0.34
Sex, female†	781 (52)	384 (52)	1
Comorbidities†			
Recurrent Stroke	328 (22)	120 (17)	0.01
Diabetes Mellitus	259 (17)	111 (16)	0.38
Dementia	138 (9)	69 (10)	0.75
Hypercholesterolemia	264 (18)	91 (13)	0.01
Hypertensive	1054 (70)	429 (61)	<0.001
Myocardial Infarction or Ischaemic Heart Disease	362(24)	155 (22)	0.26
TIA	248 (17)	92 (13)	0.04
Previous Cancer	140 (9)	55 (8)	0.25
Active Cancer	93 (6)	44 (6)	1
Depression	79 (5)	38 (5)	1
Rheumatoid Arthritis	129 (9)	25 (3)	<0.001
COPD	76 (5)	40 (6)	0.64
Pre-stroke mRS Score‡			0.62
0	765 (51)	149 (51)	
1	284 (19)	51 (17)	
2	167 (11)	24 (8)	
3	149 (10)	35 (12)	
4 & 5	131 (9)	36 (12)	
Pre-stroke Residence†			<0.001
Independent living with formal care	145 (10)	65 (9)	
Independent living without formal Institution	1215 (81)	537 (78)	
(9)	84 (12)
Haemorrhagic Stroke†	138 (9)	135 (21)	<0.001
Oxford Community Stroke Project‡			0.05
LACS	411 (27)	92 (19)	
PACS	570 (38)	214 (45)	
POCS	214 (14)	65 (14)	
TACS	301 (20)	106 (22)	
No Brain Lateralisation†	174 (12)	70 (12)	0.74
Inpatient Complication†	421 (28)	234 (32)	0.09
Discharge mRS Score‡			0.02
0	218 (15)	42 (10)	
1	295 (20)	57 (14)	
2	177 (12)	35 (9)	
3	243 (16)	48 (12)	
4	209 (14)	29 (7)	
5	121 (8)	16 (4)	
6	233 (16)	181 (44)	

Winter Admission†	770 (51)	389 (53)	0.59
Weekend Admission‡	401 (27)	213 (29)	0.32

IQR, Interquartile Range; TIA, Transient Ischaemic Attack; COPD, Chronic Obstructive Pulmonary Disease; mRS, modified Rankin Scale; LACS, Lacunar Anterior Circulation Stroke; PACS, Partial Anterior Circulation Stroke; POCS, Posterior Circulation Stroke; TACS, Total Anterior Circulation Stroke.

*Two sample *t*-test

† χ^2 test

‡ χ^2 test for trend

For peer review only

Table S4 Univariable regression analysis for multiple imputed dataset for AHLOS (n=2233)

Patient Characteristic	β	95% CI	P	R ²
Age, y	1.02	1.02 to 1.02	<0.001	4.8
Sex, female	1.20	1.10 to 1.31	<0.001	0.7
Recurrent Stroke	1.17	1.05 to 1.31	0.01	0.4
Diabetes Mellitus	1.16	1.03 to 1.31	0.02	0.3
Dementia	1.46	1.25 to 1.70	<0.001	1.1
Hypercholesterolemia	0.84	0.75 to 0.95	0.01	0.3
Hypertensive	1.02	0.93 to 1.12	0.66	0
Myocardial Infarction/ Ischaemic Heart Disease*	1.07	0.96 to 1.19	0.23	0.1
TIA	1.07	0.94 to 1.21	0.30	0.1
Previous Cancer	1.23	1.05 to 1.44	0.01	0.3
Active Cancer	0.97	0.80 to 1.16	0.72	0
Depression	1.06	0.86 to 1.29	0.59	0
Rheumatoid Arthritis	1.10	0.92 to 1.31	0.31	0.1
COPD	0.86	0.71 to 1.06	0.15	0.1
Pre-stroke mRS Score (reference 0)			<0.001	5.5
1	1.57	1.38 to 1.79	<0.001	
2	1.63	1.39 to 1.91	<0.001	
3	1.94	1.65 to 2.28	<0.001	
4 & 5	1.32	1.13 to 1.55	<0.001	
Pre-stroke Residence (reference Independent living w/o formal care)			<0.001	1.4
Independent living with formal care	1.52	1.31 to 1.77	<0.001	
Institution	1.13	0.97 to 1.31	0.11	
Haemorrhagic Stroke	0.83	0.73 to 0.96	0.01	0.3
Oxford Community Stroke Project Classification (reference LACS)			<0.001	4.0
PACS	1.62	1.44 to 1.82	<0.001	
POCS	1.22	1.05 to 1.42	0.01	
TACS	1.66	1.45 to 1.90	<0.001	
Brain Lateralisation	0.69	0.60 to 0.80	<0.001	1.2
Inpatient Complication	2.13	1.94 to 2.34	<0.001	10.3
Discharge mRS Score (reference 0)			<0.001	31.1
1.24	1.07 to 1.42	0.003
2.04	1.75 to 2.39	<0.001
3.35	2.90 to 3.87	<0.001
4.20	3.60 to 4.90	<0.001
6.67	5.62 to 7.91	<0.001
1.57	1.37 to 1.80	<0.001
Winter Admission	1.20	1.09 to 1.31	<0.001	0.7
Weekend Admission	1.08	0.98 to 1.20	0.12	0.1
Hospital (reference 1)			<0.001	2.4
2.69	1.58 to 4.58	<0.001
1.19	1.02 to 1.39	0.03
1.24	1.01 to 1.53	0.04
0.86	0.75 to 0.99	0.03
1.11	0.94 to 1.31	0.22
1.18	1.00 to 1.41	0.05

Patient Characteristic	β	95% CI	P	R^2
0.86	0.72 to 1.03	0.11

AHLOS, Acute Hospital Length of Stay; CI, Confidence Interval; TIA, Transient Ischaemic Attack; COPD, Chronic Obstructive Pulmonary Disease; mRS, modified Rankin Scale; LACS, Lacunar Anterior Circulation Stroke; PACS, Partial Anterior Circulation Stroke; POCS, Posterior Circulation Stroke; TACS, Total Anterior Circulation Stroke.

For peer review only

Table S5 Univariable linear regression complete case analysis for AHLOS

Patient Characteristic	N	β	95% CI	P	% R ²
Age, y	2231	1.02	1.02 to 1.02	<0.001	4.7
Sex, female	1165 v. 1066	1.20	1.10 to 1.31	<0.001	0.7
Recurrent Stroke	448 v. 1755	1.17	1.05 to 1.31	0.005	0.3
Diabetes Mellitus	370 v. 1833	1.16	1.03 to 1.31	0.02	0.2
Dementia	207 v. 1996	1.46	1.25 to 1.70	<0.001	1.0
Hypercholesterolemia	355 v. 1848	0.85	0.75 to 0.95	0.01	0.3
Hypertensive	1483 v. 720	1.03	0.93 to 1.13	0.57	0
Myocardial Infarction or Ischaemic Heart Disease*	517 v. 1686	1.07	0.96 to 1.19	0.23	0
TIA	340 v. 1863	1.06	0.94 to 1.20	0.32	0
Previous Cancer	195 v. 2008	1.23	1.05 to 1.44	0.01	0.3
Active Cancer	137 v. 2066	0.96	0.80 to 1.15	0.65	0
Depression	117 v. 2086	1.05	0.86 to 1.28	0.65	0
Rheumatoid Arthritis	154 v. 2049	1.10	0.92 to 1.31	0.31	0
COPD	116 v. 2087	0.86	0.70 to 1.05	0.14	0.1
Pre-stroke mRS Score (reference 0)				<0.001	5.8
1	335 v. 914	1.58	1.39 to 1.80	<0.001	
2	191 v. 914	1.62	1.38 to 1.90	<0.001	
3	184 v. 914	1.97	1.67 to 2.31	<0.001	
4 & 5	167 v. 914	1.45	1.22 to 1.71	<0.001	
Pre-stroke Residence (reference Independent living without formal care)				<0.001	1.3
Independent living with formal care	210 v. 1752	1.52	1.31 to 1.77	<0.001	
Institution	220 v. 1752	1.14	0.98 to 1.32	0.09	
Haemorrhagic Stroke	273 v. 1864	0.85	0.74 to 0.97	0.02	0.2
Oxford Community Stroke Project Classification (reference LACS)				<0.001	4.3
PACS	784 v. 503	1.62	1.44 to 1.82	<0.001	
POCS	279 v. 503	1.24	1.06 to 1.44	0.01	
TACS	407 v. 503	1.75	1.53 to 2.01	<0.001	
No Brain Lateralisation	244 v. 1822	0.68	0.59 to 0.79	<0.001	1.3
Inpatient Complication	655 v. 1578	2.13	1.94 to 2.34	<0.001	10
Discharge mRS Score (reference 0)				<0.001	30.1
352 v. 260	1.25	1.08 to 1.44	0.002
v. 260	2.01	1.72 to 2.36	<0.001
291 v. 260	3.30	2.84 to 3.82	<0.001
238 v. 260	4.17	3.57 to 4.87	<0.001
v. 260	6.97	5.81 to 8.37	<0.001
414 v. 260	1.58	1.38 to 1.81	<0.001
Winter Admission	1159 v. 1074	1.20	1.09 to 1.31	<0.001	0.6
Weekend Admission	614 v. 1619	1.08	0.98 to 1.20	0.12	0.1
Hospital (reference 1)				<0.001	2.1
16 v. 350	2.69	1.58 to 4.58	<0.001
350 v. 350	1.19	1.02 to 1.39	0.03
v. 350	1.24	1.01 to 1.53	0.04
618 v. 350	0.86	0.75 to 0.99	0.03
281 v. 350	1.11	0.94 to 1.31	0.22
252 v. 350	1.18	1.00 to 1.41	0.05

Patient Characteristic	N	β	95% CI	P	% R ²
v. 350	0.86	0.72 to 1.03	0.11

AHLOS, Acute Hospital Length of Stay; CI, Confidence Interval; TIA, Transient Ischaemic Attack; COPD, Chronic Obstructive Pulmonary Disease; mRS, modified Rankin Scale; LACS, Lacunar Anterior Circulation Stroke; PACS, Partial Anterior Circulation Stroke; POCS, Posterior Circulation Stroke; TACS, Total Anterior Circulation Stroke.

For peer review only

Table S6 Multiple linear regression complete case analysis for AHLOS (n=1496, R²=44.7%).

Patient Characteristic	N	β	95% CI	P
Age, y	1496	1.01	1.00 to 1.01	<0.001
Sex, female	781 v. 715	0.98	0.90 to 1.07	0.66
Recurrent Stroke	328 v. 1168	1.06	0.96 to 1.17	0.27
Diabetes Mellitus	259 v. 1237	0.99	0.89 to 1.11	0.91
Dementia	138 v. 1358	1.32	1.13 to 1.53	<0.001
Hypercholesterolemia	264 v. 1232	0.92	0.82 to 1.02	0.13
Myocardial Infarction or Ischaemic Heart Disease*	362 v. 1134	1.00	0.91 to 1.10	0.97
Previous Cancer	140 v. 1356	1.16	1.01 to 1.33	0.03
COPD	76 v. 1420	0.91	0.76 to 1.09	0.31
Pre-stroke mRS Score (reference 0)				<0.001
1	284 v. 765	1.08	0.96 to 1.20	0.21
2	167 v. 765	0.93	0.80 to 1.08	0.33
3	149 v. 765	1.00	0.84 to 1.19	0.99
4 & 5	131 v. 765	0.77	0.63 to 0.93	0.01
Pre-Stroke Residence (reference Independent living without formal care)				<0.001
Independent living with formal care	145 v. 1215	1.02	0.88 to 1.19	0.78
Institution	136 v. 1215	0.83	0.69 to 0.98	0.03
Haemorrhagic Stroke	138 v. 1358	0.83	0.72 to 0.96	0.01
Oxford Community Stroke Project Classification				<0.001
PACS	570 v. 411	1.27	1.15 to 1.40	<0.001
POCS	214 v. 411	1.29	1.13 to 1.47	<0.001
TACS	301 v. 411	1.36	1.19 to 1.57	<0.001
No Brain Lateralisation	174 v. 1322	0.93	0.81 to 1.05	0.24
Inpatient Complication	421 v. 1075	1.67	1.51 to 1.84	<0.001
Discharge mRS Score (reference 0)				<0.001
295 v. 218	1.15	1.00 to 1.32	0.05
v. 218	1.60	1.36 to 1.88	<0.001
243 v. 218	2.45	2.10 to 2.87	<0.001
209 v. 218	3.39	2.86 to 4.02	<0.001
121 v. 218	4.78	3.89 to 5.88	<0.001
v. 218	1.34	1.11 to 1.61	0.002
Winter Admission	770 v. 726	1.16	1.07 to 1.25	<0.001
Weekend Admission	401 v. 1095	1.06	0.97 to 1.15	0.23
Hospital (reference1)				<0.001
2	14 v. 111	2.08	1.35 to 3.21	0.001
3	278 v. 111	1.20	1.01 to 1.44	0.04
4	-	-	-	-
5	558 v. 111	0.84	0.71 to 0.98	0.03
6	142 v. 111	1.03	0.85 to 1.26	0.75
7	191 v. 111	1.35	1.13 to 1.62	0.001
8	202 v. 111	0.94	0.78 to 1.13	0.49

AHLOS, Acute Hospital Length of Stay; CI, Confidence Interval; COPD, Chronic Obstructive Pulmonary Disease; mRS, modified Rankin Scale; LACS, Lacunar Anterior Circulation Stroke; PACS, Partial Anterior Circulation Stroke; POCS, Posterior Circulation Stroke; TACS, Total Anterior Circulation Stroke.

For peer review only

Table S7 Multiple linear regression sensitivity analysis for AHLOS, excluding Hospital 2 using multiple imputed dataset (n=2217, R²=44.7%).

Patient Characteristic	e ^β *	95% CI*	P
Age, y	1.01	1.00 to 1.01	<0.001
Sex, female	1.01	0.94 to 1.08	0.86
Recurrent Stroke	1.02	0.93 to 1.12	0.68
Diabetes Mellitus	1.07	0.97 to 1.17	0.19
Dementia	1.30	1.13 to 1.48	<0.001
Hypercholesterolemia	0.95	0.86 to 1.05	0.33
Myocardial Infarction or Ischaemic Heart Disease*	1.00	0.91 to 1.08	0.92
Previous Cancer	1.13	0.99 to 1.27	0.06
COPD	0.90	0.77 to 1.06	0.21
Pre-stroke mRS Score (reference 0)			<0.001
1	1.08	0.96 to 1.21	0.19
2	0.90	0.78 to 1.04	0.16
3	0.94	0.79 to 1.10	0.47
4 & 5	0.69	0.58 to 0.83	<0.001
Pre-Stroke Residence (reference Independent living without formal care)			<0.001
Independent living with formal care	1.01	0.88 to 1.16	0.91
Institution	0.81	0.69 to 0.95	0.01
Haemorrhagic Stroke	0.80	0.71 to 0.90	<0.001
Oxford Community Stroke Project Classification (reference LACS)			<0.001
PACS	1.30	1.18 to 1.43	<0.001
POCS	1.34	1.18 to 1.53	<0.001
TACS	1.29	1.13 to 1.47	<0.001
No Brain Lateralisation	0.85	0.75 to 0.95	0.01
Inpatient Complication	1.70	1.57 to 1.85	<0.001
Discharge mRS Score (reference 0)			<0.001
1.15	1.00 to 1.32	0.04
1.74	1.48 to 2.04	<0.001
2.72	2.34 to 3.16	<0.001
3.56	3.02 to 4.20	<0.001
5.12	4.22 to 6.22	<0.001
1.25	1.05 to 1.48	0.01
Winter Admission	1.15	1.08 to 1.24	<0.001
Weekend Admission	1.03	0.95 to 1.11	0.48
Hospital (reference 1)			<0.001
1.08	0.94 to 1.22	0.29
1.07	0.89 to 1.29	0.40
0.78	0.70 to 0.87	<0.001
0.93	0.81 to 1.07	0.33
1.15	1.00 to 1.32	0.05
0.82	0.70 to 0.95	0.01

AHLOS, Acute Hospital Length of Stay; CI, Confidence Interval; COPD, Chronic Obstructive Pulmonary Disease; mRS, modified Rankin Scale; LACS, Lacunar Anterior Circulation Stroke; PACS, Partial Anterior Circulation Stroke; POCS, Posterior Circulation Stroke; TACS, Total Anterior Circulation Stroke.

For peer review only

Table S8 Multiple linear regression sensitivity analysis for AHLOS, including discharge destination using multiple imputed dataset (n=2233, R²=40%).

Patient Characteristic	e ^β *	95% CI*	P
Age, y	1.01	1.00 to 1.01	<0.001
Sex, female	0.99	0.92 to 1.07	0.80
Recurrent Stroke	1.00	0.91 to 1.10	1.00
Diabetes Mellitus	1.08	0.98 to 1.19	0.12
Dementia	1.20	1.05 to 1.38	0.01
Hypercholesterolemia	0.94	0.85 to 1.04	0.25
Myocardial Infarction or Ischaemic Heart	1.01	0.93 to 1.10	0.83
Previous Cancer	1.17	1.03 to 1.33	0.01
COPD	0.91	0.77 to 1.07	0.23
Pre-stroke mRS Score (reference 0)			<0.001
1	1.15	1.03 to 1.28	0.02
2	1.15	1.00 to 1.33	0.05
3	1.33	1.13 to 1.56	<0.001
4 & 5	1.15	0.96 to 1.38	0.12
Pre-Stroke Residence (reference Independent living without formal care)			<0.001
Independent living with formal care	0.86	0.75 to 0.99	0.04
Institution	0.52	0.44 to 0.62	<0.001
Haemorrhagic Stroke	0.84	0.75 to 0.95	<0.001
Oxford Community Stroke Project Classification (reference LACS)			<0.001
PACS	1.34	1.22 to 1.48	<0.001
POCS	1.44	1.26 to 1.63	<0.001
TACS	1.49	1.31 to 1.70	<0.001
No Brain Lateralisation	0.82	0.73 to 0.93	<0.001
Inpatient Complication	1.72	1.58 to 1.87	<0.001
Discharge Destination (reference Independent living without formal care)			<0.001
Independent living with formal care	1.99	1.74 to 2.27	<0.001
Institution	3.58	3.09 to 4.15	<0.001
Interim/Rehab Setting	2.18	1.94 to 2.46	<0.001
Death	0.85	0.74 to 0.97	0.02
Winter Admission	1.15	1.07 to 1.24	<0.001
Weekend Admission	1.04	0.96 to 1.13	0.30
Hospital (reference 1)			<0.001
2.76	1.80 to 4.22	<0.001
1.24	1.09 to 1.42	<0.001
1.36	1.15 to 1.61	<0.001
0.85	0.75 to 0.95	0.01
1.06	0.92 to 1.22	0.42
1.19	1.03 to 1.37	0.02
0.99	0.85 to 1.14	0.84

AHLOS, Acute Hospital Length of Stay; CI, Confidence Interval; COPD, Chronic Obstructive Pulmonary Disease; mRS, modified Rankin Scale; LACS, Lacunar Anterior Circulation Stroke; PACS, Partial Anterior Circulation Stroke; POCS, Posterior Circulation Stroke; TACS, Total Anterior Circulation Stroke.

Figure S1 Model estimates of mean baseline acute hospital length of stay (AHLOS) per hospital (in days) and number of stroke patients treated outside the stroke unit per day per five stroke unit beds with 95% confidence intervals. Multiple regression model was adjusted for patient covariates that had a p-value<0.3 in univariable analysis.

Figure S2 Model estimates of mean baseline acute hospital length of stay (AHLOS) per hospital (in days) and presence of vascular surgery onsite with 95% confidence intervals. Multiple regression model was adjusted for patient covariates that had a p-value<0.3 in univariable analysis.

Figure S3 Model estimates of mean baseline acute hospital length of stay (AHLOS) per hospital (in days) and distance to neurosurgical facility with 95% confidence intervals. Multiple regression model was adjusted for patient covariates that had a p-value < 0.3 in univariable analysis.

Figure S4 Model estimates of mean baseline acute hospital length of stay (AHLOS) per hospital (in days) and number of fte senior doctors per five beds available during weekdays with 95% confidence intervals. Multiple regression model was adjusted for patient covariates that had a p-value<0.3 in univariable analysis.

Figure S5 Model estimates of mean baseline acute hospital length of stay (AHLOS) per hospital (in days) and number of fte junior doctors per five beds available during weekdays with 95% confidence intervals. Multiple regression model was adjusted for patient covariates that had a p-value<0.3 in univariable analysis.

Figure S6 Model estimates of mean baseline acute hospital length of stay (AHLOS) per hospital (in days) and number of fte health care associates and nurses per five beds with 95% confidence intervals. Multiple regression model was adjusted for patient covariates that had a p-value<0.3 in univariable analysis.

Figure S7 Model estimates of mean baseline acute hospital length of stay (AHLOS) per hospital (in days) and number of fte occupational therapists per five beds with 95% confidence intervals. Multiple regression model was adjusted for patient covariates that had a p-value<0.3 in univariable analysis.

Figure S8 Model estimates of mean baseline acute hospital length of stay (AHLOS) per hospital (in days) and number of fte physiotherapists per five beds with 95% confidence intervals. Multiple regression model was adjusted for patient covariates that had a p-value<0.3 in univariable analysis.

Figure S9 Model estimates of mean baseline acute hospital length of stay (AHLOS) per hospital (in days) and number of fte speech and language therapists per five beds with 95% confidence intervals. Multiple regression model was adjusted for patient covariates that had a p-value<0.3 in univariable analysis.

Figure S10 Model estimates of mean baseline acute hospital length of stay (AHLOS) per hospital (in days) and number of total beds present on stroke unit per 100 admissions with 95% confidence intervals. Multiple regression model was adjusted for patient covariates that had a p-value<0.3 in univariable analysis.

Figure S11 Model estimates of mean baseline acute hospital length of stay (AHLOS) per hospital (in days) and number of hospital beds per computed tomography (CT) scanner with 95% confidence intervals. Multiple regression model was adjusted for patient covariates that had a p-value<0.3 in univariable analysis.

Figure S12 Model estimates of mean baseline acute hospital length of stay (AHLOS) per hospital (in days) and provision of onsite rehabilitation service with 95% confidence intervals. Multiple regression model was adjusted for patient covariates that had a p-value < 0.3 in univariable analysis.

Figure S13 Model estimates of mean baseline acute hospital length of stay (AHLOS) per hospital (in days) and presence of early supported discharge scheme with 95% confidence intervals. Multiple regression model was adjusted for patient covariates that had a p-value < 0.3 in univariable analysis.

Figure S14 Model estimates of mean baseline acute hospital length of stay (AHLOS) per hospital (in days) and number of non-stroke patients present on the stroke unit per day per five stroke unit beds with 95% confidence intervals. Multiple regression model was adjusted for patient covariates that had a p-value<0.3 in univariable analysis.

Figure S15 Model estimates of mean baseline acute hospital length of stay (AHLOS) per hospital (in days) and mean Index of Multiple Deprivation (IMD) score of the counties in which the hospital serves with 95% confidence intervals. Multiple regression model was adjusted for patient covariates that had a p-value<0.3 in univariable analysis.

STROBE 2007 (v4) Statement—Checklist of items that should be included in reports of *cohort studies*

Section/Topic	Item #	Recommendation	Reported on page #
Title and abstract	1	(a) Indicate the study's design with a commonly used term in the title or the abstract	1 & 2
		(b) Provide in the abstract an informative and balanced summary of what was done and what was found	2-3
Introduction			
Background/rationale	2	Explain the scientific background and rationale for the investigation being reported	5
Objectives	3	State specific objectives, including any prespecified hypotheses	6
Methods			
Study design	4	Present key elements of study design early in the paper	7
Setting	5	Describe the setting, locations, and relevant dates, including periods of recruitment, exposure, follow-up, and data collection	7-8
Participants	6	(a) Give the eligibility criteria, and the sources and methods of selection of participants. Describe methods of follow-up	7
		(b) For matched studies, give matching criteria and number of exposed and unexposed	NA
Variables	7	Clearly define all outcomes, exposures, predictors, potential confounders, and effect modifiers. Give diagnostic criteria, if applicable	8-10
Data sources/ measurement	8*	For each variable of interest, give sources of data and details of methods of assessment (measurement). Describe comparability of assessment methods if there is more than one group	8
Bias	9	Describe any efforts to address potential sources of bias	10-12
Study size	10	Explain how the study size was arrived at	7
Quantitative variables	11	Explain how quantitative variables were handled in the analyses. If applicable, describe which groupings were chosen and why	8-10
Statistical methods	12	(a) Describe all statistical methods, including those used to control for confounding	11
		(b) Describe any methods used to examine subgroups and interactions	NA
		(c) Explain how missing data were addressed	12
		(d) If applicable, explain how loss to follow-up was addressed	NA
		(e) Describe any sensitivity analyses	11-12
Results			

Participants	13*	(a) Report numbers of individuals at each stage of study—eg numbers potentially eligible, examined for eligibility, confirmed eligible, included in the study, completing follow-up, and analysed	14
		(b) Give reasons for non-participation at each stage	14
		(c) Consider use of a flow diagram	Fig 1
Descriptive data	14*	(a) Give characteristics of study participants (eg demographic, clinical, social) and information on exposures and potential confounders	14
		(b) Indicate number of participants with missing data for each variable of interest	14 &15
		(c) Summarise follow-up time (eg, average and total amount)	NA
Outcome data	15*	Report numbers of outcome events or summary measures over time	16
Main results	16	(a) Give unadjusted estimates and, if applicable, confounder-adjusted estimates and their precision (eg, 95% confidence interval). Make clear which confounders were adjusted for and why they were included	19-21 (and Table S4)
		(b) Report category boundaries when continuous variables were categorized	NA
		(c) If relevant, consider translating estimates of relative risk into absolute risk for a meaningful time period	NA
Other analyses	17	Report other analyses done—eg analyses of subgroups and interactions, and sensitivity analyses	22
Discussion			
Key results	18	Summarise key results with reference to study objectives	23
Limitations			
Interpretation	20	Give a cautious overall interpretation of results considering objectives, limitations, multiplicity of analyses, results from similar studies, and other relevant evidence	23-27
Generalisability	21	Discuss the generalisability (external validity) of the study results	27
Other information			
Funding	22	Give the source of funding and the role of the funders for the present study and, if applicable, for the original study on which the present article is based	28

*Give information separately for cases and controls in case-control studies and, if applicable, for exposed and unexposed groups in cohort and cross-sectional studies.

Note: An Explanation and Elaboration article discusses each checklist item and gives methodological background and published examples of transparent reporting. The STROBE checklist is best used in conjunction with this article (freely available on the Web sites of PLoS Medicine at <http://www.plosmedicine.org/>, Annals of Internal Medicine at <http://www.annals.org/>, and Epidemiology at <http://www.epidem.com/>). Information on the STROBE Initiative is available at www.strobe-statement.org.

BMJ Open

Does service heterogeneity have an impact on acute hospital length of stay in stroke? A UK-based multi-centre prospective cohort study

Journal:	BMJ Open
Manuscript ID	bmjopen-2018-024506.R2
Article Type:	Research
Date Submitted by the Author:	26-Feb-2019
Complete List of Authors:	Tørnes, Michelle; University of Aberdeen College of Life Sciences and Medicine, Ageing Clinical and Experimental Research Group McLernon, David; University of Aberdeen College of Life Sciences and Medicine, Medical Statistics Team Bachmann, Max; Univeristy of East Anglia, Norwich Medical School Musgrave, Stanley; University of East Anglia, Norwich Medical School Warburton, Elizabeth; University of Cambridge, Addenbrooke's Hospital Potter, John; University of East Anglia, Norwich Medical School; Norfolk and Norwich University Hospital, Stroke Research Group Myint, Phyo; University of Aberdeen College of Life Sciences and Medicine, Ageing Clinical and Experimental Research Group; Norfolk and Norwich University Hospital, Stroke Research Group
Primary Subject Heading:	Epidemiology
Secondary Subject Heading:	Health services research, Neurology, Geriatric medicine
Keywords:	Acute hospital, Health Services Research, Length of Stay, Outcome, Stroke < NEUROLOGY

**Does service heterogeneity have an impact on acute hospital length of stay in stroke? A UK-based**
**multi-centre prospective cohort study**

Michelle Tørnes¹, David McLernon², Max O Bachmann³, Stanley D Musgrave³, Elizabeth A
Warburton⁴, John F Potter³, Phyo Kyaw Myint^{1,5}: On behalf of the Anglia Stroke Clinical
Network Evaluation Study (ASCNES) Group

¹Ageing Clinical and Experimental Research (ACER) Group, Institute of Applied Health
Sciences, School of Medicine, Medical Sciences and Nutrition, University of Aberdeen,
Scotland, UK; ²Medical Statistics Team, Institute of Applied Health Sciences, School of
Medicine, Medical Sciences and Nutrition, University of Aberdeen, Scotland, UK; ³Norwich
Medical School, University of East Anglia, Norwich, UK; ⁴Addenbrooke's Hospital,
Cambridge, UK; ⁵Stroke Research Group, Norfolk and Norwich University Hospital,
Norwich, UK.

**Correspondence to:**

Michelle Tørnes,

C/o: Professor P K Myint; Room 4.013, Polwarth Building | School of Medicine & Dentistry |
Division of Applied Health Sciences | Foresterhill, University of Aberdeen | Aberdeen, AB25
2ZD, UK | Tel: +44 (0) 1224 437843 | Fax: +44 (0) 1224 437911 | Mail to:

michelle.tornes@abdn.ac.uk

Word count: 4821

**Keywords:** Acute hospital, Health Services Research, Length of Stay, Outcome, Stroke

[revised manuscript text omitted]

0	260 (12)	329 (15)
352 (16)
212 (9)
291 (13)
238 (11)
137 (6)
414 (19)
Winter Admission	1159 (52)	0 (0)
Weekend Admission	614 (27)	0 (0)

IQR, Interquartile Range; TIA, Transient Ischaemic Attack; COPD, Chronic Obstructive Pulmonary Disorder; mRS, modified Rankin Scale; LACS, Lacunar Anterior Circulation Stroke; PACS, Partial Anterior Circulation Stroke; POCS, Posterior Circulation Stroke; TACS, Total Anterior Circulation Stroke.

*No information was assumed to indicate absence of condition or complication

**Hospital service characteristics**

Service characteristics of each hospital are outlined in Table 2, with median AHLOS.

After standardization, by taking account of stroke admission volume, number of stroke unit
beds, and size of hospital, there was still extensive heterogeneity in bed capacity, staffing
levels, and the number of CT scanners provided at each hospital, respectively. Variations
between hospitals also existed in terms of service and facility provision. For example, a
number of hospitals provided rehabilitation care, neurosurgery or vascular surgery onsite,
whilst others did not. The overall median AHLOS (IQR) was 9 (4 to 21) days and there
appeared to be crude variations in this outcome between hospitals.

Table 2 Hospital characteristics per individual hospital self-reported by clinical leads or service managers at each hospital

Hospital Characteristics	1	2	3	4	5	6	7	8
General Characteristics								
Catchment Population	400,000	160,000	350,000	230,000	680,000	300,000	240,000	275,000
Hospital Type	Tertiary	Secondary	Secondary	Secondary	Tertiary	Secondary	Secondary	Secondary
Hospital Stroke Volume (No. of ASCNES admissions per month)	52	13	46	19	88	57	35	31
Facilities and Services								
No. of hospital beds	1000	304	800	500	1237	611	488	460
No. of stroke unit beds (per 100 admissions)	71	77	54	138	41	55	83	65
No. of hospital beds per CT scanners	500	304	400	250	518	306	244	230
Distance to Vascular Surgery (miles)	0	18	0	25	0	0	43	30
Distance to Neurosurgery (miles)	0	18	58	89	61	38	48	30
Rehabilitation Provision	Onsite	Onsite	Offsite	Offsite	Offsite	Onsite	Offsite	Onsite
Early Supported Discharge Provision	No	Yes	No	Yes	Yes	Yes	No	No
Stroke Unit Staffing Levels*								
Senior doctors †	0.34	0.25	0.49	0.47	0.42	0.31	0.62	0.87
Junior doctors †	0.55	0.65	0.72	0.59	0.56	0.64	0.12	0.25
Health care associates and nurses (band 5-7)	9.2	8	6	7.4	7	5.3	6.5	10
Physiotherapists (band 2-8)	0.55	1	0.79	0.4	0.91	0.78	0.69	1
Occupational Therapists (band 3-8)	0.49	0.5	1.4	0.59	0.6	0.58	0.52	1.1
Speech and Language Therapists	0.39	0.15	0.2	0.18	0.35	0.03	0.26	0.1
No. of non-stroke patients treated daily on stroke unit (per five stroke unit beds)	0.27	0	0.10	0.47	0.05	0.31	0.17	0
No. of patients with stroke treated daily outside stroke unit (per five stroke unit beds)	0.14	5	0	0.30	0.01	0.41	0	0
Median AHLOS (IQR)	8 (4 to 20)	29 (24 to 42)	11 (5 to 27)	14 (4 to 30)	8 (4 to 14)	10 (5 to 22)	11 (6 to 23)	7 (3 to 20)

ASCNES, Anglia Stroke Clinical Network Evaluation Study; CT, Computerised Tomography; AHLOS, Acute Hospital Length of Stay; IQR, Interquartile Range.

*Number of fte staff per five stroke unit beds (weighted average for the four study periods taken). NHS banding refers to the pay scale system of healthcare staff in the UK
and relates to their level of experience. Higher bands reflect higher pay and experience.

† Weekday numbers only

For peer review only

Univariable linear regression

In univariable linear regression (Table S4 in the online supplementary document 2), patients who were older, female, had previous cancer, a previous stroke, had diabetes mellitus, had dementia, had a pre-stroke or discharge mRS score greater than 0, had a OCSP other than a lacunar infarct, had an inpatient complication, were living independently at home without formal care (compared to those who had formal care) prior to stroke, or were a winter admission had a significantly longer AHLOS ($p < 0.05$). Patients who had a haemorrhagic stroke, hypercholesterolemia, or showed no signs of brain lateralisation were all shown to be significantly associated with a shorter AHLOS ($p < 0.01$).

The strongest associations with AHLOS were seen for inpatients who developed a complication, who had a pre-stroke mRS score of 3, who were admitted to Hospital 2 or who had a discharge mRS score of ≥ 2 . Inpatient complications were associated with twice as long an AHLOS compared to those without a complication. Similarly, patients with a pre-stroke mRS score of 3 were 94% more likely to have a longer AHLOS than those with an mRS of 0. Patients admitted to Hospital 2 had 2.69 times the AHLOS of those admitted to Hospital 1. Compared to patients with a discharge mRS score, those with a score of 2, 3, 4 or 5 had over a 2, 3, 4, and 5-fold increase in AHLOS, respectively. Unsurprisingly, discharge mRS score appeared to explain the majority of AHLOS variance ($R^2 = 31.1\%$).

Being hypertensive, having a history of a myocardial infarction or ischaemic heart disease, having previously had a TIA, having active cancer, depression, rheumatoid arthritis or chronic obstructive pulmonary disease were not shown to be significantly associated with AHLOS. Furthermore, admissions to Hospitals 6 and 8 were also not shown to be significantly associated with a difference in AHLOS compared to Hospital 1 admissions.

Multiple linear regression

Multiple linear regression results for AHLOS are summarized in Table 3 and shows that
42.7% of the variation in AHLOS has been explained. Sex, recurrent stroke, diabetes
mellitus, hypercholesterolemia, previous cancer, a pre-stroke mRS score of 1 to 3 (with
reference to a score of 0) and living at home independently without formal care prior to
stroke were no longer statistically associated with AHLOS in multiple regression ($p>0.05$).
Furthermore, being admitted to Hospital 3 or 4 as opposed to Hospital 1 were no longer
associated with a significant difference in AHLOS. No variables included from the
univariable analysis with $p>0.05$ became statistically significant in the multivariable analysis,
except for living in an institution prior to stroke which was associated with a 19% reduced
AHLOS compared to those living independently without formal care. Developing an
inpatient complication and having a discharge mRS score between 2 and 5 were still strongly
positively related to AHLOS. After adjusting for patient covariates, AHLOS was still shown
to significantly differ between hospitals, with the shortest and longest AHLOS observed for
Hospitals 5 and 2, respectively.

There were no obvious differences between the results using complete cases only (Tables S5-
6 in the online supplementary document 2) and multiple imputation.

Table 3 Multiple regression analysis for AHLOS (n=2233; R²=42.7%)

Patient Characteristic	e ^β *	95% CI*	P
Age, y	1.01	1.00 to 1.01	<0.001
Sex, female	1.01	0.94 to 1.09	0.79
Recurrent Stroke	1.03	0.94 to 1.12	0.57
Diabetes Mellitus	1.06	0.97 to 1.17	0.21
Dementia	1.28	1.12 to 1.46	<0.001
Hypercholesterolemia	0.94	0.85 to 1.05	0.27
Myocardial Infarction or Ischaemic Heart	1.00	0.92 to 1.09	0.98
Previous Cancer	1.12	0.99 to 1.27	0.08
COPD	0.90	0.77 to 1.06	0.21
Pre-stroke mRS Score (reference 0)			<0.001
1	1.06	0.95 to 1.19	0.28
2	0.90	0.77 to 1.04	0.15
3	0.94	0.80 to 1.11	0.47
4 & 5	0.71	0.59 to 0.86	<0.001
Pre-Stroke Residence (reference Independent living without formal care)			<0.001
Independent living with formal care	1.07	0.94 to 1.23	0.92
Institution	0.81	0.69 to 0.95	0.01
Haemorrhagic Stroke	0.80	0.71 to 0.90	<0.001
Oxford Community Stroke Project Classification (reference LACS)			<0.001
PACS	1.30	1.18 to 1.42	<0.001
POCS	1.34	1.18 to 1.53	<0.001
TACS	1.29	1.13 to 1.48	<0.001
No Brain Lateralisation	0.85	0.75 to 0.96	0.01
Inpatient Complication	1.70	1.56 to 1.85	<0.001
Discharge mRS Score (reference 0)			<0.001
1.15	1.01 to 1.31	0.04
1.74	1.50 to 2.04	<0.001
2.70	2.32 to 3.13	<0.001
3.51	2.98 to 4.14	<0.001
5.07	4.19 to 6.14	<0.001
1.24	1.05 to 1.48	0.01
Winter Admission	1.15	1.08 to 1.24	<0.001
Weekend Admission	1.03	0.95 to 1.11	0.50
Hospital (reference 1)			<0.001
2.09	1.38 to 3.17	0.001
1.07	0.94 to 1.22	0.29
1.08	0.90 to 1.31	0.40
0.78	0.69 to 0.87	<0.001
0.93	0.81 to 1.07	0.33
1.15	1.00 to 1.32	0.05
0.82	0.70 to 0.94	0.01

1
2
3 AHLOS, Acute Hospital Length of Stay; CI, Confidence Intervals; COPD, Chronic Obstructive Pulmonary
4 Disorder; mRS, modified Rankin Scale; LACS, Lacunar Anterior Circulation Stroke; 
[revised manuscript text omitted]

P.K.M supervised the study. D.M provided statistical analysis advice and assistance. M.O.B,
E.A.W, J.F.P and P.K.M conceptualized and designed the use of ASCNES and obtained
funding. S.D.M was ASCNES study coordinator and managed the data. All authors reviewed
the manuscript and contributed in critical revision and final manuscript preparation.

**Funding:** This work was supported by the National Institute for Health Research (NIHR)
Research for Patient Benefit Programme (PB-PG-1208-18240). EAW receives funding
support from the NIHR Biomedical Research Centre award to Cambridge. MT holds a PhD
studentship funded by the College of Life Sciences & Medicine, University Aberdeen
(CF10109-38).

**Disclaimer:** The views expressed are those of the authors and not necessarily those of the
NHS, the NIHR or the Department of Health.

**Competing interests:** The authors declare no competing interests.

**Patient consent:** Not required.

**Ethics approval:** Ethical approval was obtained from the NRES Committee East of England
– Norfolk (REC Reference number 10/H0310/44).

**Data sharing statement:** The datasets generated and analysed during the current study are
available from the ASCNES team on reasonable request.

REFERENCES

1. Feigin VL, Norrving B, Mensah GA. Global burden of stroke. *Circ Res* 2017;120:439-48. doi:10.1161/CIRCRESAHA.116.308413.
2. Krishnamurthi RV, Feigin VL, Forouzanfar MH, et al on behalf of the Global Burden of Diseases, Injuries, and Risk Factors Study 2010 (GBD 2010) and the GBD Stroke Experts Group. Global and regional burden of first-ever ischaemic and haemorrhagic stroke during 1990-2010: findings from the Global Burden of Disease Study 2010. *The Lancet Glob health*. 2013;1:e259-e281. doi:10.1016/S2214-109X(13)70089-5.
3. Mu F, Hurley D, Betts KA, et al. Real-world costs of ischemic stroke by discharge status. *Curr Med Res Opin* 2016;33:371-8. doi:10.1080/03007995.2016.1257979.
4. Ovbiagele B, Goldstein LB, Higashida RT, et al on behalf of the American Heart Association Advocacy Coordinating Committee and Stroke Council. Forecasting the future of stroke in the United States: A policy statement from the American Heart Association and American Stroke Association. *Stroke* 2010;44:2361-75. doi:10.1161/STR.0b013e31829734f2.
5. Townsend N, Bhatnagar P, Wilkins E et al. Cardiovascular disease statistics 2015. London: British Heart Foundation, 2015.
6. Marshall IJ, Wang Y, Crichton S, et al. The effects of socioeconomic status on stroke risk and outcomes. *Lancet Neurol* 2015;14:1206-18. doi:10.1016/S1474-4422(15)00200-8.
7. Bray BD, Ayis S, Campbell J, et al. Associations between stroke mortality and weekend working by stroke specialist physicians and registered nurses: prospective multicentre cohort study. *PLoS Med* 2014;11:e1001705. doi:10.1371/journal.pmed.1001705.

8. Bray BD, Ayis S, Campbell J, et al. Associations between the organisation of stroke
services, process of care, and mortality in England: prospective cohort study. *BMJ* 2013;346.
doi:10.1136/bmj.f2827.
9. Rudd AG, Jenkinson D, Grant RL, et al. Staffing levels and patient dependence in English
stroke units. *Clin Med (Lond)* 2009;9:110-5. doi:10.7861/clinmedicine.9-2-110.
10. Myint PK, Bachmann MO, Loke YK, et al. Important factors in predicting mortality
outcome from stroke: findings from the Anglia Stroke Clinical Network Evaluation Study.
*Age Ageing* 2017;46:83-90. doi:10.1093/ageing/afw175.
11. Fonarow GC, Smith EE, Reeves MJ, et al for the Get With The Guidelines Steering
Committee and Hospitals. Hospital-level variation in mortality and rehospitalization for
medicare beneficiaries with acute ischemic stroke. *Stroke* 2011;42:159-66.
doi:10.1161/STROKEAHA.110.601831.
12. Kim SM, Hwang SW, Oh E, et al. Determinants of the length of stay in stroke patients.
*Osong Public Health and Research Perspectives* 2013;4:329-41.
13. Kwok CS, Clark A, Ford GA, et al. Association between prestroke disability and inpatient
mortality and length of acute hospital stay after acute stroke. *J Am Geriatr Soc* 2012;60:726-
32. doi:10.1111/j.1532-5415.2011.03889.x.
14. Appelros P. Prediction of length of stay for stroke patients. *Acta Neurol Scand*
2007;116:15-9. doi:10.1111/j.1600-0404.2006.00756.x.

[revised manuscript text omitted]

38. Elrod JK, Fortenberry Jr JL. The hub-and-spoke organization design revisited: a lifeline
for rural hospitals. *BMC Health Serv Res* 2017;17:795. doi:10.1186/s12913-017-2755-5.
39. Matsui H, Fushimi K, Yasunaga H. Variation in risk-standardized mortality of stroke
among hospitals in Japan. *PLoS One* 2015;10:e0139216. doi:10.1371/journal.pone.0139216.
40. Tsugawa Y, Kumamaru H, Yasunaga H, et al. The association of hospital volume with
mortality and costs of care for stroke in Japan. *Med Care* 2013;51:782-8.
doi:10.1097/MLR.0b013e31829c8b70.
41. Stroke Unit Trialists' Collaboration. Organised inpatient (stroke unit) care for stroke
(Review). *Cochrane Database Syst Rev* 2013;9. doi: 10.1002/14651858.CD000197.pub3.
42. Chan DKY, Cordato D, O'Rourke F, et al. Comprehensive stroke units: a review of
comparative evidence and experience. *Int J Stroke* 2013;8:260-4. doi:10.1111/j.1747-
4949.2012.00850.x.
43. Chiu A, Shen Q, Cheuk G, et al. Establishment of a stroke unit in a district hospital:
review of experience. *Intern Med J* 2007;37:73-8. doi:10.1111/j.1445-5994.2007.01235.x
44. Tamm A, Siddiqui M, Shuaib A, et al. Impact of stroke care unit on patient outcomes in a
community hospital. *Stroke* 2014;45:211-6. doi:10.1161/STROKEAHA.113.002504.
45. Ingeman A, Andersen G, Hundborg HH, et al. In-hospital medical complications, length
of stay, and mortality among stroke unit patients. *Stroke* 2011;42:3214-8.
doi:10.1161/STROKEAHA.110.610881.

46. Tirschwell DL, Kukull WA, Longstreth WT. Medical complications of ischemic stroke
and length of hospital stay: experience in Seattle, Washington. *J Stroke Cerebrovasc Dis*
1999;8:336-43. doi:[10.1016/S1052-3057\(99\)80008-1](https://doi.org/10.1016/S1052-3057(99)80008-1).
47. Myint PK, Vowler SL, Woodhouse PR, et al. Winter excess in hospital admissions, in-
patient mortality and length of acute hospital stay in stroke: a hospital database study over six
seasonal years in Norfolk, UK. *Neuroepidemiology* 2007;28:79-85. doi:10.1159/000098550.
48. Nuyen J, Spreuwenberg PM, Groenewegen PP, et al. Impact of preexisting depression
on length of stay and discharge destination among patients hospitalized for acute stroke:
linked register-based study. *Stroke* 2008;39:132-8. doi:[10.1161/STROKEAHA.107.490565](https://doi.org/10.1161/STROKEAHA.107.490565).
49. Jia H, Damush TM, Qin H, et al. The impact of poststroke depression on healthcare use
by veterans with acute stroke. *Stroke* 2006;37:2796-801.
50. Schwamm LH, Reeves MJ, Pan W, et al. Race/ethnicity, quality of care, and outcomes in
ischemic stroke. *Circulation* 2010;121:1492-1501.
51. Ng YS, Tan KH, Chen C, et al. Predictors of acute, rehabilitation and total length of
stay in acute stroke: a prospective cohort study. *Ann Acad Med Singapore* 2016;45:394-403.
52. Office of National Statistics. *Dataset(s): 2011 census: Key statistics and quick statistics*
for local authorities in the united kingdom - part 1. KS201UK ethnic group, local authorities
in the united kingdom. London: ONS, 2013.
<https://www.ons.gov.uk/peoplepopulationandcommunity/populationandmigration/population>

estimates/datasets/2011censuskeystatisticsandquickstatisticsforlocalauthoritiesintheunitedkin
gdompart1 (accessed 12 Dec 2018).

For peer review only

**Figure 1** Flow chart of patient participation inclusion and exclusion for study analysis

**Figure 2** Model estimates of mean baseline acute hospital length of stay (AHLOS) per
hospital (in days) against hospital stroke volume and clustered by hospital type with 95%
confidence intervals. Multiple regression model was adjusted for patient covariates that had a
p-value<0.3 in univariable analysis.

For peer review only

Figure 1 Flow chart of patient participation inclusion and exclusion for study analysis

Figure 2 Model estimates of mean baseline acute hospital length of stay (AHLOS) per hospital (in days) against hospital stroke volume and clustered by hospital type with 95% confidence intervals. Multiple regression model was adjusted for patient covariates that had a p-value<0.3 in univariable analysis.

152x152mm (300 x 300 DPI)

STUDY PROTOCOL

Open Access

Evaluation of stroke services in Anglia stroke clinical network to examine the variation in acute services and stroke outcomes

Phyo K Myint^{1,2*}, John F Potter^{1,2}, Gill M Price¹, Garry R Barton¹, Anthony K Metcalf², Rachel Hale², Genevieve Dalton³, Stanley D Musgrave¹, Abraham George⁴, Raj Shekhar⁵, Peter Owusu-Agyei⁶, Kevin Walsh⁷, Joseph Ngeh⁸, Anne Nicholson⁹, Diana J Day¹⁰, Elizabeth A Warburton¹⁰, Max O Bachmann¹

Abstract

Background: Stroke is the third leading cause of death in developed countries and the leading cause of long-term disability worldwide. A series of national stroke audits in the UK highlighted the differences in stroke care between hospitals. The study aims to describe variation in outcomes following stroke and to identify the characteristics of services that are associated with better outcomes, after accounting for case mix differences and individual prognostic factors.

Methods/Design: We will conduct a cohort study in eight acute NHS trusts within East of England, with at least one year of follow-up after stroke. The study population will be a systematically selected representative sample of patients admitted with stroke during the study period, recruited within each hospital. We will collect individual patient data on prognostic characteristics, health care received, outcomes and costs of care and we will also record relevant characteristics of each provider organisation. The determinants of one year outcome including patient reported outcome will be assessed statistically with proportional hazards regression models. Self (or proxy) completed EuroQoL (EQ-5D) questionnaires will measure quality of life at baseline and follow-up for cost utility analyses.

Discussion: This study will provide observational data about health service factors associated with variations in patient outcomes and health care costs following hospital admission for acute stroke. This will form the basis for future RCTs by identifying promising health service interventions, assessing the feasibility of recruiting and following up trial patients, and provide evidence about frequency and variances in outcomes, and intra-cluster correlation of outcomes, for sample size calculations. The results will inform clinicians, public, service providers, commissioners and policy makers to drive further improvement in health services which will bring direct benefit to the patients.

Background

Stroke is the third leading cause of mortality and the number one cause of long-term disability in the UK. More than 150,000 people suffer a stroke in the UK each year [1]. It costs the NHS approximately £ 7 billion per annum [2]. Stroke incidence rises sharply with age and despite better primary and secondary preventative measures, the total number of strokes is set to rise in

the UK [3]. Nevertheless, stroke care in UK is far from ideal: patients having a worse outcome in terms of death and dependency than many other European countries [4-6], at least in part due to differences in care provided [7]. There is also variation in outcome between different localities within the UK [8-11], these local differences being highlighted in the most recent publication of the National Sentinel Stroke Audit in 2009 [12]. These differences probably arise as a result of substantial variations in how the stroke services are provided across the UK. Examples of such differences are access to neurovascular/neurosurgical service, early supported

* Correspondence: phyo.k.myint@uea.ac.uk

¹Norwich Medical School, Faculty of Medicine & Health Sciences, Norwich, UK

Full list of author information is available at the end of the article

discharge, and stroke specialist on call rota for thrombolysis. The presence or absence of variations in stroke outcomes as a result of variation in care and how much the observed variations in patients' outcomes including patient reported outcome measure (PROM) are determined by the differences in service delivery have not been examined previously.

We hypothesise that variation in patient outcomes including mortality, length of stay, institutionalisation rate, and patient reported outcomes between care providers can partly be explained by the different ways in which stroke services are delivered. The main objectives of the study are (1) to describe variation in outcomes following stroke and to identify the characteristics of services that are associated with better outcomes after accounting for case mix differences and individual prognostic factors, and (2) to obtain preliminary data to identify sample size and inform future pragmatic real world setting RCTs in the area of health service delivery in stroke.

25 **Methods/Design**

A prospective cohort study will be conducted to identify characteristics of services that are associated with the best outcomes including patient reported outcomes, taking into account case-mix and patients' prognostic features. The study will consist of two components (1) consecutive stroke admissions in selected months (a total of 8 months) and (2) a prospective study of patient reported outcome in some of these selected months.

**Sample Population**

For the first component, the sample population will be stroke patients who are admitted to any of the hospitals within the Anglia region of Stroke & Heart Clinical Network between October 2009 and September 2011. Baseline data are already recorded, prior to the study commencement, as part of routine clinical data collection by Anglia Stroke Clinical Network (as described in detail below). The study sample will be a systematically selected sample (every third month) rather than a consecutive cohort of patients admitted to eight acute NHS hospital trusts. Therefore, this is not a consecutive case study; instead it seeks to be representative of the catchment population of the hospital and has taken into account the seasonal variation in stroke incidence and outcome [13].

For the patient reported outcome component of the study the following inclusion and exclusion criteria will be used. Inclusion criteria are (1) age \geq 18 years, (2) admitted to hospital with stroke (diagnosed by stroke physicians) during the study months, (3) able to provide informed consent or patient's personal consultee agrees to study participation. Exclusion criteria include (1) age

<18 years, (2) patients with pre-existing diagnosis of dementia (for PROM component only).

The Anglia Stroke Network was funded through the NHS Improvement Programme, following the publication of the National Stroke Strategy in December 2007. The Network was established in April 2008 to support the development of stroke services in Norfolk, Suffolk and Cambridgeshire regions. Since its inception, the Network regularly collected data to capture clinical service activities of the eight acute hospital trusts in the Network for the purpose of monitoring of services benchmarked by National targets and guidance from National Institute of Health & Clinical Excellence (NICE) in England and Wales. Data collection commenced in January 2009 and involves the individual trusts collecting clinical data which is fed back to the network by monthly reports. The total number of strokes admitted to the 8 acute trusts within the Network is approximately 4,000 per annum in 2009. The stroke cases were identified prospectively data were collected by the clinical team who looked after the patients and anonymised raw clinical data were sent to the network on monthly basis. The network collates and analyses the data for above mentioned purposes.

**Sample size**

Since this is an exploratory study designed to provide information for further analytic research, sample size will be determined partly pragmatically rather than on particular hypothesis tests. For illustration purposes, a total sample of 2264 patients would provide 80% power to detect a constant Hazard ratio (HR) of 0.76 for one-year mortality between two groups of roughly equal size, based on the log-rank test. This assumes a 20% one-year mortality rate in the reference group, no loss to follow-up before one year and 2-sided type I error of 5%. If one-year mortality is 30%, then 2264 patients would provide 76% power to detect a HR of 0.81.

**Plan of investigation**

The study will have a cohort design. We will follow up a cohort of patients systematically selected from each trust. For pragmatic purposes we will sample all patients who are admitted every third month, starting from October 2009. Over one calendar month, there will be ~ 300-350 stroke cases entered into the Network Clinical Data. Between October 2009 and September 2011, the Clinical Network would have collected a total of eight 3-monthly datasets per trust (i.e. 8 study months in total: Oct 2009, Jan 2010, April 2010, July 2010, October 2010, Jan 2011, April 2011, July 2011). Therefore, the estimated total cohort size with baseline clinical data will be ~ 2,400 stroke cases

during this exercise (30% of 4000 patients admitted annually in 8 trusts = 1200 × 2 yrs).

We will collect patient data by hospital trusts and conduct a questionnaire survey of patients' outcomes. Due to the nature of the study we would need 100% follow-up in randomly selected populations. Because we will be using a partially historical cohort, to avoid selection bias for mortality outcome, informed consent from all eligible participants will not be feasible. Therefore, it is most appropriate for the clinical team to collect the outcome data to comply with current ethical guidance in the UK. Therefore, the identifiable patient data will only be held at the local NHS trusts.

Neither the network nor the investigators will have access to any identifiable patient information (e.g. name, address). For outcome data we will utilise death certificate and hospital episode data from the Patient Administrative System (PAS) as described previously [14,15]. This approach will be used in conjunction with telephone and postal follow-up for questionnaire surveys such as EQ-5 D, and Stroke Impact Scale. These data will be counter-checked using discharge coding records, which record each hospital episode.

The clinical teams will retrieve case records to collect (1) baseline measures which were not recorded in baseline Network surveys and (2) outcome measures including mortality and hospital length of stay. At study commencement (October 2010) one year follow up data can be collected immediately for October 2009 cohorts (follow up complete at end September 2010). The follow up will be completed in September 2012 as the stroke patients included in the last survey for the study conducted by the Network in July 2011 will complete one year follow-up in June 2012 and data collection of the study will be completed by July-August 2012 with the view of final cohort data arrival to research team by the end of December 2012.

Due to multi-centre nature of the study the individual sites are expected to join the study at different time points (after their respective NHS Research & Development Committees' approval). We will collect characteristics of stroke services, patient related factors, prognostic indicators, treatment options and trial/study participation. Missing prognostic data will be imputed statistically, to ensure that all eligible patients are included in the primary analysis (see also Statistical Methods).

The service characteristics of interest include:

At hospital level

- staffing (including junior doctors and therapists (whole time equivalent), physicians characteristics
 - university or district general hospital
 - distance from tertiary referral centre

- availability of vascular surgery on site, neuro-surgery and neuro ITU on site
- monitoring beds
- physician on call rota
- compliance with NICE guidelines

At patient level

- provision of thrombolysis and CT
- medication

Outcome measurements

Primary outcome of the study will be one year mortality comparison between services with different characteristics. The secondary outcomes will include (1) final discharge destination (good or poor outcome) [16], (2) length of acute hospital stay, (3) length of stay in rehabilitation, (4) complications during acute and rehab-hospital stay and significant procedures (e.g. aspiration pneumonia, myocardial infarction), (5) readmissions, (6) composite cardiovascular events (recurrent TIA/Stroke/Acute Coronary Syndrome, Myocardial infarction).

Patient Reported Outcome Measures (PROM)

PROM will consist of (1) Stroke Impact Scale, (2) health related quality of life: EQ-5 D at one year in those who completed questionnaire at the baseline, (3) modified RANKIN, (4) Barthel score and (5) health service use.

Statistical analysis

Quantitative data will be analysed by multivariate Cox-proportional hazards to examine the relationships between different aspects of health services and time to death, adjusting for prognostic characteristics. Multiple logistic or linear regression models will be constructed as appropriate for dichotomised and continuous outcome variables respectively. T tests for normally distributed data and Mann-Whitney U tests for non-normally distributed data will be used to compare continuous outcomes. Volume-outcome relationships will be investigated. Missing prognostic and EQ-5 D data will be imputed, based on each patient's other prognostic characteristics. Clustering of data by hospital trust will be investigated and, if necessary, taken into account, and intra-class correlation coefficients calculated to inform future research.

Economic evaluation

Health care resources are scarce and it is therefore important to ensure that evaluations are undertaken in order to ensure that services provided by the NHS constitute value for money. Within this study we will thereby seek to estimate the cost-effectiveness of different stroke service deliveries.

Costs will first be calculated from the perspective of the NHS and personal social services (PSS). Thus, levels

of resources use will be recorded during the follow-up period, including the length of original hospital stay, input by the multi-disciplinary team, other investigations (e.g. x-ray) and any complications (including details of any further hospital admissions). Unit costs will subsequently be assigned to each of these resource items, enabling both the total mean cost in participants and the incremental cost between two different service deliveries (chosen to compare the cost effectiveness, e.g. traditional on call rota vs. telemedicine) to be calculated after adjusting for other factors. The main measure of effectiveness to be used in the economic analysis will be the EQ-5 D [17], where responses will be sought at baseline, and at 12 month as mentioned above. This will enable the overall effect of each mode of service delivery, and the incremental effect of services to be estimated.

Outcome

As the National Institute of Health and Clinical Excellence [18] recommends use of the EQ-5 D [17] within cost-effectiveness analysis this will be our primary measure within the economic analyses. EQ-5 D data will be collected at two University Hospitals and two district general hospitals within the clinical network. We will use “mapping” strategy to estimate the cost-effectiveness analyses across the region. The use of mapping, where scores from a condition-specific (non preference-based) measure are ‘converted’ into a utility (preference-based) score using a pre-defined formulae, has been advocated (in certain instances) by the UK National Institute of Health and Clinical Excellence (NICE) [18], and has been used to estimate the utility scores, and in turn cost-effectiveness, of a number of health care interventions [19]. Mapping presents the possibility of not asking all participants to complete the EQ-5 D. In this study we propose to take advantage of this by developing a mapping algorithm based on the response from participants participating in this component to predict the EQ-5 D for participants in retrospective cohorts and those who did not participate in PROM component.

Because the quality of life measure (EQ-5D) which can be used to estimate health utility and calculate QALYs (Quality Adjusted Life Years) for economic evaluation is outside the remit of routine data collection and cannot be done retrospectively, we will collect EQ-5 D data in only the second year of the study (October 2010 and January, April and July 2011 cohorts and one year follow up data to be collected September and December 2011, and March and June 2012) in those who provide informed consent to the study (we estimate that the sample will be approximately 15-20% of the whole sample after excluding the one year pre-study period (between October 2009-September 2010) and after

taking into account of refusal rate (estimated ~ 30%) in trusts with Stroke or Comprehensive Local Research Network Research Nurses.

Economic Analysis

In the Economic analysis if one option is shown to be less costly and more effective than another option (for example, telemedicine vs. on call system) then that option will ‘dominate’ the other and be deemed cost-effective. Alternatively, the incremental cost-effectiveness ratio (ICER) associated with a particular option will be estimated and assessed in relation to a range of cost-effectiveness thresholds. The associated level of uncertainty will also be characterised by e.g. estimating the cost-effectiveness acceptability curve (CEAC) for each intervention and conducting value of information analysis [20]. Sensitivity analysis will also be undertaken to assess the robustness of conclusions to key assumptions. We will also seek to identify what resource items should be monitored in a future study (i.e. what are the big cost drivers which are likely to be affected by the intervention) and how these items should be identified.

The study is funded by the NIHR Research for Patient Benefit Programme (PB-PG-1208-18240) and obtained ethical approval from the Norfolk Research Ethics Committee.

Discussion

In this study we specifically aim to identify services that are associated with the best clinical outcomes including mortality and hospital length of stay including patient reported outcome adjusting for patient prognostic factors and potential confounders. Our study will be able to provide useful information in stroke service provision in UK and beyond. Furthermore, inclusion of patient reported outcome is novel and exciting component of our study.

Studies which have examined the delivery of specific services such as rapid imaging, have shown improvement in patients’ outcome in stroke [21]. A recent report from Germany suggested that a telestroke network may be a useful strategy to implement in their non-urban stroke services [22]. Lees *et al* (2008) [23] highlighted that there is room for improvement in terms of acute services for stroke. Interestingly, one of the observations was that centres with higher workload performed better. There is also existing evidence in Cancer literature that centres with higher surgical caseload have better outcomes [24]. There has also been a recent evaluation of the impact on stroke outcome by evidence-based practice in an Australian setting [25]. Examples of service delivery that are associated with better outcomes include organised stroke unit care [26], thrombolysis treatment and appropriate secondary prevention [27], and early supported discharge

in selected patients [28,29]. However, the cost-effectiveness of such services has yet to be fully examined.

Rodgers et al [30] highlighted the need for improvement in hospital-based stroke services e.g. stroke unit staffing levels were lower than was available in RCTs. The accumulating body of evidence has been a major driving force behind the UK Government's strategy to improve stroke care (National Stroke Strategy, 2007) [31]. A key strand of the strategy was to set up stroke networks to deliver stroke service development across geographically defined areas. The stroke networks have worked to agree minimum standards for stroke care and they have worked with commissioners to assist the commissioning process for stroke services. The acute stroke services are currently delivered by different NHS trusts and there is therefore a wide range of inequality in service availability and provision with differing structure and local support systems.

This research aims to utilise NHS data in the most meaningful and innovative way and we aim to maximize the benefit with minimum investment to produce best research output for patient care by collaborating with clinical teams and the network in providing excellent value for money. This observational study seeks to identify areas of clinical practice which merit future randomised controlled trials (RCTs) to identify best practice in improving stroke care which will be of maximum benefit to patients. We also aim to obtain preliminary data to estimate sample sizes and conduct value of information analyses to design future pragmatic RCTs of innovative ways of delivering stroke care.

As we include eight diverse NHS trusts, the findings are likely to be generalisable in the UK setting and beyond. This study will provide observational data about health service factors associated with variations in patient outcomes and health care costs following hospital admission for acute stroke. This will form the basis for future RCTs by identifying promising health service interventions, assessing the feasibility of recruiting and following up trial patients, and provide evidence about frequency and variances in outcomes, and intra-cluster correlation of outcomes, for sample size calculations. The results will also inform clinicians, public, service providers, commissioners and policy makers to drive further improvement in health services and bring direct benefit to patients.

The study will describe the variation in outcomes between different stroke services, and identify the characteristics of services associated with better outcomes after accounting for case-mix. We will also estimate the relative costs of and health gain estimated as Quality Adjusted Life Year (QALY) gain that may be demonstrated by different services. The commissioners of services will be informed as to which service delivery

structures are likely to provide value for money to make purchasing decisions. They will also be better informed about the types of service associated with better patient reported outcome. Hospital trusts will be able to evaluate their services systematically and plan their care appropriately to meet local and regional needs and demands based on our study findings. Professionals will be able to reflect on the impact of services they are delivering to help improve their performance and the way services are organised by adopting the most effective and cost effective approaches. As an observational study, the study limitations include inability to control for unknown confounders and residual confounding effect of known confounders which are adjusted for. The causal relationship cannot be implied but as we stated the findings will provide knowledge about areas that requires further evaluation in clinical trial setting.

There is very little work which assesses service provision robustly against patients' own reported outcomes. This exciting study may lead to a clearer drive for patients to define what makes a good service. We hope that the best clinical practices are adopted to suit the local populations' needs and demand. As we included eight diverse NHS trusts, the findings will be generalisable in the UK setting and likely to be applicable in international setting. All these will become drivers of improvement in stroke services for the benefit of stroke sufferers.

Acknowledgements

We would like to thank the participants of the study. We gratefully acknowledge the contribution of Stroke Research Nurses and the study steering committee members including representatives from the Regional Stroke Association, and Patient and Public Involvement in Research Panel. We would like to thank our colleagues from all participating trusts, site data co-ordinator and staff from Anglia Stroke & Heart Network for their assistance. Norfolk and Norwich University Hospital NHS Foundation Trust sponsors the study.

Manuscripts that are under submission based on this protocol

None.

Author details

¹Norwich Medical School, Faculty of Medicine & Health Sciences, Norwich, UK. ²Norfolk & Norwich University Hospital, Norwich, UK. ³Anglia Stroke & Heart Clinical Network, Cambridge, UK. ⁴James Paget University Hospital, Lowestoft, UK. ⁵Queen Elizabeth Hospital, King's Lynn, UK. ⁶Peterborough City Hospital, Peterborough, UK. ⁷Hinchingbrooke Hospital, Huntingdon, UK. ⁸Ipswich Hospital, Ipswich, UK. ⁹West Suffolk Hospital, Bury St Edmund, UK. ¹⁰Addenbrooke's University Hospital, Cambridge, UK.

Authors' contributions

PKM, DJD, MOB designed the outline of the study. PKM, JFP, MOB, EAW, GMP, GAB and AKM obtained the funding for the study. SDM & RH contributed in protocol preparation. All authors contributed in writing of the paper. All authors read and approved the final manuscript. PKM is the guarantor.

Competing interests

The authors declare that they have no competing interests.

Received: 21 January 2011 Accepted: 28 February 2011

Published: 28 February 2011

References

- National Institute for Health and Clinical Excellence: **STROKE: National clinical guideline for diagnosis and initial management of acute stroke and transient ischaemic attack (TIA). The National Collaborative Centre for Chronic Conditions.** *Royal College of Physicians of London* 2008 [<http://www.nice.org.uk/nicemedia/pdf/CG68FullGuideline.pdf>], ISBN 978-1-86016-339-5.
- National Audit Office: **Reducing brain damage: faster access to better stroke care.** London: Stationery Office, 2005 National Audit Office Report; 2005.
- Rothwell PM, Coull AJ, Silver LE, Fairhead JF, Giles MF, Lovelock CE, Redgrave JN, Bull LM, Welch SJ, Cuthbertson FC, Binney LE, Gutnikov SA, Anslow P, Banning AP, Mant D, Mehta Z, Oxford Vascular Study: **Population-based study of event-rate, incidence, case fatality and mortality for all acute vascular events in all arterial territories (Oxford Vascular Study).** *Lancet* 2005, **366**:1773-83.
- Grieve R, Hutton J, Bhalla A, Rastenyte D, Ryglewicz D, Sarti C, Lamassa M, Giroud M, Dundas R, Wolfe CD: **A comparison of the costs and survival of hospital-admitted stroke patients across Europe.** *Stroke* 2001, **32**:1684-91.
- Weir NU, Sandercock PA, Lewis SC, Signorini DF, Warlow CP: **Variations between countries in outcome after stroke in the International Stroke Trial (IST).** *Stroke* 2001, **32**:1370-7.
- Gray LJ, Sprigg N, Bath PM, Sørensen P, Lindenstrøm E, Boysen G, De Deyn PP, Friis P, Leys D, Marttila R, Olsson JE, O'Neill D, Ringelstein B, van der Sande JJ, Turpie AG, TAIST Investigators: **Significant variation in mortality and functional outcome after acute ischaemic stroke between Western countries: data from the tinzaparin in acute ischaemic stroke trial (TAIST).** *J Neurol Neurosurg Psychiatry* 2006, **77**:327-33.
- Markus H: **Improving the outcome of stroke.** *BMJ* 2007, **335**:359-60.
- Rudd AG, Irwin P, Rutledge Z, Lowe D, Wade DT, Pearson M: **Regional variations in stroke care in England, Wales and Northern Ireland: results from the National Sentinel Audit of Stroke.** Royal College of Physicians Intercollegiate Stroke Working Party. *Clin Rehabil* 2001, **15**:562-72.
- Rudd AG, Lowe D, Hoffman A, Irwin P, Pearson M: **Secondary prevention for stroke in the United Kingdom: results from the National Sentinel Audit of Stroke.** *Age Ageing* 2004, **33**:280-6.
- Rudd AG, Hoffman A, Irwin P, Pearson M, Lowe D, Intercollegiate Working Party for Stroke: **Stroke units: research and reality. Results from the National Sentinel Audit of Stroke.** *Qual Saf Health Care* 2005, **14**:7-12.
- Howell E, Graham C, Hoffman A, Lowe D, McKeivitt C, Reeves R, Rudd AG: **Comparison of patients' assessments of the quality of stroke care with audit findings.** *Qual Saf Health Care* 2007, **16**:450-5.
- National Sentinel Stroke Audit: **Phase II (clinical audit), 2008. Report for England, Wales and Northern Ireland.** Intercollegiate Stroke Working Party. *Clinical Effectiveness and Evaluation Unit, Royal College of Physicians of London* 2009.
- Myint PK, Vowler SL, Woodhouse PR, Redmayne O, Fulcher RA: **Winter excess in hospital admissions, in-patient mortality and length of acute hospital stay in stroke: a hospital database study over six seasonal years in Norfolk, UK.** *Neuroepidemiology* 2007, **28**:79-85.
- Myint PK, Kamath AV, Vowler SL, Maisey DN, Harrison BD: **The CURB (confusion, urea, respiratory rate and blood pressure) criteria in community-acquired pneumonia (CAP) in hospitalised elderly patients aged 65 years and over: a prospective observational cohort study.** *Age Ageing* 2005, **34**:75-7.
- Myint PK, Vowler SL, Redmayne O, Fulcher RA: **Utilisation of diagnostic computerised tomography imaging and immediate clinical outcomes in older people with stroke before and after introduction of the National Service Framework for older people. A comparative study of hospital-based stroke registry data (1997-2003): Norfolk experience.** *Age Ageing* 2006, **35**:399-403.
- Myint PK, Vowler SL, Redmayne O, Fulcher RA: **Cognition, continence and transfer status at the time of discharge from an acute hospital setting and their associations with an unfavourable discharge outcome after stroke.** *Gerontology* 2008, **54**:202-9.
- Brooks R: **EuroQol: the current state of play.** *Health Policy* 1996, **37**:53-72.
- National Institute for Clinical Excellence: 2004 [http://www.nice.org.uk/aboutnice/howwework/devicetech/technologyappraisalprocessguides/guide_to_the_methods_of_technology_appraisal_reference_n0515.jsp], (reference N0515). ISBN: 1-84257-595-3.
- Barton GR, Sach TH, Jenkinson C, Avery AJ, Doherty M, Muir KR: **Do estimates of cost-utility based on the EQ-5 D differ from those based on the mapping of utility scores?** *Health Qual Life Outcomes* 2008, **6**:51.
- Barton GR, Briggs AH, Fenwick EA: **Optimal cost-effective decisions: the role of the cost-effectiveness acceptability curve (CEAC), cost-effectiveness acceptability frontier (CEAF) and expected value of perfect information (EVPI).** *Value Health* 2008, **11**:886-897.
- Wardlaw JM, Seymour J, Cairns J, Keir S, Lewis S, Sandercock P: **Immediate computed tomography scanning of acute stroke is cost-effective and improves quality of life.** *Stroke* 2004, **35**:2477-83.
- Audebert HJ, Kukla C, Vatankeh B, Gotzler B, Schenkel J, Hofer S, Fürst A, Haberl RL: **Comparison of tissue plasminogen activator administration management between Telestroke Network hospitals and academic stroke centers: the Telemedical Pilot Project for Integrative Stroke Care in Bavaria/Germany.** *Stroke* 2006, **37**:1822-7.
- Lees KR, Ford GA, Muir KW, Ahmed N, Dyker AG, Atula S, Kalra L, Warburton EA, Baron JC, Jenkinson DF, Wahlgren NG, Walters MR, SITS-UK Group: **Thrombolytic therapy for acute stroke in the United Kingdom: experience from the safe implementation of thrombolysis in stroke (SITS) register.** *QJM* 2008, **101**:863-9.
- Bachmann MO, Alderson D, Edwards D, Wotton S, Bedford C, Peters TJ, Harvey IM: **Cohort study in South and West England of the influence of specialization on the management and outcome of patients with oesophageal and gastric cancers.** *Br J Surg* 2002, **89**:914-22.
- Gattellari M, Worthington J, Jalaludin B, Mohsin M: **Stroke unit care in a real-life setting: can results from randomized controlled trials be translated into every-day clinical practice? An observational study of hospital data in a large Australian population.** *Stroke* 2009, **40**:10-7.
- Stroke Unit Trialists' Collaboration: **Organised inpatient (stroke unit) care for stroke.** *Cochrane Database Syst Rev* 2007, **4**: CD000197, Review.
- Higgins P, Lees KR: **Advances in emerging therapies.** *Stroke* 2009, **40**(5): e292-4.
- Langhorne P, Taylor G, Murray G, Dennis M, Anderson C, Bautz-Holter E, Dey P, Indredavik B, Mayo N, Power M, Rodgers H, Ronning OM, Rudd A, Suwanwela N, Widen-Holmqvist L, Wolfe C: **Early supported discharge services for stroke patients: a meta-analysis of individual patients' data.** *Lancet* 2005, **365**:501-6.
- Early Supported Discharge Trialists: **Services for reducing duration of hospital care for acute stroke patients.** *Cochrane Database Syst Rev* 2005, **2**: CD000443.
- Rodgers H, Dennis M, Cohen D, Rudd A, British Association of Stroke Physicians: **British Association of Stroke Physicians: benchmarking survey of stroke services.** *Age Ageing* 2003, **32**:211-7.
- National Stroke Strategy: **Department of Health. 2007** [http://www.dh.gov.uk/en/Publicationsandstatistics/Publications/PublicationsPolicyandGuidance/DH_081062].

Pre-publication history

The pre-publication history for this paper can be accessed here:
<http://www.biomedcentral.com/1472-6963/11/50/prepub>

doi:10.1186/1472-6963-11-50

Cite this article as: Myint *et al.*: Evaluation of stroke services in Anglia stroke clinical network to examine the variation in acute services and stroke outcomes. *BMC Health Services Research* 2011 **11**:50.

Submit your next manuscript to BioMed Central and take full advantage of:

- Convenient online submission
- Thorough peer review
- No space constraints or color figure charges
- Immediate publication on acceptance
- Inclusion in PubMed, CAS, Scopus and Google Scholar
- Research which is freely available for redistribution

Submit your manuscript at
www.biomedcentral.com/submit

Supplementary document to:

**Does service heterogeneity have an impact on acute hospital length of stay in stroke? A UK-based**
**multi-centre prospective cohort study**

Michelle Tørnes, David McLernon, Max O Bachmann, Stanley D Musgrave, Elizabeth A
Warburton, John F Potter, Phyo Kyaw Myint: On behalf of the Anglia Stroke Clinical
Network Evaluation Study (ASCNES) Group

For peer review only

CONTENT

Table S1 Variables used to inform multiple imputation of missing data

Table S2 Sample characteristics distribution of the 2333 patients included in analysis per individual hospital (n (%)) unless otherwise stated)

Table S3 Sample characteristics of complete cases and those with at least one variable missing

Table S4 Univariable regression analysis for multiple imputed dataset for AHLOS (n=2233)

Table S5 Univariable linear regression complete case analysis for AHLOS

Table S6 Multiple linear regression complete case analysis for AHLOS (n=1496, $R^2=44.7\%$)

Table S7 Multiple linear regression sensitivity analysis for AHLOS, excluding Hospital 2 using multiple imputed dataset (n=2217, $R^2=44.7\%$)

Table S8 Multiple linear regression sensitivity analysis for AHLOS, including discharge destination using multiple imputed dataset (n=2233, $R^2=40\%$)

Figure S1 Model estimates of mean baseline acute hospital length of stay (AHLOS) per hospital (in days) and number of stroke patients treated outside the stroke unit per day per five stroke unit beds with 95% confidence intervals. Multiple regression model was adjusted for patient covariates that had a p-value<0.3 in univariable analysis.

Figure S2 Model estimates of mean baseline acute hospital length of stay (AHLOS) per hospital (in days) and presence of vascular surgery onsite with 95% confidence intervals. Multiple regression model was adjusted for patient covariates that had a p-value<0.3 in univariable analysis.

**Figure S3** Model estimates of mean baseline acute hospital length of stay (AHLOS) per
hospital (in days) and distance to neurosurgical facility with 95% confidence intervals.
Multiple regression model was adjusted for patient covariates that had a p-value<0.3 in
univariable analysis.

**Figure S4** Model estimates of mean baseline acute hospital length of stay (AHLOS) per
hospital (in days) and number of fte senior doctors per five beds available during weekdays
with 95% confidence intervals. Multiple regression model was adjusted for patient covariates
that had a p-value<0.3 in univariable analysis.

**Figure S5** Model estimates of mean baseline acute hospital length of stay (AHLOS) per
hospital (in days) and number of fte junior doctors per five beds available during weekdays
with 95% confidence intervals. Multiple regression model was adjusted for patient covariates
that had a p-value<0.3 in univariable analysis.

**Figure S6** Model estimates of mean baseline acute hospital length of stay (AHLOS) per
hospital (in days) and number of fte health care associates and nurses per five beds with 95%
confidence intervals. Multiple regression model was adjusted for patient covariates that had a
p-value<0.3 in univariable analysis.

**Figure S7** Model estimates of mean baseline acute hospital length of stay (AHLOS) per
hospital (in days) and number of fte occupational therapists per five beds with 95%
confidence intervals. Multiple regression model was adjusted for patient covariates that had a
p-value<0.3 in univariable analysis.

**Figure S8** Model estimates of mean baseline acute hospital length of stay (AHLOS) per
hospital (in days) and number of fte physiotherapists per five beds with 95% confidence
intervals. Multiple regression model was adjusted for patient covariates that had a p-
value<0.3 in univariable analysis.

**Figure S9** Model estimates of mean baseline acute hospital length of stay (AHLOS) per
hospital (in days) and number of fte speech and language therapists per five beds with 95%
confidence intervals. Multiple regression model was adjusted for patient covariates that had a
p-value<0.3 in univariable analysis.

**Figure S10** Model estimates of mean baseline acute hospital length of stay (AHLOS) per
hospital (in days) and number of total beds present on stroke unit per 100 admissions with
95% confidence intervals. Multiple regression model was adjusted for patient covariates that
had a p-value<0.3 in univariable analysis.

**Figure S11** Model estimates of mean baseline acute hospital length of stay (AHLOS) per
hospital (in days) and number of hospital beds per computed tomography (CT) scanner with
95% confidence intervals. Multiple regression model was adjusted for patient covariates that
had a p-value<0.3 in univariable analysis.

**Figure S12** Model estimates of mean baseline acute hospital length of stay (AHLOS) per
hospital (in days) and provision of onsite rehabilitation service with 95% confidence
intervals. Multiple regression model was adjusted for patient covariates that had a p-
value<0.3 in univariable analysis.

**Figure S13** Model estimates of mean baseline acute hospital length of stay (AHLOS) per
hospital (in days) and presence of early supported discharge scheme with 95% confidence
intervals. Multiple regression model was adjusted for patient covariates that had a p-
value<0.3 in univariable analysis.

**Figure S14** Model estimates of mean baseline acute hospital length of stay (AHLOS) per
hospital (in days) and number of non-stroke patients present on the stroke unit per day per
five stroke unit beds with 95% confidence intervals. Multiple regression model was adjusted
for patient covariates that had a p-value<0.3 in univariable analysis.

**Figure S15** Model estimates of mean baseline acute hospital length of stay (AHLOS) per
hospital (in days) and mean Index of Multiple Deprivation (IMD) score of the counties in
which the hospital serves with 95% confidence intervals. Multiple regression model was
adjusted for patient covariates that had a p-value<0.3 in univariable analysis.

For peer review only

Table S1 Variables used to inform multiple imputation of missing data

Variable	Measure
I. Independent Variables	
Trust	0=Trust 1 1=Trust 2 2=Trust 3 3=Trust 4 4=Trust 5 4=Trust 6 5=Trust 7 6=Trust 8
Sex	0=Male 1=Female
Age	Continuous, years
Recurrent Stroke	0=No 1=Yes
Diabetes Mellitus	0=No 1=Yes
Dementia	0=No 1=Yes
Hypercholesterolemia	0=No 1=Yes
Myocardial Infarction or Ischaemic Heart Disease	0=No 1=Yes
Transient Ischaemic Attack	0=No 1=Yes
Previous Cancer	0=No 1=Yes
Active Cancer	0=No 1=Yes
Depression	0=No 1=Yes
Rheumatoid Arthritis	0=No 1=Yes
Chronic Obstructive Pulmonary Disease	0=No 1=Yes
Pre-Stroke modified Rankin Score (mRS)	0=0 1=1 2=2 3=3 4=4 & 5
Pre-Stroke Residence	0=Independent living without formal care 1=Independent living with formal care 2=Institutional care
Stroke Type	0=Ischaemic 1=Haemorrhagic
Oxfordshire Community Stroke Classification	0=LACS 1=PACS 2=POCS 3=TACS
Brain Lateralisation	0=Yes 1=No
Inpatient Complication	0=No 1=Yes
Discharge modified Rankin Score (mRS)	0=0 1=1 2=2 3=3 4=4 5=5 6=6
Season of Admission	0=Summer 1=Winter
Day of Admission	0=Weekday 1=Weekend
II. Dependent Variable	
Logarithmic acute hospital LOS	Continuous, days
III. Auxiliary Variables	
Discharge Destination	0=Independent living without formal care 1=Independent living with formal care 2=Institutional care 3=Interim or rehabilitation setting 4=Death
Atrial Fibrillation	0=No 1=Yes
Baseline Systolic Blood Pressure	Continuous, mmHg
Baseline Diastolic Blood Pressure	Continuous, mmHg
Glucose Concentration on Admission	Continuous, mmol/L
Weight	Continuous, kg
Heart Rate	Continuous, beats per minute
Temperature	Continuous, °C
Oxygen Saturation	Continuous, %
ITU or HDU admission	0. No 1. Yes
Systolic Blood Pressure at Discharge	Continuous, mmHg
Diastolic Blood Pressure at Discharge	Continuous, mm Hg

LOS, Length of Stay; ITU, Intensive Care Unit; HDU, High Dependency Unit.

Table S2 Sample characteristics of the 2333 patients included in analysis per individual hospital (n (%)) unless otherwise stated)

Variables	Hospital 1	Hospital 2	Hospital 3	Hospital 4	Hospital 5	Hospital 6	Hospital 7	Hospital 8
	350 (16)	16 (1)	350 (16)	143 (6)	618 (28)	281 (13)	252 (11)	223 (10)
Age, y, median (IQR)	78 (68 to 85)	87 (81 to 92)	79 (72 to 86)	79 (70 to 86)	79 (71 to 85)	78 (71 to 85)	80 (68 to 85)	80 (71 to 87)
Sex, female	180 (52)	9 (56)	197 (56)	76 (53)	309 (50)	155 (55)	116 (46)	123 (55)
Recurrent Stroke	50 (14)	5 (31)	61 (17)	19 (17)	143 (23)	62 (22)	66 (26)	42 (19)
Diabetes Mellitus	48 (14)	1 (6)	59 (17)	17 (15)	92 (15)	66 (23)	44 (17)	43 (19)
Dementia	26 (7)	1 (6)	35 (10)	10 (9)	58 (9)	29 (10)	23 (9)	25 (11)
Hypercholesterolemia	48 (14)	3 (19)	24 (7)	7 (6)	61 (10)	80 (28)	38 (15)	94 (42)
Hypertensive	225 (64)	8 (50)	202 (58)	56 (50)	446 (72)	200 (71)	187 (74)	159 (71)
Myocardial Infarction or Ischaemic Heart Disease	45 (13)	3 (19)	87 (25)	30 (27)	142 (23)	80 (28)	49 (19)	81 (36)
Transient Ischaemic Attack	32 (9)	3 (19)	58 (17)	17 (15)	113 (18)	40 (14)	47 (19)	30 (13)
Previous Cancer	33 (9)	1 (6)	38 (11)	12 (11)	41 (7)	18 (6)	21 (8)	31 (14)
Active Cancer	24 (7)	2 (12)	8 (2)	10 (9)	49 (8)	9 (3)	20 (8)	15 (7)
Depression	13 (4)	0 (0)	17 (5)	8 (7)	33 (5)	11 (4)	18 (7)	17 (8)
Rheumatoid Arthritis	11 (3)	1 (6)	43 (12)	3 (3)	83 (13)	2 (1)	7 (3)	4 (2)
COPD	15 (4)	1 (6)	20 (6)	6 (5)	26 (4)	20 (7)	11 (4)	17 (8)
Pre-stroke mRS Score								
0	84 (43)	3 (19)	117 (36)	-	330 (56)	126 (64)	136 (56)	118 (53)
1	60(31)	3 (19)	75 (23)	-	87 (15)	16 (8)	61 (25)	33 (15)
2	24 (12)	3 (19)	51 (16)	-	56 (9)	17 (9)	16 (7)	24 (11)
3	21 (11)	2 (12)	38 (12)	-	60 (10)	20 (10)	15 (6)	28 (13)
4 & 5	7 (4)	5 (31)	44 (14)	-	57 (10)	18 (9)	16 (7)	20 (9)
Pre-Stroke Residence								
Independent living with formal care	21 (6)	4 (25)	23 (7)	15 (14)	62 (10)	30 (11)	34 (13)	21 (10)
Independent living w/o formal care	292 (86)	9 (56)	285 (82)	86 (77)	493 (80)	215 (77)	193 (77)	179 (82)
Institution	28 (8)	3 (19)	40 (11)	10 (9)	63 (10)	35 (12)	23 (9)	18 (8)

Variables	Hospital 1	Hospital 2	Hospital 3	Hospital 4	Hospital 5	Hospital 6	Hospital 7	Hospital 8
	350 (16)	16 (1)	350 (16)	143 (6)	618 (28)	281 (13)	252 (11)	223 (10)
Stroke Type								
Ischaemic	293 (85)	14 (100)	286 (87)	90 (91)	541 (88)	233 (85)	213 (87)	194 (88)
Haemorrhagic	50 (15)	0 (0)	43 (13)	9 (9)	73 (12)	40 (15)	32 (13)	26 (12)
Oxford Community Stroke Project Classification								
LACS	64 (24)	1 (7)	95 (29)	20 (28)	149 (25)	51 (18)	39 (19)	84 (39)
PACS	117 (43)	11 (79)	109 (33)	38 (54)	216 (37)	147 (53)	80 (39)	66 (30)
POCS	51 (19)	-	29 (9)	3 (4)	117 (20)	21 (8)	33 (16)	25 (12)
TACS	38 (14)	2 (14)	99 (30)	10 (14)	107 (18)	57 (21)	52 (25)	42 (19)
No Brain Lateralisation	50 (15)	2 (13)	14 (4)	9 (9)	129 (21)	1 (0.4)	30 (12)	9 (4)
Inpatient Complication	108 (31)	4 (25)	34 (10)	36 (25)	229 (37)	109 (39)	83 (33)	52 (23)
Discharge mRS Score								
0	37 (15)	0 (0)	11 (3)	0 (0)	114 (19)	34 (16)	42 (17)	22 (10)
1	65 (25)	2 (12)	55 (17)	0 (0)	97 (16)	25 (12)	55 (23)	53 (24)
2	36 (14)	1 (6)	46 (14)	0 (0)	57 (10)	20 (9)	33 (14)	19 (9)
3	41 (16)	4 (25)	40 (12)	0 (0)	87 (15)	36 (17)	34 (14)	49 (22)
4	19 (7)	3 (19)	57 (17)	0 (0)	89 (15)	25 (12)	16 (7)	29 (13)
5	4 (2)	1 (6)	47 (14)	0 (0)	40 (7)	14 (7)	16 (7)	15 (7)
6	53 (21)	5 (31)	77 (23)	29 (100)	110 (19)	58 (27)	47 (19)	35 (16)
Winter Admission	172 (49)	16 (100)	181 (52)	73 (51)	332 (54)	140 (50)	131 (52)	114 (51)
Weekend Admission	113 (32)	3 (19)	98 (28)	43 (30)	177 (29)	74 (26)	55 (22)	51 (23)

IQR, Interquartile Range; COPD, Chronic Obstructive Pulmonary Disease; mRS, modified Rankin Scale; LACS, Lacunar Anterior Circulation Stroke; PACS, Partial Anterior Circulation Stroke; POCS, Posterior Circulation Stroke; TACS, Total Anterior Circulation Stroke.

Table S3 Sample characteristics of complete cases and those with at least one variable missing

Patient Characteristic	Complete Cases (n=1496)	Cases with at least one missing variable	P
	Median (IQR) or No. (%)		
Age, y*	79 (71 to 86)	79 (70 to 86)	0.34
Sex, female†	781 (52)	384 (52)	1
Comorbidities†			
Recurrent Stroke	328 (22)	120 (17)	0.01
Diabetes Mellitus	259 (17)	111 (16)	0.38
Dementia	138 (9)	69 (10)	0.75
Hypercholesterolemia	264 (18)	91 (13)	0.01
Hypertensive	1054 (70)	429 (61)	<0.001
Myocardial Infarction or Ischaemic Heart Disease	362(24)	155 (22)	0.26
TIA	248 (17)	92 (13)	0.04
Previous Cancer	140 (9)	55 (8)	0.25
Active Cancer	93 (6)	44 (6)	1
Depression	79 (5)	38 (5)	1
Rheumatoid Arthritis	129 (9)	25 (3)	<0.001
COPD	76 (5)	40 (6)	0.64
Pre-stroke mRS Score‡			0.62
0	765 (51)	149 (51)	
1	284 (19)	51 (17)	
2	167 (11)	24 (8)	
3	149 (10)	35 (12)	
4 & 5	131 (9)	36 (12)	
Pre-stroke Residence†			<0.001
Independent living with formal care	145 (10)	65 (9)	
Independent living without formal Institution	1215 (81)	537 (78)	
(9)	84 (12)
Haemorrhagic Stroke†	138 (9)	135 (21)	<0.001
Oxford Community Stroke Project‡			0.05
LACS	411 (27)	92 (19)	
PACS	570 (38)	214 (45)	
POCS	214 (14)	65 (14)	
TACS	301 (20)	106 (22)	
No Brain Lateralisation†	174 (12)	70 (12)	0.74
Inpatient Complication†	421 (28)	234 (32)	0.09
Discharge mRS Score‡			0.02
0	218 (15)	42 (10)	
1	295 (20)	57 (14)	
2	177 (12)	35 (9)	
3	243 (16)	48 (12)	
4	209 (14)	29 (7)	
5	121 (8)	16 (4)	
6	233 (16)	181 (44)	

Winter Admission†	770 (51)	389 (53)	0.59
Weekend Admission‡	401 (27)	213 (29)	0.32

IQR, Interquartile Range; TIA, Transient Ischaemic Attack; COPD, Chronic Obstructive Pulmonary Disease; mRS, modified Rankin Scale; LACS, Lacunar Anterior Circulation Stroke; PACS, Partial Anterior Circulation Stroke; POCS, Posterior Circulation Stroke; TACS, Total Anterior Circulation Stroke.

*Two sample *t*-test

† χ^2 test

‡ χ^2 test for trend

For peer review only

Table S4 Univariable regression analysis for multiple imputed dataset for AHLOS (n=2233)

Patient Characteristic	β	95% CI	P	R ²
Age, y	1.02	1.02 to 1.02	<0.001	4.8
Sex, female	1.20	1.10 to 1.31	<0.001	0.7
Recurrent Stroke	1.17	1.05 to 1.31	0.01	0.4
Diabetes Mellitus	1.16	1.03 to 1.31	0.02	0.3
Dementia	1.46	1.25 to 1.70	<0.001	1.1
Hypercholesterolemia	0.84	0.75 to 0.95	0.01	0.3
Hypertensive	1.02	0.93 to 1.12	0.66	0
Myocardial Infarction/ Ischaemic Heart Disease*	1.07	0.96 to 1.19	0.23	0.1
TIA	1.07	0.94 to 1.21	0.30	0.1
Previous Cancer	1.23	1.05 to 1.44	0.01	0.3
Active Cancer	0.97	0.80 to 1.16	0.72	0
Depression	1.06	0.86 to 1.29	0.59	0
Rheumatoid Arthritis	1.10	0.92 to 1.31	0.31	0.1
COPD	0.86	0.71 to 1.06	0.15	0.1
Pre-stroke mRS Score (reference 0)			<0.001	5.5
1	1.57	1.38 to 1.79	<0.001	
2	1.63	1.39 to 1.91	<0.001	
3	1.94	1.65 to 2.28	<0.001	
4 & 5	1.32	1.13 to 1.55	<0.001	
Pre-stroke Residence (reference Independent living w/o formal care)			<0.001	1.4
Independent living with formal care	1.52	1.31 to 1.77	<0.001	
Institution	1.13	0.97 to 1.31	0.11	
Haemorrhagic Stroke	0.83	0.73 to 0.96	0.01	0.3
Oxford Community Stroke Project Classification (reference LACS)			<0.001	4.0
PACS	1.62	1.44 to 1.82	<0.001	
POCS	1.22	1.05 to 1.42	0.01	
TACS	1.66	1.45 to 1.90	<0.001	
Brain Lateralisation	0.69	0.60 to 0.80	<0.001	1.2
Inpatient Complication	2.13	1.94 to 2.34	<0.001	10.3
Discharge mRS Score (reference 0)			<0.001	31.1
1.24	1.07 to 1.42	0.003
2.04	1.75 to 2.39	<0.001
3.35	2.90 to 3.87	<0.001
4.20	3.60 to 4.90	<0.001
6.67	5.62 to 7.91	<0.001
1.57	1.37 to 1.80	<0.001
Winter Admission	1.20	1.09 to 1.31	<0.001	0.7
Weekend Admission	1.08	0.98 to 1.20	0.12	0.1
Hospital (reference 1)			<0.001	2.4
2.69	1.58 to 4.58	<0.001
1.19	1.02 to 1.39	0.03
1.24	1.01 to 1.53	0.04
0.86	0.75 to 0.99	0.03
1.11	0.94 to 1.31	0.22
1.18	1.00 to 1.41	0.05

Patient Characteristic	β	95% CI	P	R^2
0.86	0.72 to 1.03	0.11

AHLOS, Acute Hospital Length of Stay; CI, Confidence Interval; TIA, Transient Ischaemic Attack; COPD, Chronic Obstructive Pulmonary Disease; mRS, modified Rankin Scale; LACS, Lacunar Anterior Circulation Stroke; PACS, Partial Anterior Circulation Stroke; POCS, Posterior Circulation Stroke; TACS, Total Anterior Circulation Stroke.

For peer review only

Table S5 Univariable linear regression complete case analysis for AHLOS

Patient Characteristic	N	β	95% CI	P	% R ²
Age, y	2231	1.02	1.02 to 1.02	<0.001	4.7
Sex, female	1165 v. 1066	1.20	1.10 to 1.31	<0.001	0.7
Recurrent Stroke	448 v. 1755	1.17	1.05 to 1.31	0.005	0.3
Diabetes Mellitus	370 v. 1833	1.16	1.03 to 1.31	0.02	0.2
Dementia	207 v. 1996	1.46	1.25 to 1.70	<0.001	1.0
Hypercholesterolemia	355 v. 1848	0.85	0.75 to 0.95	0.01	0.3
Hypertensive	1483 v. 720	1.03	0.93 to 1.13	0.57	0
Myocardial Infarction or Ischaemic Heart Disease*	517 v. 1686	1.07	0.96 to 1.19	0.23	0
TIA	340 v. 1863	1.06	0.94 to 1.20	0.32	0
Previous Cancer	195 v. 2008	1.23	1.05 to 1.44	0.01	0.3
Active Cancer	137 v. 2066	0.96	0.80 to 1.15	0.65	0
Depression	117 v. 2086	1.05	0.86 to 1.28	0.65	0
Rheumatoid Arthritis	154 v. 2049	1.10	0.92 to 1.31	0.31	0
COPD	116 v. 2087	0.86	0.70 to 1.05	0.14	0.1
Pre-stroke mRS Score (reference 0)				<0.001	5.8
1	335 v. 914	1.58	1.39 to 1.80	<0.001	
2	191 v. 914	1.62	1.38 to 1.90	<0.001	
3	184 v. 914	1.97	1.67 to 2.31	<0.001	
4 & 5	167 v. 914	1.45	1.22 to 1.71	<0.001	
Pre-stroke Residence (reference Independent living without formal care)				<0.001	1.3
Independent living with formal care	210 v. 1752	1.52	1.31 to 1.77	<0.001	
Institution	220 v. 1752	1.14	0.98 to 1.32	0.09	
Haemorrhagic Stroke	273 v. 1864	0.85	0.74 to 0.97	0.02	0.2
Oxford Community Stroke Project Classification (reference LACS)				<0.001	4.3
PACS	784 v. 503	1.62	1.44 to 1.82	<0.001	
POCS	279 v. 503	1.24	1.06 to 1.44	0.01	
TACS	407 v. 503	1.75	1.53 to 2.01	<0.001	
No Brain Lateralisation	244 v. 1822	0.68	0.59 to 0.79	<0.001	1.3
Inpatient Complication	655 v. 1578	2.13	1.94 to 2.34	<0.001	10
Discharge mRS Score (reference 0)				<0.001	30.1
352 v. 260	1.25	1.08 to 1.44	0.002
v. 260	2.01	1.72 to 2.36	<0.001
291 v. 260	3.30	2.84 to 3.82	<0.001
v. 260	4.17	3.57 to 4.87	<0.001
v. 260	6.97	5.81 to 8.37	<0.001
414 v. 260	1.58	1.38 to 1.81	<0.001
Winter Admission	1159 v. 1074	1.20	1.09 to 1.31	<0.001	0.6
Weekend Admission	614 v. 1619	1.08	0.98 to 1.20	0.12	0.1
Hospital (reference 1)				<0.001	2.1
16 v. 350	2.69	1.58 to 4.58	<0.001
350 v. 350	1.19	1.02 to 1.39	0.03
v. 350	1.24	1.01 to 1.53	0.04
618 v. 350	0.86	0.75 to 0.99	0.03
v. 350	1.11	0.94 to 1.31	0.22
252 v. 350	1.18	1.00 to 1.41	0.05

Patient Characteristic	N	β	95% CI	P	% R ²
v. 350	0.86	0.72 to 1.03	0.11

AHLOS, Acute Hospital Length of Stay; CI, Confidence Interval; TIA, Transient Ischaemic Attack; COPD, Chronic Obstructive Pulmonary Disease; mRS, modified Rankin Scale; LACS, Lacunar Anterior Circulation Stroke; PACS, Partial Anterior Circulation Stroke; POCS, Posterior Circulation Stroke; TACS, Total Anterior Circulation Stroke.

For peer review only

Table S6 Multiple linear regression complete case analysis for AHLOS (n=1496, R²=44.7%).

Patient Characteristic	N	β	95% CI	P
Age, y	1496	1.01	1.00 to 1.01	<0.001
Sex, female	781 v. 715	0.98	0.90 to 1.07	0.66
Recurrent Stroke	328 v. 1168	1.06	0.96 to 1.17	0.27
Diabetes Mellitus	259 v. 1237	0.99	0.89 to 1.11	0.91
Dementia	138 v. 1358	1.32	1.13 to 1.53	<0.001
Hypercholesterolemia	264 v. 1232	0.92	0.82 to 1.02	0.13
Myocardial Infarction or Ischaemic Heart Disease*	362 v. 1134	1.00	0.91 to 1.10	0.97
Previous Cancer	140 v. 1356	1.16	1.01 to 1.33	0.03
COPD	76 v. 1420	0.91	0.76 to 1.09	0.31
Pre-stroke mRS Score (reference 0)				<0.001
1	284 v. 765	1.08	0.96 to 1.20	0.21
2	167 v. 765	0.93	0.80 to 1.08	0.33
3	149 v. 765	1.00	0.84 to 1.19	0.99
4 & 5	131 v. 765	0.77	0.63 to 0.93	0.01
Pre-Stroke Residence (reference Independent living without formal care)				<0.001
Independent living with formal care	145 v. 1215	1.02	0.88 to 1.19	0.78
Institution	136 v. 1215	0.83	0.69 to 0.98	0.03
Haemorrhagic Stroke	138 v. 1358	0.83	0.72 to 0.96	0.01
Oxford Community Stroke Project Classification				<0.001
PACS	570 v. 411	1.27	1.15 to 1.40	<0.001
POCS	214 v. 411	1.29	1.13 to 1.47	<0.001
TACS	301 v. 411	1.36	1.19 to 1.57	<0.001
No Brain Lateralisation	174 v. 1322	0.93	0.81 to 1.05	0.24
Inpatient Complication	421 v. 1075	1.67	1.51 to 1.84	<0.001
Discharge mRS Score (reference 0)				<0.001
v. 218	1.15	1.00 to 1.32	0.05
v. 218	1.60	1.36 to 1.88	<0.001
v. 218	2.45	2.10 to 2.87	<0.001
209 v. 218	3.39	2.86 to 4.02	<0.001
121 v. 218	4.78	3.89 to 5.88	<0.001
233 v. 218	1.34	1.11 to 1.61	0.002
Winter Admission	770 v. 726	1.16	1.07 to 1.25	<0.001
Weekend Admission	401 v. 1095	1.06	0.97 to 1.15	0.23
Hospital (reference 1)				<0.001
2	14 v. 111	2.08	1.35 to 3.21	0.001
3	278 v. 111	1.20	1.01 to 1.44	0.04
4	-	-	-	-
5	558 v. 111	0.84	0.71 to 0.98	0.03
6	142 v. 111	1.03	0.85 to 1.26	0.75
7	191 v. 111	1.35	1.13 to 1.62	0.001
8	202 v. 111	0.94	0.78 to 1.13	0.49

AHLOS, Acute Hospital Length of Stay; CI, Confidence Interval; COPD, Chronic Obstructive Pulmonary Disease; mRS, modified Rankin Scale; LACS, Lacunar Anterior Circulation Stroke; PACS, Partial Anterior Circulation Stroke; POCS, Posterior Circulation Stroke; TACS, Total Anterior Circulation Stroke.

For peer review only

Table S7 Multiple linear regression sensitivity analysis for AHLOS, excluding Hospital 2 using multiple imputed dataset (n=2217, R²=44.7%).

Patient Characteristic	e ^β *	95% CI*	P
Age, y	1.01	1.00 to 1.01	<0.001
Sex, female	1.01	0.94 to 1.08	0.86
Recurrent Stroke	1.02	0.93 to 1.12	0.68
Diabetes Mellitus	1.07	0.97 to 1.17	0.19
Dementia	1.30	1.13 to 1.48	<0.001
Hypercholesterolemia	0.95	0.86 to 1.05	0.33
Myocardial Infarction or Ischaemic Heart Disease*	1.00	0.91 to 1.08	0.92
Previous Cancer	1.13	0.99 to 1.27	0.06
COPD	0.90	0.77 to 1.06	0.21
Pre-stroke mRS Score (reference 0)			<0.001
1	1.08	0.96 to 1.21	0.19
2	0.90	0.78 to 1.04	0.16
3	0.94	0.79 to 1.10	0.47
4 & 5	0.69	0.58 to 0.83	<0.001
Pre-Stroke Residence (reference Independent living without formal care)			<0.001
Independent living with formal care	1.01	0.88 to 1.16	0.91
Institution	0.81	0.69 to 0.95	0.01
Haemorrhagic Stroke	0.80	0.71 to 0.90	<0.001
Oxford Community Stroke Project Classification (reference LACS)			<0.001
PACS	1.30	1.18 to 1.43	<0.001
POCS	1.34	1.18 to 1.53	<0.001
TACS	1.29	1.13 to 1.47	<0.001
No Brain Lateralisation	0.85	0.75 to 0.95	0.01
Inpatient Complication	1.70	1.57 to 1.85	<0.001
Discharge mRS Score (reference 0)			<0.001
1.15	1.00 to 1.32	0.04
1.74	1.48 to 2.04	<0.001
2.72	2.34 to 3.16	<0.001
3.56	3.02 to 4.20	<0.001
5.12	4.22 to 6.22	<0.001
1.25	1.05 to 1.48	0.01
Winter Admission	1.15	1.08 to 1.24	<0.001
Weekend Admission	1.03	0.95 to 1.11	0.48
Hospital (reference 1)			<0.001
1.08	0.94 to 1.22	0.29
1.07	0.89 to 1.29	0.40
0.78	0.70 to 0.87	<0.001
0.93	0.81 to 1.07	0.33
1.15	1.00 to 1.32	0.05
0.82	0.70 to 0.95	0.01

AHLOS, Acute Hospital Length of Stay; CI, Confidence Interval; COPD, Chronic Obstructive Pulmonary Disease; mRS, modified Rankin Scale; LACS, Lacunar Anterior Circulation Stroke; PACS, Partial Anterior Circulation Stroke; POCS, Posterior Circulation Stroke; TACS, Total Anterior Circulation Stroke.

For peer review only

Table S8 Multiple linear regression sensitivity analysis for AHLOS, including discharge destination using multiple imputed dataset (n=2233, R²=40%).

Patient Characteristic	e ^β *	95% CI*	P
Age, y	1.01	1.00 to 1.01	<0.001
Sex, female	0.99	0.92 to 1.07	0.80
Recurrent Stroke	1.00	0.91 to 1.10	1.00
Diabetes Mellitus	1.08	0.98 to 1.19	0.12
Dementia	1.20	1.05 to 1.38	0.01
Hypercholesterolemia	0.94	0.85 to 1.04	0.25
Myocardial Infarction or Ischaemic Heart	1.01	0.93 to 1.10	0.83
Previous Cancer	1.17	1.03 to 1.33	0.01
COPD	0.91	0.77 to 1.07	0.23
Pre-stroke mRS Score (reference 0)			<0.001
1	1.15	1.03 to 1.28	0.02
2	1.15	1.00 to 1.33	0.05
3	1.33	1.13 to 1.56	<0.001
4 & 5	1.15	0.96 to 1.38	0.12
Pre-Stroke Residence (reference Independent living without formal care)			<0.001
Independent living with formal care	0.86	0.75 to 0.99	0.04
Institution	0.52	0.44 to 0.62	<0.001
Haemorrhagic Stroke	0.84	0.75 to 0.95	<0.001
Oxford Community Stroke Project Classification (reference LACS)			<0.001
PACS	1.34	1.22 to 1.48	<0.001
POCS	1.44	1.26 to 1.63	<0.001
TACS	1.49	1.31 to 1.70	<0.001
No Brain Lateralisation	0.82	0.73 to 0.93	<0.001
Inpatient Complication	1.72	1.58 to 1.87	<0.001
Discharge Destination (reference Independent living without formal care)			<0.001
Independent living with formal care	1.99	1.74 to 2.27	<0.001
Institution	3.58	3.09 to 4.15	<0.001
Interim/Rehab Setting	2.18	1.94 to 2.46	<0.001
Death	0.85	0.74 to 0.97	0.02
Winter Admission	1.15	1.07 to 1.24	<0.001
Weekend Admission	1.04	0.96 to 1.13	0.30
Hospital (reference 1)			<0.001
2.76	1.80 to 4.22	<0.001
1.24	1.09 to 1.42	<0.001
1.36	1.15 to 1.61	<0.001
0.85	0.75 to 0.95	0.01
1.06	0.92 to 1.22	0.42
1.19	1.03 to 1.37	0.02
0.99	0.85 to 1.14	0.84

AHLOS, Acute Hospital Length of Stay; CI, Confidence Interval; COPD, Chronic Obstructive Pulmonary Disease; mRS, modified Rankin Scale; LACS, Lacunar Anterior Circulation Stroke; PACS, Partial Anterior Circulation Stroke; POCS, Posterior Circulation Stroke; TACS, Total Anterior Circulation Stroke.

Figure S1 Model estimates of mean baseline acute hospital length of stay (AHLOS) per hospital (in days) and number of stroke patients treated outside the stroke unit per day per five stroke unit beds with 95% confidence intervals. Multiple regression model was adjusted for patient covariates that had a p-value<0.3 in univariable analysis.

Figure S2 Model estimates of mean baseline acute hospital length of stay (AHLOS) per hospital (in days) and presence of vascular surgery onsite with 95% confidence intervals. Multiple regression model was adjusted for patient covariates that had a p-value<0.3 in univariable analysis.

Figure S3 Model estimates of mean baseline acute hospital length of stay (AHLOS) per hospital (in days) and distance to neurosurgical facility with 95% confidence intervals. Multiple regression model was adjusted for patient covariates that had a p-value<0.3 in univariable analysis.

Figure S4 Model estimates of mean baseline acute hospital length of stay (AHLOS) per hospital (in days) and number of fte senior doctors per five beds available during weekdays with 95% confidence intervals. Multiple regression model was adjusted for patient covariates that had a p-value<0.3 in univariable analysis.

Figure S5 Model estimates of mean baseline acute hospital length of stay (AHLOS) per hospital (in days) and number of fte junior doctors per five beds available during weekdays with 95% confidence intervals. Multiple regression model was adjusted for patient covariates that had a p-value<0.3 in univariable analysis.

Figure S6 Model estimates of mean baseline acute hospital length of stay (AHLOS) per hospital (in days) and number of fte health care associates and nurses per five beds with 95% confidence intervals. Multiple regression model was adjusted for patient covariates that had a p-value<0.3 in univariable analysis.

Figure S7 Model estimates of mean baseline acute hospital length of stay (AHLOS) per hospital (in days) and number of fte occupational therapists per five beds with 95% confidence intervals. Multiple regression model was adjusted for patient covariates that had a p-value<0.3 in univariable analysis.

Figure S8 Model estimates of mean baseline acute hospital length of stay (AHLOS) per hospital (in days) and number of fte physiotherapists per five beds with 95% confidence intervals. Multiple regression model was adjusted for patient covariates that had a p-value<0.3 in univariable analysis.

Figure S9 Model estimates of mean baseline acute hospital length of stay (AHLOS) per hospital (in days) and number of fte speech and language therapists per five beds with 95% confidence intervals. Multiple regression model was adjusted for patient covariates that had a p-value<0.3 in univariable analysis.

Figure S10 Model estimates of mean baseline acute hospital length of stay (AHLOS) per hospital (in days) and number of total beds present on stroke unit per 100 admissions with 95% confidence intervals. Multiple regression model was adjusted for patient covariates that had a p-value<0.3 in univariable analysis.

Figure S11 Model estimates of mean baseline acute hospital length of stay (AHLOS) per hospital (in days) and number of hospital beds per computed tomography (CT) scanner with 95% confidence intervals. Multiple regression model was adjusted for patient covariates that had a p-value<0.3 in univariable analysis.

Figure S12 Model estimates of mean baseline acute hospital length of stay (AHLOS) per hospital (in days) and provision of onsite rehabilitation service with 95% confidence intervals. Multiple regression model was adjusted for patient covariates that had a p-value < 0.3 in univariable analysis.

Figure S13 Model estimates of mean baseline acute hospital length of stay (AHLOS) per hospital (in days) and presence of early supported discharge scheme with 95% confidence intervals. Multiple regression model was adjusted for patient covariates that had a p-value < 0.3 in univariable analysis.

Figure S14 Model estimates of mean baseline acute hospital length of stay (AHLOS) per hospital (in days) and number of non-stroke patients present on the stroke unit per day per five stroke unit beds with 95% confidence intervals. Multiple regression model was adjusted for patient covariates that had a p-value<0.3 in univariable analysis.

Figure S15 Model estimates of mean baseline acute hospital length of stay (AHLOS) per hospital (in days) and mean Index of Multiple Deprivation (IMD) score of the counties in which the hospital serves with 95% confidence intervals. Multiple regression model was adjusted for patient covariates that had a p-value<0.3 in univariable analysis.

STROBE 2007 (v4) Statement—Checklist of items that should be included in reports of *cohort studies*

Section/Topic	Item #	Recommendation	Reported on page #
Title and abstract	1	(a) Indicate the study's design with a commonly used term in the title or the abstract	1 & 2
		(b) Provide in the abstract an informative and balanced summary of what was done and what was found	2-3
Introduction			
Background/rationale	2	Explain the scientific background and rationale for the investigation being reported	5
Objectives	3	State specific objectives, including any prespecified hypotheses	6
Methods			
Study design	4	Present key elements of study design early in the paper	7
Setting	5	Describe the setting, locations, and relevant dates, including periods of recruitment, exposure, follow-up, and data collection	7-8
Participants	6	(a) Give the eligibility criteria, and the sources and methods of selection of participants. Describe methods of follow-up	7
		(b) For matched studies, give matching criteria and number of exposed and unexposed	NA
Variables	7	Clearly define all outcomes, exposures, predictors, potential confounders, and effect modifiers. Give diagnostic criteria, if applicable	8-10
Data sources/ measurement	8*	For each variable of interest, give sources of data and details of methods of assessment (measurement). Describe comparability of assessment methods if there is more than one group	8
Bias	9	Describe any efforts to address potential sources of bias	10-12
Study size	10	Explain how the study size was arrived at	7
Quantitative variables	11	Explain how quantitative variables were handled in the analyses. If applicable, describe which groupings were chosen and why	8-10
Statistical methods	12	(a) Describe all statistical methods, including those used to control for confounding	11
		(b) Describe any methods used to examine subgroups and interactions	NA
		(c) Explain how missing data were addressed	12
		(d) If applicable, explain how loss to follow-up was addressed	NA
		(e) Describe any sensitivity analyses	11-12
Results			

Participants	13*	(a) Report numbers of individuals at each stage of study—eg numbers potentially eligible, examined for eligibility, confirmed eligible, included in the study, completing follow-up, and analysed	14
		(b) Give reasons for non-participation at each stage	14
		(c) Consider use of a flow diagram	Fig 1
Descriptive data	14*	(a) Give characteristics of study participants (eg demographic, clinical, social) and information on exposures and potential confounders	14
		(b) Indicate number of participants with missing data for each variable of interest	14 &15
		(c) Summarise follow-up time (eg, average and total amount)	NA
Outcome data	15*	Report numbers of outcome events or summary measures over time	16
Main results	16	(a) Give unadjusted estimates and, if applicable, confounder-adjusted estimates and their precision (eg, 95% confidence interval). Make clear which confounders were adjusted for and why they were included	19-21 (and Table S4)
		(b) Report category boundaries when continuous variables were categorized	NA
		(c) If relevant, consider translating estimates of relative risk into absolute risk for a meaningful time period	NA
Other analyses	17	Report other analyses done—eg analyses of subgroups and interactions, and sensitivity analyses	22
Discussion			
Key results	18	Summarise key results with reference to study objectives	23
Limitations			
Interpretation	20	Give a cautious overall interpretation of results considering objectives, limitations, multiplicity of analyses, results from similar studies, and other relevant evidence	23-27
Generalisability	21	Discuss the generalisability (external validity) of the study results	27
Other information			
Funding	22	Give the source of funding and the role of the funders for the present study and, if applicable, for the original study on which the present article is based	28

*Give information separately for cases and controls in case-control studies and, if applicable, for exposed and unexposed groups in cohort and cross-sectional studies.

Note: An Explanation and Elaboration article discusses each checklist item and gives methodological background and published examples of transparent reporting. The STROBE checklist is best used in conjunction with this article (freely available on the Web sites of PLoS Medicine at <http://www.plosmedicine.org/>, Annals of Internal Medicine at <http://www.annals.org/>, and Epidemiology at <http://www.epidem.com/>). Information on the STROBE Initiative is available at www.strobe-statement.org.